# The sex-specific factor SOA controls dosage compensation in *Anopheles* mosquitoes

Agata Izabela Kalita[1], Eric Marois[2,6], Magdalena Kozielska[3], Franz J. Weissing[3], Etienne Jaouen[2], Martin M. Möckel[1], Frank Rühle[1], Falk Butter[1,4], M. Felicia Basilicata[1,5,6] & Claudia Isabelle Keller Valsecchi[1,6 ✉]

The *Anopheles* mosquito is one of thousands of species in which sex differences play a central part in their biology, as only females need a blood meal to produce eggs. Sex differentiation is regulated by sex chromosomes, but their presence creates a dosage imbalance between males (XY) and females (XX). Dosage compensation (DC) can re-equilibrate the expression of sex chromosomal genes. However, because DC mechanisms have only been fully characterized in a few model organisms, key questions about its evolutionary diversity and functional necessity remain unresolved[1]. Here we report the discovery of a previously uncharacterized gene (*sex chromosome activation* (*SOA*)) as a master regulator of DC in the malaria mosquito *Anopheles gambiae*. Sex-specific alternative splicing prevents functional SOA protein expression in females. The male isoform encodes a DNA-binding protein that binds the promoters of active X chromosomal genes. Expressing male SOA is sufficient to induce DC in female cells. Male mosquitoes lacking SOA or female mosquitoes ectopically expressing the male isoform exhibit X chromosome misregulation, which is compatible with viability but causes developmental delay. Thus, our molecular analyses of a DC master regulator in a non-model organism elucidates the evolutionary steps that lead to the establishment of a chromosome-specific fine-tuning mechanism.

Malaria is a life-threatening disease, with 241 million cases and 627,000 deaths reported by the World Health Organization in 2021 (ref. 2). It is caused by *Plasmodium* parasites and is transmitted most effectively by mosquitoes of the *A. gambiae* species complex. Mosquitoes are sexually dimorphic, with only females being able to take blood and thereby transmit malaria. However, despite the high relevance of understanding the molecular basis of sexual dimorphism in *Anopheles*, the onset and development of sexually distinct gene-expression pathways have been little studied to date.

*Anopheles* mosquitoes have heteromorphic sex chromosomes, in which males are XY and females are XX. Sex chromosomes generally evolve from a pair of ancestral autosomes, a process in which the Y chromosome typically becomes highly degenerated and is left with only few functional genes[1]. One of the Y-linked genes in *A. gambiae* is the master-switch gene of sexual differentiation *Yob*, which triggers maleness[3]. Along with sex chromosome differentiation, some species evolve DC, which corrects the expression imbalance of the X chromosomal genes (one in males compared with two in females; ZZ/ZW are not discussed here for simplicity)[1]. Transcriptome studies performed at the pupal and adult stages have revealed complete DC of the single male X chromosome in several *Anopheles* species[4–7].

Fruit flies and *Anopheles* mosquitoes belong to the same insect order Diptera. Their X chromosomes evolved independently but from the same ancestral autosome; hence, their X chromosomes and the encoded genes are similar[8,9]. *Drosophila melanogaster* is one of only three model organisms for which the molecular cascades that mediate DC have been elucidated[10]. The master regulator of *Drosophila* DC, the male-specific lethal 2 protein (MSL2) is only present in males. MSL2 recruits the MSL complex to the X chromosome, where the deposition of histone H4 lysine 16 acetylation (H4K16ac) contributes to an approximately twofold increase in gene expression. Loss of any MSL complex subunit causes male-specific lethality[11]. Conversely, ectopic expression of MSL2, but none of the other MSL subunits, is sufficient to induce X chromosome upregulation in females, which can trigger lethality[11,12].

Although *A. gambiae* and *D. melanogaster* have similar X chromosomes and both exhibit X chromosome upregulation, mosquitoes do not achieve DC through MSL2 and the H4K16ac pathway[13]. Until now, the genes and mechanisms that mediate DC in *Anopheles* remained unknown.

## *SOA* produces sex-specific isoforms

To uncover *A. gambiae* DC factors, we determined the developmental window of DC onset using RNA sequencing (RNA-seq) (Fig. 1a). We observed a substantial imbalance between the sexes in the expression of X-linked but not autosomal genes shortly after zygotic genome activation (ZGA). This imbalance was compensated by 5–9 h of embryogenesis, with further fine-tuning at later stages. We then searched

[1]Institute of Molecular Biology (IMB), Mainz, Germany. [2]INSERM U1257, CNRS UPR9022, Université de Strasbourg, Strasbourg, France. [3]Groningen Institute for Evolutionary Life Sciences, University of Groningen, Groningen, Netherlands. [4]Institute of Molecular Virology and Cell Biology, Friedrich Loeffler Institute, Greifswald, Germany. [5]Institute of Human Genetics, University Medical Center of the Johannes Gutenberg University Mainz, Mainz, Germany. [6]These authors contributed equally: Eric Marois, M. Felicia Basilicata, Claudia Isabelle Keller Valsecchi. ✉e-mail: c.keller@imb-mainz.de

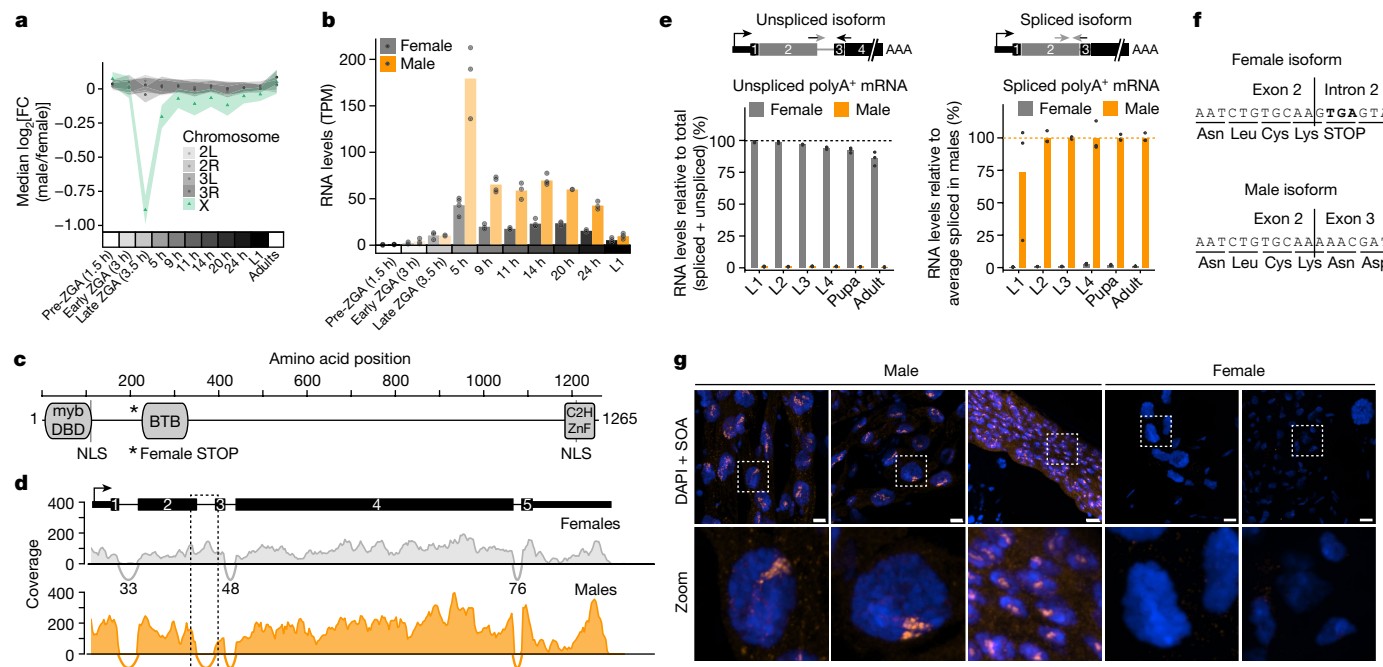

**Fig. 1 | Identification of the sex-specifically spliced *SOA* gene. a**, Dot plot showing the median log₂ fold change (log₂(FC)) of RNA levels between males and females from single-embryo RNA-seq (shading indicates 95% confidence intervals). Genes with read count > 0 were grouped on the basis of chromosomal location. Raw data points and replicate numbers provided in Supplementary Table 3. Adult dataset from ref. 4. L1, first instar larva. **b**, Bar plot showing *SOA* RNA levels from RNA-seq in transcripts per million (TPM). Overlaid data points are biological replicates. **c**, Scheme of the protein domain architecture of SOA. NLS, nuclear localization signal. **d**, RNA-seq coverage and splice junctions (arcs) at the *SOA* locus at 11 h of embryogenesis in females and males. Read numbers spanning respective exon–exon junctions are shown below the arcs (Supplementary Table 1). **e**, RT–PCR quantification of polyadenylated (polyA⁺) *SOA* mRNA isoform levels in females and males at larval (L1–L4), pupal and adult

stages. The scheme (top) shows the primer strategy. Left, percentage unspliced relative to total (spliced and unspliced) mRNA. Right, percentage spliced mRNA relative to the average male spliced mRNA level at each stage. The bars represent the mean of $n = 2$ or $n = 3$ independent biological replicates indicated by overlaid data points. *Rp49* was used for normalization (Extended Data Fig. 4c and raw data in Supplementary Table 1). **f**, Nucleotide and amino acid sequence of the exon 2–intron 2 junction (female isoform) and exon 2–exon 3 junction (male isoform). **g**, Representative SOA immunostaining (orange) and DAPI (blue) conducted on adult mosquito tissues (Malpighian tubules or gut). Images on the bottom row are close-ups of the white square in the above images. Images represent 3D views of a z-stack. Scale bar, 10 μm. Complete panel with single channels and additional staining shown in Extended Data Fig. 5g.

for transcripts that were male-biased from 5 h onwards (Fig. 1b and Extended Data Fig. 1a). This analysis uncovered *Yob*, which encodes the Y-linked, male master sex determination gene[3], and *AGAP005748*, an uncharacterized protein-coding gene that we name after its putative function: *sex chromosome activation* (*SOA*). *SOA* encodes a 1,265 amino acid protein with three predicted domains: a myb DNA-binding domain; a broad-complex, tramtrack and bric à brac (BTB) (also known as POZ) domain; and a C2H2 zinc finger (ZnF) (Fig. 1c). It evolved through a tandem gene duplication event from *AGAP005747*. *SOA* orthologues are present in Anophelinae but not in Culicinae (for example, *Aedes aegypti*) (Extended Data Figs. 1b–h, 2 and 3a,b, Supplementary Table 1 and Supplementary Note 1). The lack of *SOA* in Culicinae is consistent with the absence of heteromorphic sex chromosomes in this subfamily, which therefore obviates the need for chromosome-wide DC.

*SOA* produces two sex-specific, alternatively spliced mRNA isoforms. Males express a canonical transcript, whereas females retain the second intron (Fig. 1d). This pattern is conserved among *Anopheles* (Extended Data Fig. 4a). We performed a gene-specific reverse transcription coupled to PCR (RT–PCR) experiment and found that after ZGA, *SOA* splicing seems identical between sexes, with both isoforms present. Shortly thereafter, a sex-specific pattern is established, which persisted in all post-embryonic stages (Extended Data Fig. 4b). Quantification of the polyadenylated *SOA* mRNA isoforms by quantitative RT–PCR (RT–qPCR) revealed that males express around 100-fold more spliced isoform than females (Fig. 1e, Extended Data Fig. 4c and Supplementary Table 1). Notably, intron retention led to the presence of

an in-frame premature stop codon (Fig. 1f), which is evolutionarily conserved (Extended Data Fig. 4d) and only allows the production of a truncated 229 amino acid protein. We note that this in-frame stop codon could provide an explanation for the lower overall transcript levels in females (approximately 3–6-fold less; Extended Data Fig. 4c), as it could trigger the nonsense-mediated decay pathway[14].

To analyse the SOA protein, we generated an antibody against the amino-terminal myb domain compatible with detecting male and female isoforms (validation in Extended Data Fig. 5a–e; see also Supplementary Table 1 and Methods). Because endogenous SOA was below the detection limit of western blotting, we used mass spectrometry to capture SOA after immunoprecipitation (IP). As predicted, we only detected peptides corresponding to the short SOA(1–229) isoform in females, whereas peptides covering the full-length male SOA(1–1265) protein were exclusively found in males (Extended Data Fig. 5f and Supplementary Table 1). We then performed immunofluorescence (IF) stainings of adult mosquito tissues. SOA localized to a distinct subnuclear territory in males, whereas no specific staining could be detected in females (Fig. 1g; full panel in Extended Data Fig. 5g). The male-specific SOA territory was also observed in imaginal discs of the fourth larval stage 4 (L4) and interphase cells of embryos (Extended Data Fig. 5h–j).

## SOA binds X chromosomal gene promoters

Because localization in a nuclear territory is a hallmark of DC[15,16], we investigated whether SOA is associated with the X chromosome.

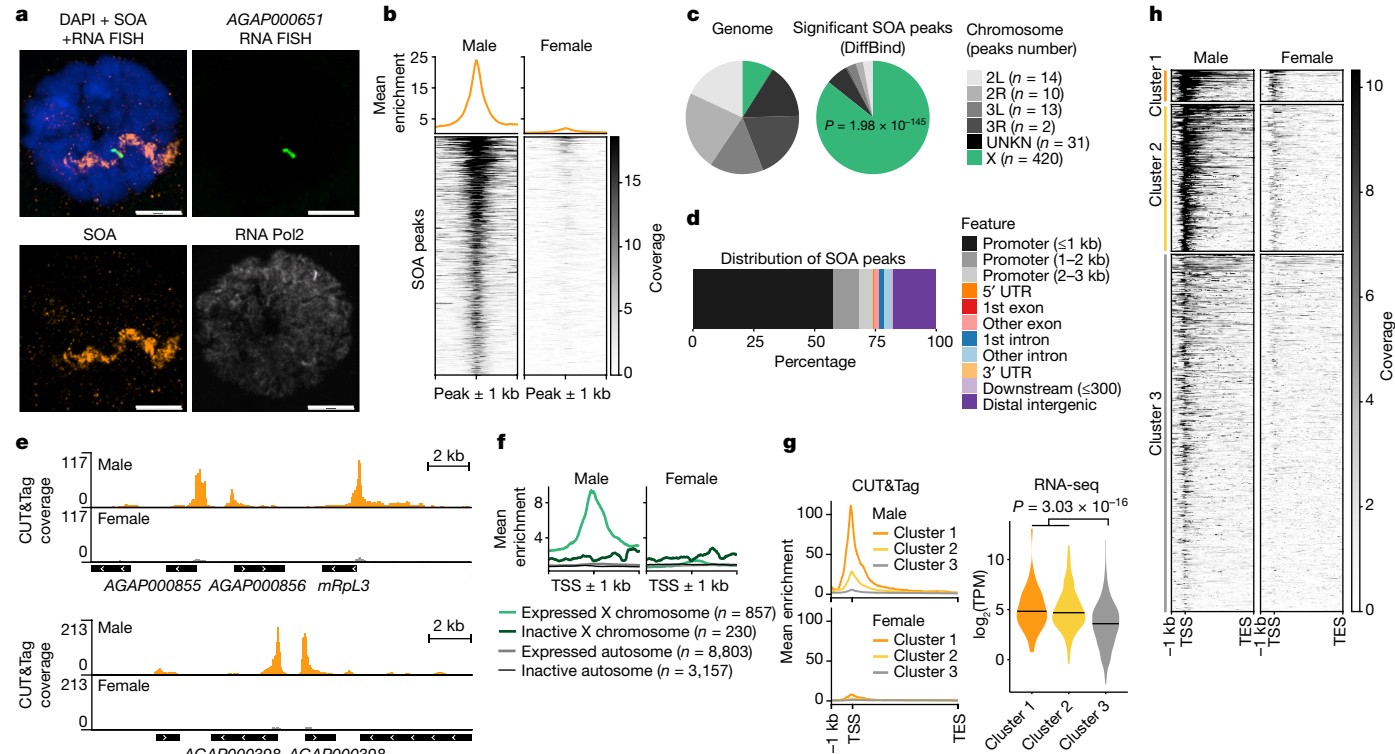

**Fig. 2 | SOA binds to male X chromosomal gene promoters. a**, Representative immunostaining of SOA (orange), RNA polymerase 2 (Pol2; grey) with RNA FISH (green) of a X-linked transcription site (*AGAP000651* intron). DAPI in blue. Scale bar, 10 µm. **b**, Heatmap showing normalized SOA CUT&Tag coverage for significant peaks (males versus females) and metaplot showing mean enrichment (top). **c**, Pie chart of the significant SOA peaks versus the *A. gambiae* genome. *P* value: one-sided Fisher's test for overrepresentation of peaks on the X chromosome. UNKN, scaffolds that could not be assigned to any chromosome. **d**, Bar plot of SOA peak annotations for genomic features. UTR, untranslated region. **e**, Genome browser snapshots of SOA CUT&Tag coverage. **f**, Metaplot of SOA CUT&Tag coverage at the TSS ± 1 kb (all genes). Lines reflect gene groups by chromosomal location and expression levels based on RNA-seq of wild-type male pupae. Genes with fewer than ten average read counts across replicates were considered as not expressed. **g**, Left, metaplot of SOA CUT&Tag coverage at 3 random *k*-means clusters generated from expressed, X-linked genes (*n* = 857 genes, see also **f**). The TSS is a reference point to plot 1 kb upstream; gene bodies (TSS to the transcription end site (TES)) were scaled to 5 kb. Right, violin plot of $\log_2$(TPM) values by RNA-seq of wild-type male pupae. The centre line indicates the median. *P* value: two-sided Wilcoxon rank-sum comparing combined clusters 1 and 2 versus cluster 3. **h**, As in **g**. Heatmap showing the SOA CUT&Tag coverage at expressed X-linked genes. Three random *k*-means clusters were generated that separated the groups on the basis of SOA binding strength. Biological replicates (*n* = 4 male, *n* = 2 female) were merged for visualization (**b**,**e**–**h**).

In stainings of polytene chromosome preparations from L4 larvae, SOA decorated one chromosome of males, but not females (Extended Data Fig. 6a). SOA staining overlapped with the transcription site of the X-linked *AGAP000651*, as visualized by RNA fluorescence in situ hybridization (FISH) and SOA IF (Fig. 2a). To investigate what genomic regions SOA binds to, we used the CUT&Tag method, in which a protein A (pA)–Tn5 transposase fusion protein is directed to an antibody-bound target (SOA) on chromatin[17]. In situ visualization of the DNA sequences tagmented by pA–Tn5 with fluorescent oligonucleotides (CUT&See) revealed an overlap with the male SOA territory by IF (Extended Data Fig. 6b). CUT&Tag sequencing was then performed using male and female pupae with the SOA antibody and an IgG control (Extended Data Fig. 6c and Methods). After differential binding analysis comparing males and females, we identified a total of 490 peaks with significant enrichment in males, but only 39 with significant enrichment in females (Fig. 2b and Supplementary Table 2). In total, 420 of the male-specific peaks were localized to the X chromosome (Fig. 2c and Extended Data Fig. 6d). The majority of them were found at gene promoters, typically residing within 1 kb of the transcription start site (TSS) (Fig. 2d,e and Extended Data Fig. 6e). Because DC is expected to affect expressed, but not inactive genes, we grouped all *A. gambiae* genes on the basis of their chromosomal location and expression status. Using this approach, which is independent of peak calling, we observed SOA binding exclusively at the promoters of X-linked expressed genes (*n* = 857), but at

none of the other three groups (Fig. 2f). Further analysis of these 857 genes by unsupervised clustering distinguished them on the basis of the strength of SOA binding: *n* = 50 genes with strong binding, *n* = 230 genes with intermediate binding and *n* = 577 genes with weak binding (Fig. 2g,h). Cluster 3 (weak SOA binding) showed significantly lower RNA expression levels compared with cluster 1 and cluster 2 genes (Fig. 2g and Supplementary Table 3). To identify DNA sequence motifs bound by SOA, a MEME motif analysis of SOA peaks was performed. Three motifs were enriched, of which a simple CA dinucleotide repeat sequence was the most significant (Extended Data Fig. 6f). Last, investigation of the few autosomal peaks bound in males showed that they display specific but reduced enrichment levels (Extended Data Fig. 6g,h). Most of these peaks were located to genes close to telomeres (Supplementary Table 2). We speculate that the spatial proximity to the X chromosome territory could cause their binding.

## Male SOA is sufficient to induce DC

Having established that SOA specifically binds the X chromosome, we set out to assess its effect on gene expression and asked whether it is sufficient to induce DC. To this end, we ectopically expressed either the male or female isoform in a cell line without DC; that is, female Ag55 cells (Fig. 3a). We performed RNA-seq (Extended Data Fig. 6i and Methods) and found that after expression of the female SOA(1–229)

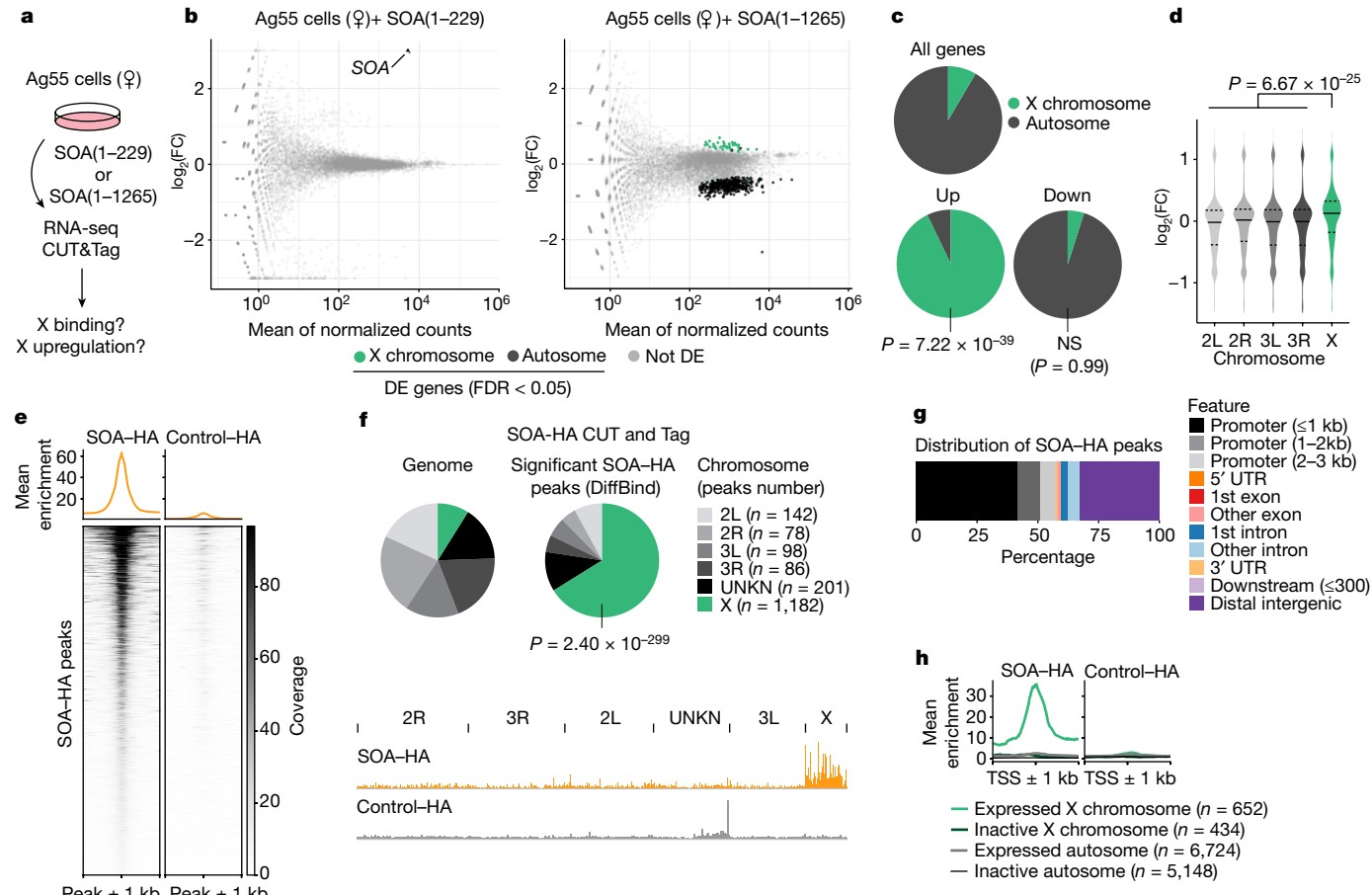

**Fig. 3 | Expression of male SOA is sufficient to induce DC. a**, Scheme illustrating transient expression of female isoform (SOA(1–229)–HA), male isoform (SOA(1–1265)–HA) or empty vector control with baculovirus in female Ag55 cells. **b**, MA plots from RNA-seq (*n* = 3 biological replicates) showing normalized read counts versus log2(FC) comparing SOA(1–229) with empty vector control (left) or SOA(1–1265) with SOA(1–229) (right). Differentially expressed (DE) genes are green (X chromosome) or black (autosomes), others are grey. Arrow indicates *SOA* (triangle) and cistronic *eGFP* (circle). FDR, false discovery rate. **c**, As in **b**. Pie charts of differentially expressed and all *A. gambiae* genes. *P* value: one-sided Fisher's test for overrepresentation of X-linked genes. NS, not significant. **d**, As in **b**. Violin plot of log2(FC) values of female Ag55 cells with SOA(1–1265). The centre line indicates the median. All genes with average read count > 0 were plotted. Median log2(FC) for

X-chromosomal genes equals 0.122 (FC = 1.088). *P* value: two-sided Wilcoxon rank-sum test comparing X-linked versus autosomal genes. **e**, Heatmap showing normalized CUT&Tag coverage on significant peaks in Ag55 cells expressing SOA(1–1265) versus empty vector control (*n* = 2 biological replicates merged for visualization) and mean enrichment as a metaplot. **f**, As in **e**. Top, pie chart of significant CUT&Tag peaks. *P* value: one-sided Fisher's test for overrepresentation of peaks on the X chromosome. Bottom, genome browser snapshot of CUT&Tag coverage. **g**, As in **e**. Bar plot of SOA–HA peak annotations for genomic features. **h**, As in **e**. Metaplot of CUT&Tag coverage at the TSS ± 1 kb (all genes). Lines reflect gene groups by chromosomal location and expression levels based on RNA-seq of empty vector control Ag55 cells. Genes with fewer than ten average read counts across replicates were considered as not expressed.

isoform, there was only a single differentially expressed gene compared with the empty vector control-*SOA* itself (Fig. 3b and Extended Data Fig. 6j). By contrast, ectopic expression of male SOA(1–1265) induced a global upregulation of X chromosomal genes (Fig. 3b,c), irrespective of whether a gene was scored as differentially expressed or not (Fig. 3d). The differentially expressed genes upregulated by SOA were almost exclusively X-linked (Fig. 3c). This was accompanied by the downregulation of many genes on autosomes, probably as a secondary consequence of perturbed transcription regulators encoded on the X chromosome (for example, *AGAP000189*; Supplementary Table 2).

To analyse the SOA binding pattern in this ectopic system, we performed CUT&Tag using the HA tag present in our constructs (Extended Data Fig. 7a and Methods). A total of 1,787 peaks were scored significant for being more strongly bound by SOA(1–1265) compared with the empty vector control (Fig. 3e). Out of these, 1,182 (66%) localized to the X chromosome (Fig. 3f). As in the in vivo context (Fig. 2d,f), SOA–HA associated with active X chromosomal promoters (Fig. 3g,h and Extended Data Fig. 7b) and showed substantial enrichment at highly

expressed genes (Extended Data Fig. 7c,d). Motif analysis also revealed binding to CA repeats (Extended Data Fig. 7e). Overall, the binding profiles of endogenous SOA in tissue and SOA–HA in cells were similar (Extended Data Fig. 7f,g). The improved signal-to-noise ratio explains the higher total number of significant peaks called in cells, whereas the non-endogenous *EF1a* promoter used in that context appeared to cause some spillover to autosomal genes, at which endogenous SOA is not found (Extended Data Fig. 6g,h).

We investigated whether SOA localization depended on an RNA co-factor such as roX1/roX2 (ref. 16) or Xist[18]. However, the SOA territory localization observed by IF remained intact after treatment with RNase A (Extended Data Fig. 7h). Similarly, X chromosome binding of SOA was insensitive to transcription inhibition by actinomycin D (Extended Data Fig. 7i,j). To investigate the potential involvement of a DNA-guided mechanism in X chromosome recruitment, we directed our attention towards the CA-repeat motif. First, we used the Repeat-Masker annotation to analyse the distribution of repeats on the different chromosomal arms (Extended Data Fig. 8a–d). Second, we used

the FIMO tool to search the top-scoring $(CA)_7$ motif sequence in *A. gambiae* in comparison to *A. aegypti* (no DC, therefore used as a control) (Extended Data Fig. 8e,f). The RepeatMasker approach revealed that the X chromosome per se is repeat-rich (Extended Data Fig. 8a). Moreover, simple repeats such as $(CA)_n$ sequences were not only highly abundant, but were among the repeat families that are enriched on the X chromosome (Extended Data Fig. 8d). Both RepeatMasker and FIMO analyses showed that compared to autosomes, the frequency and length of X-linked CA repeats were significantly higher (Extended Data Fig. 8b,c,f). Such features are not observed in *A. aegypti*[19] (Extended Data Fig. 8e,f), which indicated that the SOA-bound motif is specific to the *Anopheles* X chromosome.

Next, we investigated how the different SOA protein domains (Extended Data Fig. 8g–i) contribute to CA-repeat binding. We used electrophoretic mobility shift assays (Extended Data Fig. 8j,k) and fluorescence polarization (Extended Data Fig. 8l) to quantify the binding affinity of recombinant SOA(1–112) (which contains the myb domain), SOA(1–331) (which contains the myb and BTB domains) and SOA(1195–1265) (which contains the ZnF domain) to CA-containing and non-CA-containing DNA sequences. The myb DNA-binding domain, but not the ZnF domain, associated with DNA in vitro (Extended Data Fig. 8j,l). In line with the fact that oligomerization provided by BTB domains can confer stable chromatin association[20], the DNA-binding property of the myb domain was enhanced in the presence of BTB (for $CA_{10}$ dsDNA, $K_d = 59$ µM for SOA(1–112) compared with $K_d = 40$ nM for SOA(1–331)). Size-exclusion chromatography coupled to multi-angle light scattering confirmed the oligomerization function of the BTB domain, as SOA(1–122) and SOA(1–229) appeared as monomers, but SOA(1–331) was present in monomeric and multiple oligomeric species (Extended Data Fig. 8m). Nonetheless, in this in vitro setup with isolated domains, none of the fragments showed specificity towards CA-containing compared with non-CA containing sequences. To explore this effect in vivo, we expressed a SOA mutant without the myb domain in Ag55 cells and performed CUT&Tag (Extended Data Fig. 8n–p). In comparison to full-length SOA, SOA without the myb domain showed a substantial reduction in X chromosome association that was close to background levels.

## Compromised DC in *SOA* mutant males

To understand its physiological roles, we generated transgenic mosquitoes that lack *SOA* by virtue of a CRISPR-mediated targeted knock-in in front of the *SOA* coding sequence (Extended Data Fig. 9a and Methods). The transgenic line, referred to as *SOA-KI*, was made homozygous and then verified by PCR and RT–qPCR (Extended Data Fig. 9a,b). The RT-qPCR assay showed substantially decreased *SOA* RNA levels in these mosquitoes. In CUT&Tag, the enrichment at male-specific SOA-binding sites was lost in *SOA-KI* compared with the wild-type mosquitoes (Fig. 4a and Extended Data Fig. 6d,e,g,h). IF showed that localization of SOA to the X chromosome territory was lost in *SOA-KI* males (Fig. 4b). RNA-seq analyses of gene expression changes (Extended Data Fig. 9c,d) revealed global downregulation of the X chromosome in SOA mutant males (Fig. 4c and Extended Data Fig. 9e). This result confirms that SOA mediates DC in vivo. Out of the 204 downregulated genes scored as differentially expressed (Supplementary Table 2), 164 were X-linked ($P = 6.73 \times 10^{-54}$, Fisher's exact test). We also analysed the expression changes in the three groups of genes that exhibited strong, intermediate and weak SOA association in CUT&Tag (clusters in Fig. 2g). The reduced gene expression in *SOA-KI* males correlated with the strength of SOA binding in wild-type males (Fig. 4d). Genes from cluster 1 with strong SOA binding were notable (median fold change of 0.608) providing support for a role for SOA in DC.

To investigate whether this effect is associated with changes in chromatin accessibility, we performed assay for transposase-accessible chromatin with sequencing (ATAC–seq) in wild-type and *SOA-KI* mosquitoes (Extended Data Fig. 9f,g). The accessibility of X-linked promoter regions remained unchanged, regardless of RNA expression changes in *SOA-KI* mosquitoes (Extended Data Fig. 9h) or direct SOA binding (Extended Data Fig. 9i). Furthermore, the male and female X chromosome displayed comparable accessibility (Extended Data Fig. 9j), which suggested that SOA binding at the TSS does not change the level of promoter opening per se, but presumably affects features after pre-initiation complex loading[21].

We next examined the phenotypic consequences of SOA loss. Homozygous *SOA-KI* mosquitoes of both sexes were viable and fertile. However, in a mixed mosquito culture of *SOA-KI* and wild-type genotypes, the mutant allele frequency diminished over time, which indicated a fitness defect (Fig. 4e; heterozygous *SOA-KI* males showed no phenotype). Of note, unlike the wild-type mosquitoes, adult male SOA mutants tended to emerge after females, which indicated a sex-specific developmental delay. Accordingly, a gene ontology (GO) term analysis of the differentially expressed genes based on RNA-seq revealed an enrichment of mitochondrial function and organization, oxidative phosphorylation and metabolic processes (Extended Data Fig. 9k and Supplementary Table 2). To quantify the developmental delay, we sorted neonate wild-type and *SOA-KI* larvae of both sexes ($n = 100$ for each of the 4 genotypes) and monitored their development in the same mixed culture. We precisely scored the timing of the appearance of pupae for all four genotypes indicating the time required to complete the larval stages (scheme in Fig. 4f). Male *SOA-KI* pupae emerged on average 4 h later than the wild-type males, whereas there was no effect on the development of the females (Fig. 4f, right, and Extended Data Fig. 9l).

## Impact of ectopic *SOA* in female mosquitoes

We next wanted to explore the physiological consequences of expressing the male SOA isoform in female mosquitoes. In this transgenic line, referred to as *SOA-R* (for rescue), the spliced SOA(1–1265) cDNA (male isoform) was integrated immediately upstream of the *SOA-KI* cassette. The rationale behind this strategy was to express SOA in both sexes from its endogenous promoter while rescuing the loss-of-function condition in males (Fig. 5a). The transgenic *SOA-R* line was made homozygous and showed the same *SOA* mRNA expression levels in both sexes, which was slightly higher than the endogenous *SOA* mRNA levels in males (Fig. 5b and Extended Data Fig. 10a). In IF stainings of *SOA-R*, both sexes exhibited a subnuclear SOA territory, which overlapped with the transcription site of the X-linked *AGAP000651* (Fig. 5c and Extended Data Fig. 10b,c). SOA CUT&Tag corroborated that ectopic X chromosome binding was induced in female *SOA-R* pupae (Fig. 5d and Extended Data Fig. 10d,e). The majority of peaks were localized to the X chromosome (Fig. 5e), overlapped with the ones found in wild-type males (Extended Data Fig. 10f) and were more enriched at highly expressed genes (Extended Data Fig. 10g,h).

We performed RNA-seq (Extended Data Fig. 10i) and found that *SOA-R* females displayed a significant overrepresentation of X-linked genes among the upregulated population (upregulated, 300 on the X chromosome, 531 on autosomes, $P = 6.49 \times 10^{-43}$; downregulated, 51 on the X chromosome, 1,003 on autosomes, $P = 0.9998$, Fisher's exact test; Fig. 5f). The increase in RNA levels was most notable at genes with strong binding in CUT&Tag (cluster 1, median fold change of 1.53; Extended Data Fig. 10j), but significant upregulation was also observed when all expressed X-linked genes were taken into account (Extended Data Fig. 10k,l). We analysed the *SOA-R* transgenic line for developmental delay by scoring the timing of pupation. Compared with the parental *SOA-KI* line, the *SOA-R* males developed equally fast as the wild-type line. This rescue of the loss-of-function phenotype confirms the functionality of the *SOA-R* cDNA and that the *SOA-KI* phenotype was not caused by off-target mutations. By contrast, the *SOA-R* females showed a significant developmental delay of a few hours in comparison to all other genotypes (wild-type controls and *SOA-R* males) (Fig. 5g and Extended Data Fig. 10m).

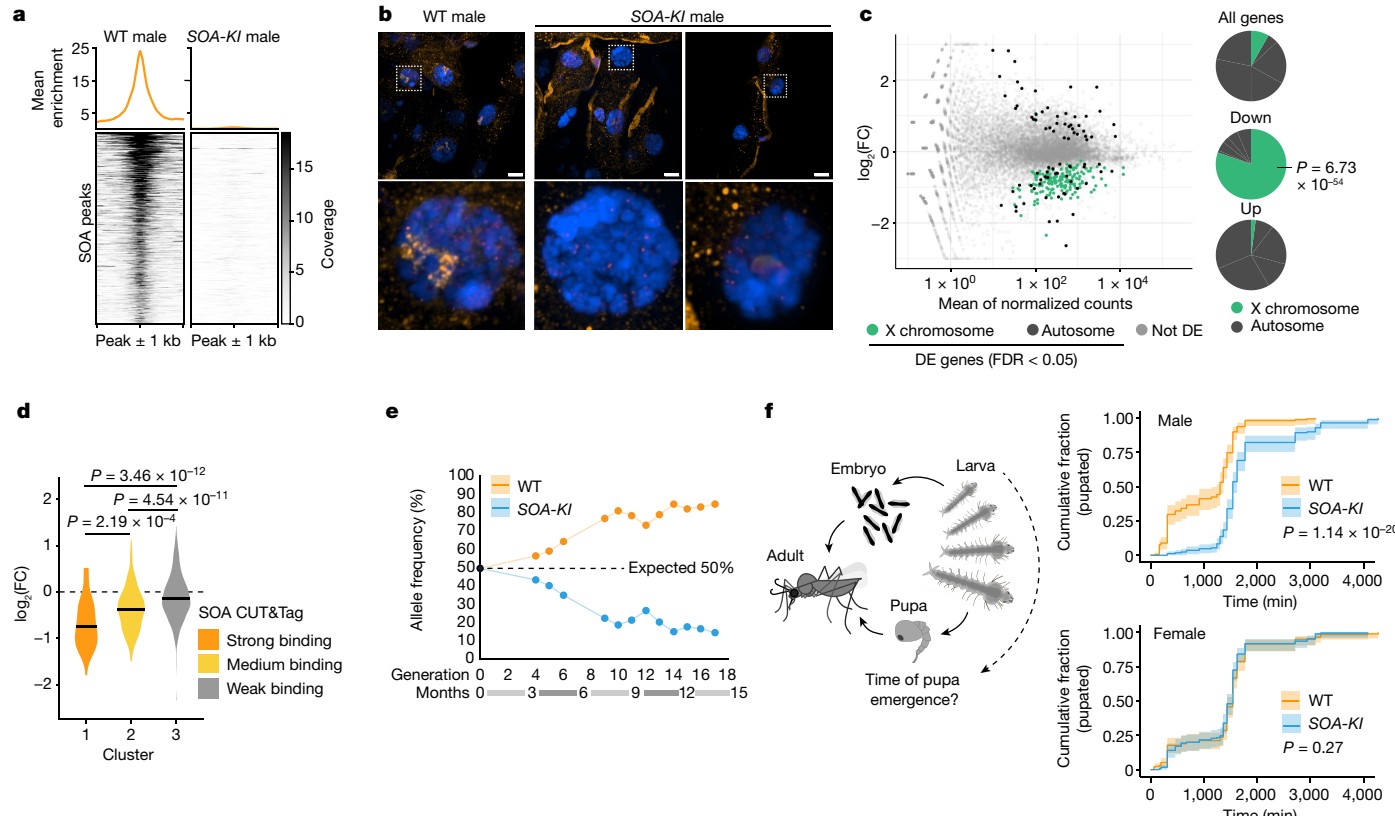

**Fig. 4 | Loss of SOA-mediated DC leads to a male-specific developmental delay. a**, Heatmap showing normalized CUT&Tag coverage in male wild-type (WT) and homozygous *SOA-KI* pupae (*n* = 4 and *n* = 2 biological replicates, respectively; merged for visualization) at significant peaks with binding in males > females. Metaplot (top) show mean enrichment. Datasets for Figs. 2 and 4 were generated together. **b**, Representative SOA immunostaining (orange) and DAPI (blue) conducted on WT and homozygous *SOA-KI* male adult mosquito Malpighian tubules. Images on the bottom row are close-ups of the white square in the top row. Images represent 3D views of a *z*-stack. Scale bar, 10 μm. **c**, Left, MA plots from RNA-seq showing normalized read counts versus log₂(FC) comparing WT with homozygous *SOA-KI* male pupae (*n* = 4 biological replicates). DE genes are green (X chromosome) or black (autosomes), others are grey. Right, pie charts of DE and all *A. gambiae* genes. *P* value: one-sided Fisher's test for overrepresentation of X-linked genes. **d**, As in **c**. Violin plot of log₂(FC) values obtained by DESeq2 analysis of RNA-seq in *SOA-KI* versus WT male pupae. Centre line indicates the median. X-linked genes with average read count > 0 were plotted and split into 3 groups according to the SOA-binding strength (Fig. 2g,h). Bonferroni-corrected *P* values: two-sided Wilcoxon rank-sum test; underlying data provided in Supplementary Table 3. **e**, Line plot illustrating allele frequencies observed in a mixed rearing of WT and *SOA-KI* transgenic mosquitoes (*n* = 1 population). Dashed line shows expected 50:50 allele frequencies. Raw values in Supplementary Table 1. **f**, Left, schematic of *Anopheles* development. Right, line plot (average of *n* = 4 replicate cultures with 95% confidence intervals) of developmental timing of WT and homozygous *SOA-KI* quantified as a cumulative distribution of pupa emergence over time. Each replicate culture reflects 100 neonate larvae of each genotype seeded for development through the larval stages (L1–L4). *P* value: log-rank test for stratified data (Mantel–Haenszel test), second independent experiment in Extended Data Fig. 10a.

In view of these results, we wanted to investigate how a developmental difference of only a few hours can explain the spread and fixation of the *SOA* allele in ancestral *Anopheles*. We considered the standard one-locus model for differential selection in the two sexes[22]. The fitness of males and females in a primordial *SOA*-less state was standardized to one. According to *Anopheles*-specific models, a 4-h acceleration in male development corresponds to a selection coefficient of $s_m = 0.0177$ in males (Methods), yielding a relative fitness of $1 + s_m = 1.0177$ of *SOA*-bearing males (assuming that *SOA*⁺ is dominant over *SOA*⁻ in males). *SOA* would spread relatively rapidly and eventually reach fixation if it had no negative fitness effects in females (Fig. 5h, first panel). However, the results of the *SOA-R* transgenic line imply that before the 'invention' of alternative splicing, *SOA* was detrimental in females, as its presence may have led to dosage imbalance by overexpression of the entire X chromosome (Fig. 5f). This result is in line with the strict conservation of sex-specific splicing among Anophelinae, thereby preventing the expression of a full-length SOA protein in females (Extended Data Fig. 4a). We therefore assumed that the relative fitness of *SOA*-bearing females is $1 - s_f$ in homozygous females and $1 - h_f s_f$ in heterozygous females. The model predicts that the

*SOA* allele will still spread until stable coexistence with the *SOA*⁻ allele is obtained, unless the selection coefficient $s_f$ in females is much higher than the selection coefficient $s_m$ in males (Fig. 5h and Extended Data Fig. 10n). When both alleles are present in the population, any factor alleviating the negative effect of *SOA* in females (such as alternative splicing, marked with an asterisk in Fig. 5h) will lead to the rapid fixation of *SOA* in the population, irrespective of how large the fitness benefit is in males.

## Discussion

The expression of SOA in females is controlled through sex-specific alternative splicing, which parallels the regulatory mechanism of *msl-2* in *Drosophila*[23]. The female sex determination factor SXL binds to an alternatively spliced intron to prevent *msl-2* RNA export and translation. In contrast to MSL2, truncated *Anopheles* SOA protein was detectable in females by mass spectrometry, but it did not accumulate on the X chromosome and is nonfunctional for DC. A female protein present already during early embryogenesis could prevent intron 2 excision. One potential candidate is the sex determination factor *Femaleless* (*Fle*),

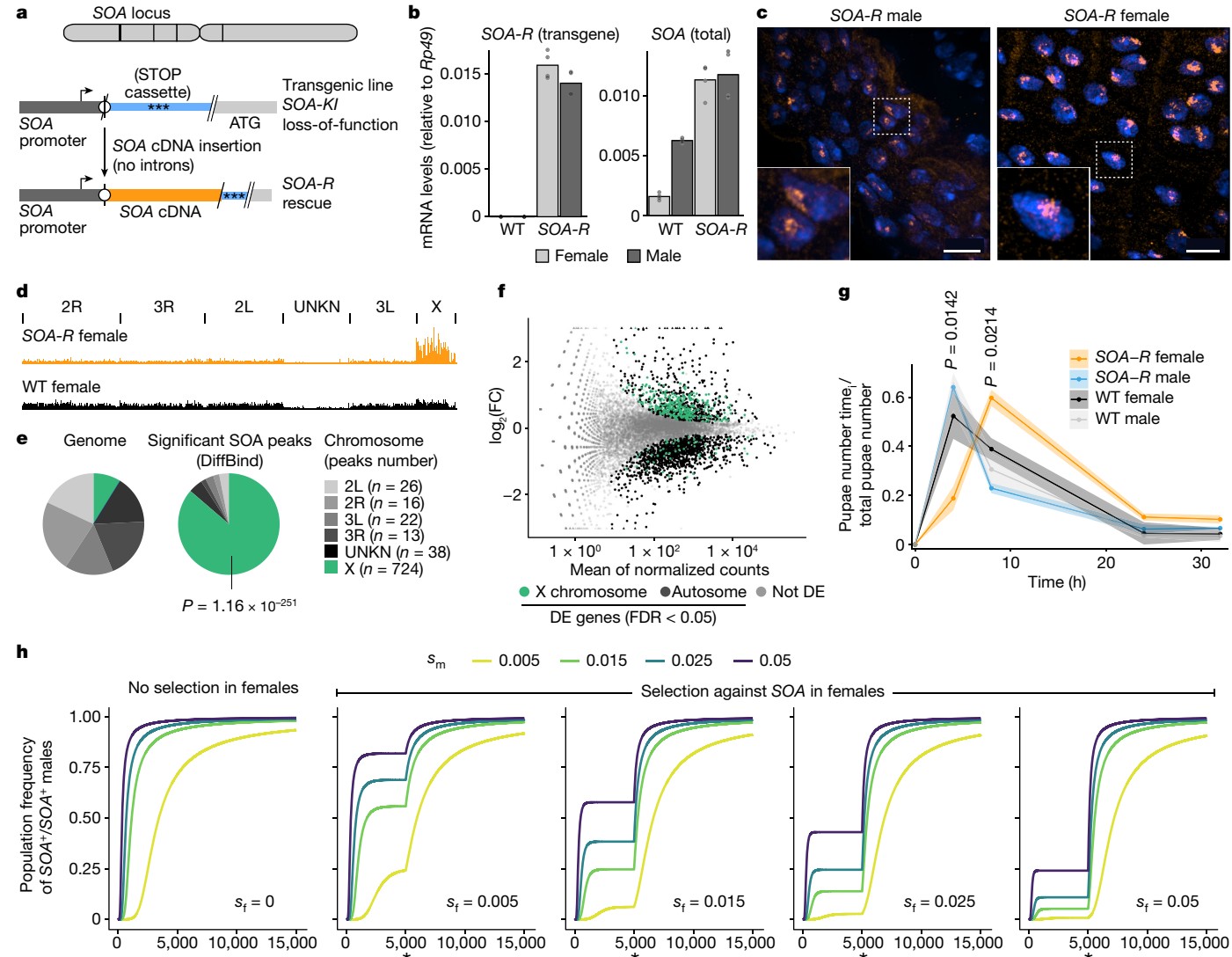

**Fig. 5 | Spliced SOA isoform expression in female mosquitoes results in ectopic DC. a**, Scheme outlining the strategy to create *SOA-R* transgenic mosquitoes. The *att*P landing site (circle) in the *SOA-KI* cassette was used to insert the *SOA* coding sequence. **b**, Bar plots (height: mean of *n* = 4 biological replicates) showing *SOA* mRNA levels normalized to *Rp49* in WT and homozygous *SOA-R* pupae measured by RT–qPCR. Left, expressed from the *SOA-R* cassette (SV40 terminator in the 3′ UTR). Right, total *SOA* mRNA. **c**, Representative SOA immunostainings (orange) and DAPI (blue) conducted on homozygous *SOA-R* male and female adult guts. Images on the bottom left are close-ups of the white square in the main images. Images represent 3D views of a *z*-stack. Scale bar, 10 µm (also see Extended Data Fig. 10c). **d**, Genome browser snapshot of SOA CUT&Tag coverage in homozygous *SOA-R* and WT female pupae (*n* = 2 biological replicates, merged for visualization). **e**, As in **d**. Pie charts of the

significant CUT&Tag peaks versus the *A. gambiae* genome. *P* value: one-sided Fisher's test for overrepresentation of X-linked genes. **f**, MA plot from RNA-seq showing normalized read counts versus log₂(FC) comparing homozygous *SOA-R* (*n* = 4 biological replicates) with WT female pupae (*n* = 3). DE genes are green (X chromosome) or black (autosomes), others are in grey. **g**, Line plot (average of *n* = 3 replicate cultures with shaded areas indicating the s.e.m.) of developmental progression of *SOA-R* quantified by pupa emergence over time. Benjamini–Hochberg-corrected *P* values: two-sided *t*-test with pairwise comparisons between the genotypes. Only significant *P* values (*SOA-R* versus WT females) shown. All data in Supplementary Table 1. **h**, Model predictions of the evolution of *SOA*. $s_m$, fitness increase of *SOA⁺* versus *SOA⁻/SOA⁻* males. $s_f$, fitness decrease of *SOA⁺/SOA⁺* versus *SOA⁻* females. Asterisk indicates evolution of alternative splicing at 5,000 generations.

which contains RNA-binding domains and the knockdown of which in females is associated with misregulation of X-linked transcripts[24]. FLE controls the sex-specific splicing of, for example, *fruitless* or *doublesex*[24], which are well conserved among insects[25]. Thus, SOA may have hijacked pre-existing sequences from such genes after duplication from its non-sex-specific paralogue.

By directly associating with the X chromosome, SOA joins a small list of master regulators that are sufficient to induce chromosome-wide expression alterations (MSL2 in *D. melanogaster*[12], SDC-2 in *Caenorhabditis elegans*[26] and Xist in mammals[18]). Unlike the *Drosophila* MSL complex, which initially targets high-affinity sites and then spreads to

X-linked genes, SOA directly binds the promoters of active genes. Specificity may involve cooperative binding at CA dinucleotide repeats in a similar fashion as for *Drosophila* GAGA factor (GAF). GAF contains a BTB domain important for selecting proper GAF target sites, despite the relatively high abundance of individual GAGA motifs across the genome[20]. The SOA myb-BTB fragment alone is not sufficient for distinguishing CA sequences. We propose that co-factor recruitment through the carboxy-terminal part of SOA probably contributes to faithful target site recognition. After SOA recruitment to X-linked promoters, transcription itself (for example, pause release or elongation[21]) or co-transcriptional RNA processing events[27] may be altered to achieve DC.

In *Anopheles*, the loss of DC in males or its ectopic induction in females was associated with developmental delay. This effect differs from mutants in the sex determination pathway, which show sex reversal, sterility or lethality of variable penetrance[3,24,28]. The expression of *Guy1*, the Y-linked maleness gene in *Anopheles stephensi*, confers complete female-specific lethality accompanied by an upregulation of X-linked genes[29]. The molecular functions of Guy1 and Yob are not known yet, but our data showed that SOA directly binds to the X chromosome and that interfering with its function is not lethal. We favour a model in which Guy1 and Yob induce SOA, but also other yet to be identified factors, the latter of which or their combination with X-misregulation, is causal to lethality after their ectopic expression in females.

It is unclear why DC is essential in organisms such as *Drosophila*, but non-essential in *Anopheles*, whereas many species with heteromorphic sex chromosomes (for example, birds) do not exhibit chromosome-wide DC at all[1,10]. Despite an imbalance in X chromosomal expression already at early embryogenesis[30], *msl* mutants of *Drosophila* are viable for about 6 days and only die when they reach late larval/early pupal stages[31]. In *roX1/roX2* mutants, there are even rare survivors that reach adulthood[32]. Indeed, the molecular activities of the DC complexes have been studied in detail in model organisms, but the physiological consequences of their absence and the causation of lethality remain enigmatic. Hypotheses range from misregulation of a few, putative haplo-lethal genes encoded on the X chromosome to a global gene-dosage imbalance that causes perturbation of gene regulatory networks, overload of cellular machineries such as the ribosome and chaperones, leading to proteotoxicity[33]. This dosage-imbalance model attributes lethality to the degree of disequilibrium rather than the identity of X-linked genes. The difference in phenotypic outcome would accordingly be supported by the 2,500 protein-coding genes in *Drosophila* compared with 1,063 in *Anopheles* on the X chromosome, despite similar overall gene numbers[10]. In addition, autosomal retrocopies of X-linked genes could mitigate phenotypic consequences in *Anopheles* by allowing dosage-sensitive genes to evade the X chromosome and thus eliminating the need for DC[34]. Apparently, there is a continuum in phenotypic outcome, whereby non-essentiality may permit the evolution of a DC master regulator despite being beneficial for one sex but reducing the fitness of the other one. Our model predicts that under these circumstances, genes such as *SOA* can be polymorphic, which underscores the importance of a sufficient sampling rate, as DC alleles might be rare in a population. Alternative splicing would then be strongly selected, as it may alleviate or even resolve the conflict, whereupon DC can spread to fixation.

Last, we note that exploiting X chromosome misregulation has been proposed to artificially generate single-sex populations or sex ratio distortion gene drives for vector control programmes[29,35]. Our discovery that induction of the SOA–DC pathway—at least under the conditions studied by us—is not strongly detrimental for females warrants further studies to uncover factors and mechanisms that underlie sex-specific lethality to eventually harness them in malaria vector control programmes.

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

# Methods

## Mosquito rearing and *SOA* mutagenesis

*A. gambiae* mosquitoes were maintained in standard insectary conditions (26–28 °C, 75–80% humidity and 12–12-h light–dark cycle). To obtain the *SOA* mutant, we used the CRISPR–Cas9 system to insert a fluorescent marker cassette (3×P3-mTurquoise2) into the first *SOA* exon. In addition, an *att*P docking site for PhiC31-mediated plasmid integration was included at the start of the fluorescence marker cassette and at a position corresponding to the *SOA* initiator ATG codon to later allow the possibility of rescuing the mutation with a new copy of *SOA* (see below). The knocked-in fluorescent marker cassette was designed with a strong transcription terminator and multiple stop codons to halt the expression of *SOA* at both the transcriptional and translational level. For this, we built a gRNA-expressing and repair template donor plasmid in the pDSARN vector[36] as previously described[37]. This plasmid expressed two gRNAs under the control of the *AGAP013557* U6 promoter, recognizing target sites 5′-GTCAGCAGCCAGCTTGATGC-3′ and 5′-GCATCAAGCTGGCTGCTGAC-3′ in *SOA*. The 5′ and 3′ regions of homology from the *SOA* genomic sequence (each around 1.1-kb long) adjacent to the gRNA target sites were cloned in this plasmid, flanking the 3×P3-mTurquoise marker cassette. The sequence of the resulting genomic insertion is provided in Supplementary Table 1. The plasmid was microinjected into approximately 40–90 min-old embryos of an *A. gambiae* strain expressing *Cas9* in the germline from a YFP-marked transgene[37]. The progeny of surviving injected mosquitoes, backcrossed to WT, was screened for blue fluorescent larvae using a Nikon SMZ-18 binocular microscope equipped with a Lumencor Sola Light engine and CFP excitation and emission filters. Several dozens of mTurquoise-positive larvae were recovered, and the *SOA-KI* line was established from a single founder female. Junctions between the knocked-in synthetic sequence and the genome were amplified by PCR and sequence-verified. Homozygous and heterozygous *SOA-KI* lines were derived by COPAS sorting[38]. To track the natural dynamics of genotype frequencies across generations, the heterozygous (*WT/SOA-KI*) line was left to evolve naturally for >16 generations. At each generation, the entire population of newly hatched neonate L1 larvae was subjected to COPAS analysis to record the numbers of homozygous mutant, heterozygous and WT individuals as scored by the presence and intensity of mTurquoise marker present in the *SOA-KI* allele (WT is not fluorescent). Genetic crosses were used to combine the *SOA-KI* mutation with the T4 sexing transgene expressing GFP from the Y chromosome[39], allowing COPAS sorting of all-male or all-female populations of *SOA-KI* homozygous mutant and control mosquito larva populations for use in biochemistry experiments. To create the *SOA-R* transgenic mosquito line in which the *SOA* mutation is rescued with a *SOA* cDNA sequence encoding the male *SOA* isoform, we constructed a plasmid harbouring a PhiC31 *att*B site immediately preceding the full-length *SOA* coding sequence, itself followed by the SV40 3′ terminator sequence. A 3×P3-DsRed fluorescence marker was included in the plasmid as a transgenesis selection marker downstream of this *SOA* rescue cassette (the sequence of the rescue plasmid is provided in Supplementary Table 1). This plasmid was co-injected with a PhiC31 integrase-encoding helper plasmid[36] at a concentration of 320 and 80 ng µl⁻¹, respectively, in embryos of the *SOA-KI* line. Integration of the entire plasmid into the *SOA-KI att*P site placed the *SOA* male cDNA isoform under control of the endogenous *SOA* promoter. Transgenic mosquitoes were selected based on DsRed expression in addition to CFP, resulting in the *SOA-R* transgenic line. Work with genetically modified mosquitoes was evaluated by Haut Conseil des Biotechnologies and authorized by MESRI (déclaration d'utilisation d'OGM en milieu confiné no. 3243 and agreement no. 3912).

Developmental timing was scored by counting the appearance of pupae over time, starting from the moment when the first pupa appeared in the culture. At each sampling time, the newly formed pupae were removed from the culture.

## Mice

Mice (CD-1 strain) were maintained in social groups of 4–5 individuals in Techniplast 2L type cages (365 × 207 × 140 mm) with Safe Select litter and nest-building wood, paper and cotton materials, 12–12-h dark–light cycle, 22 °C temperature and 50 ± 10% humidity and fed with Safe R04-25 pellets. For mosquito blood feeding, female CD-1 mice (>35 g) were anaesthetized with a mixture of Zoletil (42.5 mg kg⁻¹) and Rompun (8.5 mg kg⁻¹) in 0.9% NaCl solution, according to animal care procedures validated by regional CREMEAS ethics committee and by the French ministry of higher education, research and innovation under the agreement APAFIS no. 20562–2019050313288887 v.3. We complied with all relevant ethical regulations regarding the use of animals.

## Genotyping

Pupae were homogenized in TRIzol (Fisher Scientific, 15-596-026). After adding chloroform and removing the aqueous phase, the phenol–chloroform phase was used for DNA isolation following the manufacturers' instruction manual. PCR was performed with LA Taq HS polymerase (Takara, RR042A). The PCR products were run on a 1% Tris-borate-EDTA (TBE) agarose gel and imaged using ChemiDoc MP v.3 (Bio-Rad).

## RNA isolation, library generation and sequencing

RNA was extracted using TRIzol (Fisher Scientific, 15-596-026) and a Direct-zol RNA MicroPrep Kit (Zymo Research, R2062). For pupa samples, only the aqueous phase formed after phenol–chloroform separation was loaded on the column after mixing with 100% ethanol. NGS library preparation was performed using an Illumina Stranded mRNA Prep Ligation kit according to the Stranded mRNA Prep Ligation Reference Guide (June 2020; document no. 1000000124518 v00). For the Ag55 cell culture RNA-seq, libraries were prepared with a starting amount of 100 ng and 2 µl of ERCC spike-ins (Ambion, 4456740) in a 1:1,000 dilution and amplified in 12 PCR cycles. For the pupa RNA-seq, libraries were prepared with a starting amount of 1,000 ng and 2 µl of ERCC spike-ins (Ambion, 4456740) in a 1:100 dilution and amplified in 10 PCR cycles. Libraries were profiled in a High Sensitivity DNA on a 2100 Bioanalyzer (Agilent technologies), and quantified using a Qubit dsDNA HS Assay kit in a Qubit 2.0 Fluorometer (Life Technologies). Pooled samples were sequenced on a NextSeq 500 High Output, PE for 2× 73 cycles plus 2× 10 cycles for the dual index read.

## RNA-seq data processing and visualization

For *SOA-KI* RNA-seq, the reads were mapped to the ribosomal RNA sequences extracted from the Ensembl AgamP4 genome using the Ensembl AgamP4 annotation (release 48) with STAR (v.2.7.3a) with the following parameters: outFilterMultimapNmax 1000000 outFilterMismatchNoverLmax 0.04 outFilterMismatchNmax 999. Reads mapping to rRNA were discarded, and unmapped reads were used in downstream processing. For the *SOA-R* and Ag55 RNA-seq, trimming and mapping against rRNA were not performed as there were few rRNA reads. In all experiments, the reads were mapped to the Ensembl AgamP4 genome using the Ensembl AgamP4 annotation (release 48) together with lncRNA annotation[40] and experiment-specific sequences (such as elements of the *SOA-KI* or *SOA-R* cassette, or sequences from the baculovirus in the Ag55 experiment to assess infection rates; more information is provided together with the uploaded data in the Genome Expression Omnibus database) with STAR (v.2.7.3a) using the following parameters: outFilterMismatchNoverLmax 0.04 outFilterMismatchNmax 999. Only uniquely mapped reads were used for downstream analysis. Coverage signal tracks (bigWigs) of primary alignments were generated using deepTools (v.3.1.0). Primary alignments were assigned to features using subread (v.1.6.5) with the AgamP4 annotation (release 48) combined with lncRNA annotation[40] as a reference. Differential

expression analysis was performed using DESeq2 (v.1.26.0), and only genes with FDR < 0.05 were considered as differentially expressed. The visualization of the RNA-seq data of *SOA* in *Anopheles gambiae*, *A. arabiensis*, *A. minimus* and *A. albimanus* was obtained using the genome browser tool from VectorBase (https://vectorbase.org).

## CUT&Tag library generation and sequencing

CUT&Tag was performed as previously described[17]. In total, 0.4 million cells were used for each reaction. The pupa experiments were performed with flash-frozen tissue samples, which were homogenized in cold PBS and passed through a cell strainer (Corning, 352235). In the initial pupa experiment (WT and *SOA-KI* male and female pupae), the homogenate was fixed with 0.2% paraformaldehyde (PFA) for 2 min at room temperature. For the *SOA-R* CUT&Tag, no fixation was applied. The cell culture experiments were all performed on freshly collected cells with a native protocol. The antibodies used are listed in the Supplementary Table 4. We used pA–Tn5 prepared by the IMB Protein Production Core Facility and 15 PCR cycles in the library amplification step. Pooled samples were sequenced on NextSeq 500 High Output, PE for 2×75 cycles plus 2×8 cycles for the dual index read.

## CUT&Tag data processing and analysis

Reads were trimmed using cutadapt (v.4.0) to remove Illumina adapter sequences and subsequently mapped to the reference genome with bowtie2 (v.2.4.5). For the WT male versus female pupa experiment, we performed an initial analysis to inspect the antibody specificity and therefore removed the multimapping and duplicate reads. We then called peaks using macs2 (v.2.1.2) with the corresponding IgG samples as controls, which identified 139 and 393 filtered peaks in female replicates 1 and 2, respectively, but 1,025, 653, 627 and 808 filtered peaks in males. Because we could not a priori exclude SOA binding to repetitive regions, we then performed a second analysis, in which multimapping and duplicate reads were retained for peak calling using macs2 (v.2.1.2). Note that CUT&Tag fragments can share exact starting and ending positions because the integration sites are affected by DNA accessibility. Therefore, duplicates observed in CUT&Tag are not necessarily a consequence of overamplification by PCR[41,42]. A greylist was generated on the basis of IgG samples using the R package GreyListChIP (v.1.22.0) and applied for peak filtering in the pupa experiments. This provided 7,742 consensus peaks for downstream analysis with DiffBind (v.3.4) to identify sites that were significantly (FDR < 0.05) differentially bound between samples (results in Supplementary Table 2). Note that the greylist was applied for the pupa datasets and the myb-less experiment in Ag55, whereas no greylist was applied to the long SOA versus empty Ag55 (cell culture) dataset, as this experiment contained almost no background. Background bins instead of library size were used for normalization. Downstream visualization of differentially bound peaks (for example, heatmaps) were generated using deepTools (v.3.5.1). To identify SOA-bound motifs, the sequences of peaks (±200 bp from the summit) with higher binding (FDR < 0.05) in males (pupa) or SOA(1–1265) were extracted using bedtools (v.2.29.2). Peak sequences were then used for motif discovery analysis using MEME-ChIP (MEME v.5.4.1), with the genome sequence as a background. The MEME output was then used in FIMO (v.5.4.1) with default settings and selecting the available metazoan upstream sequences for *A. gambiae* (AgamP4.34_2019-03-11) or *A. aegypti* (AaegL3.34_2019-03-11) databases. Overlapping CA motifs identified by FIMO were merged into a single CA motif using ggRanges. For the analysis of repeats, the RepeatMasker annotation was downloaded from https://www.repeatmasker.org/species/anoGam.html, RepeatMasker open-4.0.5-Repeat Library 20140131. Downstream analysis and statistical tests were performed using R studio.

## ATAC–seq library generation and sequencing

ATAC–seq was performed as previously described[43] with the following changes. The starting material was flash-frozen pupae. After thawing, whole pupae were homogenized in cold PBS and passed through a cell strainer (Corning, 352235). The cell suspension was counted, and 50,000 cells were used for each reaction. We used 250 ng of Tn5 prepared by the IMB Protein Production Core Facility per reaction and 15 PCR cycles in the library amplification step. Pooled samples were sequenced on NextSeq 500 High Output, PE for 2×75 cycles plus 2×8 cycles for the dual index read.

## ATAC–seq data processing and analysis

Reads were trimmed using cutadapt (v.4.0) to remove Illumina adapter sequences and subsequently mapped to the reference genome with bowtie2 (v.2.4.5). We excluded multimapping and duplicate reads from downstream analysis. We then called peaks using macs2 (v.2.1.2). Peaks with a length of at least 100 nt were used in downstream analysis with DiffBind (v.3.6.1) to identify sites that were significantly (FDR < 0.05) differentially bound between samples. Coverage signal tracks were generated using deepTools (v.3.5.1). The replicates were merged for visualization in heatmaps by calculating the mean normalized coverage using WiggleTools (v.1.2.8). multiBigwigSummary (Galaxy v.3.5.1.0.0.) was used to calculate the average scores for 20-kb bins on the merged bigwig files visualized in box plots. Heatmaps used to assess the changes in accessibility of SOA bound peaks or genes downregulated in *SOA-KI* males were generated using deepTools (v.3.5.1).

## qPCR

RNA extracted as per the RNA-seq protocol was used for generating cDNA with oligo(dT) as primers. qPCR was performed with FastStart Universal SYBR Green Master (ROX) mix (Roche, 04913850001) in a 7 µl reaction at 300 nM final primer concentration. We used *SOA* as template and *Rp49* as an endogenous control. *SOA* expressed from the *SOA-R* cassette was specifically detected with a primer targeting a part of the exogenous SV40 terminator included in the mRNA 3′ UTR. Total *SOA* mRNA was detected with primers targeting the coding sequence, which enabled comparisons of *SOA* levels in homozygous *SOA-R* and WT conditions. Cycling conditions as recommended by the manufacturer were applied. We corrected for primer efficiency using serial dilutions.

## RT–PCR

RT–PCR was conducted using a OneStep Reverse Transcription-PCR kit (Qiagen, 210212) according to the user manual. In this kit, the reaction mixture contains all of the reagents required for both RT and PCR. For each reaction, 2 ng of RNA was used with primers for *SOA* binding to exons 2 and 3 (rt15 + rt16, Supplementary Table 5). Hence, RT is primed in a gene-specific fashion from the primer in exon 3. *S7* was used as a loading control (rt01 + rt02). A total of 33 PCR cycles were used for *SOA*, 27 cycles for *S7*. The PCR products were separated on a 2% TBE agarose gel and imaged using ChemiDoc MP V3 (Bio-Rrad). Uncropped gel pictures are provided in Supplementary Fig. 1.

## Cloning of plasmids for baculovirus expression

The expression cassettes for Ag55 cells were cloned into a pFastBac Dual backbone (Thermo Fisher, 10712024) used for baculovirus generation. Plasmids were generated by Gibson assembly and restriction cloning (details can be provided upon request). The *EF1a* promoter (approximately 1 kb upstream of the TSS of *AGAP007405*) was amplified from genomic DNA with primers s047 and s048 (Supplementary Table 5) using LA Taq polymerase (Takara, RR002A). The coding sequence of *SOA* was amplified from cDNA generated from an adult male RNA sample. Primstar GXL (Takara, R050A) was used to amplify the coding sequence from the start codon to the end, excluding the stop codon. The vector expressing *SOA(1–229)* was cloned from the vector with full-length *SOA* coding sequence, as was the vector expressing *SOA(112–1265)* (myb-less). All constructs contain a C-terminal 2×HA tag followed by a T2A cleavage site and eGFP, which enables assessment of the infection rate.

## Generation of baculoviruses

pFastBac vectors with expression cassettes were transposed into the baculoviral genome using chemically competent DH10Bac cells (Thermo Fisher Scientific) according to the manufacturer's protocol. Preparation of the baculoviral genome, transfection/P0 virus generation and P1 virus amplification were performed as described in the Bac-to-Bac manual (Thermo Fisher Scientific), with the exception of using Cellfectin® II transfection reagent and Sf-900 III serum-free medium (Thermo Fisher Scientific).

## Cell culture and baculovirus infections

Ag55 cells provided by M. Adang were cultured in Leibovitz L15 medium with 10% FBS (Gibco, 10270-10,6 lot: 2260092) and 1× penicillin–streptomycin (Gibco, 15140122) at 27 °C, 80% humidity. Ag55 cells were authenticated by RNA-seq. Cells were tested every 6 months for mycoplasma (MycoAlert PLUS Mycoplasma Detection kit, Lonza LT07-701). All tests were negative. For the CUT&Tag experiment, 2 million cells were seeded in a 6-well plate. After 16 h, 600 µl of baculovirus in Sf-900 III serum-free medium was added to the cells. For the RNA-seq experiment, 0.75 million cells were seeded per each well of a 24-well plate. After 16 h, 200 µl of baculovirus in Sf-900 III serum-free medium was added. In both experiments, after 6 h the medium was changed to fresh L15. For the western blotting, 20 million cells were seeded in a 10-cm dish and infected with 6 ml of baculovirus on the next day and the baculovirus was not removed. Cells were collected for further processing 48 h after the addition of the baculovirus.

## Nuclear extracts and IP from Ag55 cells

Cells were collected and washed with PBS. The cell pellet was resuspended in hypotonic lysis buffer (25 mM HEPES, pH 7.6, 10 mM NaCl, 5 mM $MgCl_2$, 0.1 mM EDTA and 1× protease inhibitor cocktail) and incubated on ice for 15 min. Next, NP-40 was added to a final concentration of 0.1% and the cells were vortexed for 30 s. The nuclei were pelleted and washed with sucrose buffer (25 mM HEPES, pH 7.6, 2 mM $MgCl_2$, 3 mM $CaCl_2$, 0.3 M sucrose and 1× protease inhibitor cocktail). The nuclear pellet was then resuspended in HMG-K400 buffer (25 mM HEPES, pH 7.6, 2.5 mM $MgCl_2$, 10% glycerol, 0.2% Tween, 400 mM KCl and 1× protease inhibitor cocktail) and rotated for 30 min at 4 °C. After centrifugation, the supernatant was either used directly for western blotting or for IP with the HA antibody. IP was performed by incubating 0.160 mg of nuclear soluble protein extract with 2 µl of HA antibody overnight. The bound SOA–antibody complexes were captured using Protein G dynabeads (1 h at 4 °C) followed by 3 washes in HMGT-K400 buffer. IPs were eluted by incubation in 2× LDS buffer with 200 mM DTT (37 °C, 10 min). For the SOA antibody IP, chromatin extracts from Ag55 cells infected with male SOA(1–1265), female SOA(1–229) or empty baculovirus control, which are all tagged with a C-terminal 2×HA epitope, were prepared. Cells were fixed in 0.1% PFA and nuclei prepared by using a previously published Nexson protocol[44]. The chromatin was sheared by sonication and diluted into the final IP buffer (0.05% SDS, 125 mM NaCl, 10 mM Tris (pH 8), 1 mM EDTA). Next, 5% of the input was removed and the remaining material was incubated with SOA antibody overnight. The bound SOA–antibody complexes were captured using Protein G dynabeads (1 h at 4 °C) followed by 3 washes in RIPA (25 mM HEPES pH 7.6, 150 mM NaCl, 1 mM EDTA, 1% Triton-X 100, 0.1% SDS, 0.1% DOC and protease inhibitors), 1 wash in LiCl buffer (250 mM LiCl, 10 mM Tris-HCl, 1 mM EDTA, 0.5% NP-40 and 0.5% DOC) and 2 washes in TE buffer. IPs were boiled in 1× Laemmli buffer (95 °C, 10 min).

## SDS–PAGE and western blotting

Proteins were separated by 4–12% NuPAGE gradient gels in 1× MOPS buffer. Gels were transferred to a 0.45 µm PVDF membrane in Tris-glycine transfer buffer with 10% methanol (16 h at 60 mA). Membranes were blocked for 1 h in 5% milk in PBS–0.2% Tween, then incubated with primary antibodies (Supplementary Table 4) overnight at 4 °C. For SOA antibody, 5% horse serum was used as a blocking agent. Secondary HRP-coupled antibodies were used at 1:5,000 dilution for 1 h. Blots were developed using Lumi-Light Western Blotting substrate (Roche, 12015200001) and/or SuperSignal West Femto (Thermo Fisher, 34094) and imaged on a ChemiDoc MP V3 (Bio-Rad). Uncropped western blots are provided in Supplementary Fig. 1.

## Recombinant protein purification

The untagged SOA fragments were generated from $His_6$–GST-3C–SOA expression vectors and used for electrophoretic mobility shift assay (EMSA), size-exclusion chromatography coupled to multi-angle light scattering (SEC–MALS) and antibody generation. $His_6$–GST-3C–SOA fragments (1–122, 1–229 and 1–331) were expressed from pET vectors in *Escherichia coli* (BL21 DE3 codon⁺) overnight at 18 °C using 1 mM IPTG in LB medium. Cells were lysed in lysis buffer (50 mM Tris-Cl pH 8.0, 800 mM NaCl, 1 mM EDTA, 1 mM DTT, 5% glycerol and EDTA-free complete protease inhibitor cocktail) using a Branson Sonifier 450 and cleared by centrifugation (40,000*g*, 30 min at 4 °C). Additional 250 mM NaCl was added to the cleared lysates and a PEI-based precipitation of nucleic acids (0.2% w/v polyethylenimine, 40 kDa, pH 7.4) for 5 min at 4 °C was performed, followed by a second round of centrifugation (4,000*g*, 4 °C, 15 min). Recombinant proteins were affinity-purified from cleared lysates using a NGC Quest Plus FPLC system (Bio-Rad) and a GSTrap HP 5 ml column (Cytiva) following the manufacturer's protocols. Proteins were digested with 3C protease (1:100 w/w) overnight at 4 °C during dialysis in 50 mM Tris-Cl pH 8.0, 800 mM NaCl, 1 mM DTT and 5% glycerol to cleave off the $His_6$–GST tag. Digested proteins were re-run over the GSTrap HP 5 ml column to absorb out the $His_6$–GST, concentrated using Amicon 15 ml spin concentrators (Merck Millipore) and subjected to gel filtration (Superdex 200 16/60 pg in 25 mM Na-HEPES, 800 mM NaCl, 1 mM DTT and 10% glycerol, pH 7.4). Peak fractions containing the recombinant proteins after gel filtration were pooled, and protein concentration was determined by using absorbance spectroscopy and the respective extinction coefficient at 280 nm before aliquots were flash-frozen in liquid nitrogen and stored at −80 °C. The $His_6$–MBP-tagged SOA fragments and $His_6$–MBP control were used in EMSA and fluorescence polarization (FP) experiments. $His_6$–MBP-tagged SOA fragments and $His_6$–MBP control were expressed from a pET vector in *E. coli* (BL21-CodonPlus(DE3)-RIL, Agilent) using LB medium and overnight incubation with 0.5 mM IPTG at 18 °C. Cells were lysed in lysis buffer (30 mM Tris-Cl, 500 mM NaCl, 10 mM imidazole, 0.5 mM TCEP, complete protease inhibitors, 2 mM $MgCl_2$ and 150 U ml⁻¹ benzonase, pH 8.0) using a high-pressure homogenizer (constant systems CF1 at 1.9 kBar). The lysate was cleared by centrifugation (40,000*g*, 4 °C, 30 min) and loaded onto a HisTrap FF 5 ml column (Cytiva) using a NGC Quest Plus FPLC system (Bio-Rad). The column was washed with buffer A (30 mM Tris-Cl, 500 mM NaCl and 10 mM imidazole, pH 8.0), followed by a second wash with buffer A containing 1 M NaCl and a third wash with buffer A containing 25 mM imidazole. Recombinant proteins were eluted by applying a linear gradient of 25–500 mM imidazole (pH 8.0) in buffer A over 15 column volumes. Peak elution fractions were pooled and concentrated using an Amicon 15 ml spin concentrator with 10 kDa cut-off (Merck Millipore). Concentrated proteins were applied to a gel filtration column (Superdex 200 16/60 pg, Cytiva, in 10 mM Na-HEPES pH 7.4, 150 mM NaCl, 1 mM TCEP and 5% glycerol). Peak fractions containing recombinant proteins were pooled and concentrated to 200 µM using an Amicon 15 ml spin concentrator with 10 kDa cut-off. Aliquots of the recombinant proteins were snap-frozen in liquid nitrogen and stored at −80 °C. The recombinant proteins were analysed by SDS–PAGE and visualized by Coomassie staining.

## Antibody generation

Tagless SOA(1–122) was re-buffered in PBS using a PD-10 column (Cytiva) for immunization. Immunization was carried out by Eurogentec using

their polyclonal 28-day speedy programme. For epitope purification of the SOA antibody from the serum, 2 ml sulfolink resin (Thermo Fisher Scientific) was covalently conjugated with 3 mg tagless SOA(1–122) according to the manufacturer's protocol. Next, 10 ml final bleed was incubated with the SOA(1–122)-conjugated sulfolink resin at 4 °C overnight while rotating. After incubation, the resin was washed with PBS containing 0.1% Triton X-100, followed by PBS in a gravity-flow poly-prep column (Bio-Rad). Elution was performed using low pH (100 mM glycine-Cl and 150 mM NaCl, pH 2.3) followed by immediate neutralization of elution fractions with Tris-Cl pH 8.0. The eluted antibody was re-buffered using a PD-10 column (PBS, 0.05% NaN$_3$ and 10% glycerol) and concentrated to 1 mg ml$^{-1}$ using an Amicon spin-concentrator before flash-freezing in liquid nitrogen and storage at −80 °C.

## Antibody validation

To validate the specificity of the SOA antibody described in this study, we performed western blotting comparing female Ag55 cells ectopically expressing full-length SOA(1–1265), SOA lacking the myb-domain epitope or an empty control. The SOA constructs additionally contained a C-terminal HA-tag. This revealed a specific band present in only full-length, but not the two control conditions (Extended Data Fig. 5a), and two nonspecific bands present in all conditions. Note that we were unable to detect endogenous SOA proteins by western blotting from Ag55 cells or from male/female tissues, which is probably due to the low abundance of the SOA protein. We conducted IP experiments with HA antibody or SOA antibody and detected the captured proteins by western blotting with the other antibody (SOA antibody for HA-IP and HA antibody for SOA-IP, respectively; Extended Data Fig. 5b,c). The specific SOA band detected in the input was also enriched by IP. Furthermore, SOA antibody could not recognize a SOA version lacking the myb domain (amino acids 1–112, the epitope used to raise the antibody), whereas the SOA(1–229) fragment (female isoform) could be successfully detected. We also conducted IP experiments with SOA antibody versus IgG control from male pupal extracts. The bound proteins in this endogenous setup were then identified in an unbiased fashion by mass spectrometry (MS) (Extended Data Fig. 5d,e and Supplementary Table 1). SOA was the only protein not detected in the control and displayed by far the highest enrichment relative to the few contaminants, both in terms of the number of identified unique peptides identified ($n = 12, 11, 13$ and 12 for the 4 replicates) as well as the intensity. We also validated the specificity of the antibody by CUT&Tag and IF using the *SOA-KI* loss-of-function mutants as a control. In both cases, the detected signals and peaks vanished (Fig. 4a,b), which directly supports specificity. Last, the CUT&Tag experiment from Ag55 cells expressing HA-tagged SOA(1–1265) was performed in parallel with SOA and HA-tag antibodies. The two profiles (HA antibody, SOA antibody) produced similar profiles (data not shown).

## EMSA

The desired amount of protein was diluted into 10 μl of 1× EMSA buffer (20 mM HEPES-KOH (pH 7.5), 100 mM KCl and 0.05% NP-40). GST or MBP was used as a negative control. The protein amounts were 100 fmol (1×) to 12.5 pmol (125-fold excess over DNA). Next, 100 fmol of the DNA probe (601-sequence, 147 bp[45] or X-chromosome promoter sequences bound by SOA, 300 bp; Supplementary Table 1) was added, incubated at room temperature for 30 min and subjected to gel electrophoresis (1.6% TBE agarose). DNA was stained with SYBR Safe and detected using a Typhoon FLA9500 gel scanner. The experiment was repeated three times with similar results. Uncropped gel pictures are provided in Supplementary Fig. 1.

## SEC−MALS measurement

SEC−MALS measurements were performed at 25 °C in 25 mM HEPES (pH 7.5), 500 mM NaCl and 1 mM DTT as the column buffer using a GE Healthcare Superdex 200 10/300 Increase column on an Agilent 1260 HPLC at a flow rate of 0.5 ml min$^{-1}$. Loading concentrations were 200 μM for the SOA(1–112) and SOA(1–229) fragments and 11 μM for the SOA(1–331) fragment. Elution was monitored using an Agilent multi-wavelength absorbance detector (data collected at 280 and 260 nm), a Wyatt Heleos II 8+ multi-angle light scattering detector and a Wyatt Optilab differential refractive index detector. The column was equilibrated overnight in the running buffer to obtain stable baseline signals from the detectors before data collection. Inter-detector delay volumes, band-broadening corrections and light-scattering detector normalization were calibrated using an injection of 2 mg ml$^{-1}$ BSA solution (Thermo Pierce) and standard protocols in ASTRA 8. Weight-averaged molar mass ($M_w$), elution concentration and mass distributions of the samples were calculated using ASTRA 8 software (Wyatt Technology).

## DNA oligomer interaction measurements in vitro using FP

To generate dsDNA oligonucleotide substrates, Cy5-labelled ssDNA 20-mers were annealed with reverse-complement 20-mer oligonucleotides at 50 μM in TE buffer by heating to 90 °C for 1 min and subsequent incubation on ice (all oligonucleotides synthesized and HPLC-purified by Integrated DNA Technologies, sequences in Supplementary Table 1). Using a 384-well plate (Corning, low-volume, polystyrene, black), Cy5-labelled ssDNA and dsDNA oligonucleotide substrates (5 nM) were incubated with varying concentrations of His$_6$–MBP-tagged SOA fragments or with a His$_6$–MBP control in a total volume of 20 μl FP buffer (10 mM Na-HEPES pH 7.4, 150 mM NaCl, 1 mM TCEP, 0.1 g l$^{-1}$ BSA, 5% glycerol and 0.05% Triton X-100). After 10 min of incubation at 20 °C, FP of the Cy5-labelled oligonucleotides were analysed on a Tecan Spark 20M plate reader at 20 °C (excitation wavelength of 625 nm; emission wavelength of 665 nm; gain of 120; flashes of 15; integration time of 40 μs). Normalized FP values were calculated by subtracting the FP value of each oligonucleotide-only measurement from all conditions that contained variable amounts of the respective recombinant protein. The normalized FP values from three independent experiments, including standard deviations, were plotted using GraphPad Prism 8. EC$_{50}$ values, which serve as a proxy for the binding constant ($K_d$), were determined by applying a four parameter [agonist] versus response fit with variable slope in GraphPad Prism 8 if applicable.

## Sample preparation for MS

Approximately 0.2 ml (dry volume) of sex-separated pupae were homogenized for each replicate in 0.5 ml of cytoplasm isolation buffer (Cell Signaling Technologies, 9038S) using a handheld homogenizer. After 5 min of incubation on ice, the homogenate was cleaned by spinning through a cell strainer (Corning, 352235) on a FACS tube (500$g$ for 5 min). Cell fractionation of nuclei was continued according to the manual using a Cell Fractionation kit (Cell Signaling Technologies, 9038S). The nuclei were resuspended in 0.125 ml of NIB (250 mM NaCl, 50 mM HEPES, pH 7.6, 0.1% IGEPAL, 10 mM MgCl$_2$, 10% glycerol and protease inhibitors complete, Roche). For the antibody validation experiment, NIB contained 600 mM NaCl. This was sonicated using a Bioruptor Plus, 5 cycles on/off (high), 30 s each followed by 5 min of centrifugation at 12,000$g$. The supernatant was quantified using Bradford reagent (Avantor PanReac AppliChem, A6932.0250) and 0.4 mg nuclear protein extract used per replicate with $n = 5$ males and $n = 5$ female extracts used in total. For the antibody validation experiment, $n = 4$ male replicates were used for each condition (SOA antibody, IgG control). Per IP and replicate, 20 μl of Protein G dynabeads (Thermo Fisher, 10004D) were washed 2× with NIB, then incubated with 4 μl of SOA antibody (rabbit polyclonal, clone 87) in 40 μl NIB for 45 min on a wheel. This was washed 2× with NIB and resuspended in 40 μl of NIB, which was then added to the nuclear extracts and incubated for 30 min at 4 °C on a wheel. Unbound proteins were removed by three washing steps with 200 μl NIB. Bound proteins eluted by heating beads in 30 μl

1×LDS buffer (Thermo Fisher Scientific) supplemented with 100 mM DTT for 10 min at 70 °C and 1,400 r.p.m. in a thermomixer (Eppendorf). Proteins were subsequently run on a 4–12% NOVEX NuPage gel (Thermo Fisher Scientific) for 8 min at 180 V in 1× MOPS buffer (Thermo Fisher Scientific). Proteins were fixed and stained with 0.25% Coomassie Blue G-250 (Roth) in 10% acetic acid (Sigma)–43% ethanol (Roth). The gel lane was minced and destained with a 50% ethanol–50 mM ammonium bicarbonate (ABC) pH 8.0 solution. Proteins were reduced in 10 mM DTT–50 mM ABC pH 8.0 for 1 h at 56 °C and then alkylated with 50 mM iodoacetamide–50 mM ABC pH 9.0 for 45 min at room temperature in the dark. Proteins were digested with mass-spectrometry-grade trypsin (Sigma) overnight at 37 °C. Peptides were extracted from the gel using twice a mixture of 30% acetonitrile (VWR) and 50 mM ABC pH 8.0 solution followed by two times with pure acetonitrile, which was ultimately evaporated in a concentrator (Eppendorf) and loaded on an activated self-made C18 mesh (AffiniSep) StageTips[46].

## MS data acquisition and analysis
Peptides were separated on a 25 cm self-packed column (New Objective) with 75 µm inner diameter filled with ReproSil-Pur 120 C18-AQ (Dr. Maisch). The EASY-nLC 1000 (Thermo) column was mounted onto a Q Exactive Plus mass spectrometer (Thermo), and peptides were eluted from the column in an optimized 90 min gradient from 2 to 40% acetonitrile–0.1% formic acid solution at a flow rate of 200 nl min$^{-1}$. The mass spectrometer was operated in a data-dependent acquisition mode with one MS full scan and up to ten MS/MS scans using HCD fragmentation. MS raw data were searched against Anopheles_gambiae.AgamP4. pep.all (15,125 entries) with the Andromeda search engine[47] of the Max-Quant software suite (v.1.6.5.0)[48]. Cys-carbamidomethylation was set as fixed modification and Met-oxidation and protein N-acetylation were considered as variable modifications. Match between run option was activated. Before further processing, protein groups marked with reverse, only identified by site or with fewer than two peptides (one of them unique) were removed.

## IF staining
In our initial IF stainings, tissues were dissected and then fixed in 4% formaldehyde in PEM (0.1 M PIPES (pH 6.9), 1 mM EGTA and 1 mM MgCl$_2$) for 20 min and washed three times with PBS. Samples were blocked for 1 h rocking with freshly prepared 0.5% BSA, 0.3% Triton X-100 in 1×PBS solution. The samples were washed with Basilicata-blocking (BB) buffer (0.5% BSA in PBS–0.2% Tween (Sigma Aldrich, P1379)), followed by overnight incubation with primary antibody (anti-SOA, rabbit polyclonal, 1:300 in BB). Samples were washed three times in BB and then stained with a secondary antibody (Alexa fluorophore-labelled goat anti-rabbit, ThermoFisher, A21430, 1:400 in BB). Samples were thoroughly washed with BB, then with 1×PBS–0.2% Tween. For the embryo staining, 19 h AEL-stage embryos were placed in small baskets (Falcon 40 µm cell strainers, 352340) and dechorionated in bleach (4.8% chlorine) for 1–2 min with visual monitoring of chorion dissolution under a binocular microscope. As soon as chorion disappeared, they were rinsed with PBS followed by fixation in PBS, 4% PFA and 0.1% Triton X-100 for 20 min at room temperature. They were then rinsed 3 times with PBS and then stored in methanol at −20 °C. Before IF staining, the black endochorion was then manually peeled off with a needle under a binocular microscope using a Petri dish with a double-sided tape with embryos submerged in 100% methanol. The peeled embryos were transferred using a 1.5 ml pipette into a 1.5 ml Eppendorf tube containing PBS. Blocking and antibody incubations were performed as for the dissected tissues. During the course of the project, we realized that lower PFA concentrations significantly improved the signal-to-noise of the SOA staining; therefore we changed the fixation step in our protocol to 1% PFA for 15 min. We also noted that prolonged incubation with primary antibody (60–72 h) improved signal-to-noise; for embryos prolonged incubation was crucial to obtain SOA staining. For the RNaseA

experiment, midguts were dissected in PBS and then rinsed 2× with CSK buffer (10 mM PIPES-KOH, pH 7.0, 100 mM NaCl, 300 mM sucrose and 3 mM MgCl$_2$), then incubated for 10 min in CSK, 0.5% Triton X-100 and 1 mg ml$^{-1}$ RNaseA (or control). The midguts were then rinsed 2× in CSK buffer. For each condition, 2 midguts (2 replicates) were then put in 0.15 ml TRIzol for RNA isolation to check the effectiveness of the RNase treatment versus control. Meanwhile, the remaining midguts were fixed with 1% PFA in PEM for 15 min at room temperature and stained as per the standard conditions described above. For actinomycin D treatment, the tissues were dissected and put into 0.5 ml of L15 tissue culture medium, 10% FBS and penicillin–streptomycin. Actinomycin D was added to a final concentration of 5 µg ml$^{-1}$ to half of the samples, the other half was left untreated (control), and both conditions were incubated for 1 h at 26 °C in a tissue culture incubator. The tissues were then fixed in PEM and 1% PFA for 15 min at room temperature and the staining was conducted as described above. As a positive control, we co-stained for phosphorylated RNA Pol2, which has been previously described to increase after actinomycin D treatment[49].

## Polytene chromosome preparations
Fourth instar larva were immobilized on ice for 15–20 min, then they were placed in a drop of 75 mM KCl and the head and abdomen was cut off with an ultrafine dissection scissor and discarded. The thorax was placed in a fresh drop of 75 mM KCl on a glass microscopy slide and the gut and tissues attached to it were gently pulled out with forceps and discarded. The remaining thorax piece containing the imaginal discs and salivary glands was gently opened and placed in a fresh drop of fixative (25% acetic acid, 1% methanol-free PFA in H$_2$O). Imaginal discs and salivary glands immediately turn white and are now easy to spot. They were dissected in approximately 5–7 min under a binocular microscope, attempting to completely remove the fat and cuticle. After 7–8 min, the fixative was removed and a fresh drop of PBS–0.1% Tween containing 1:1,000 of DAPI solution was added. A coverslip was put on the dissected discs and salivary glands and excess solution carefully removed with a Kimtech wipe. The coverslip was gently tapped with the rubber of a pencil while observing squashing under a fluorescent microscope. When spreading was sufficient, the slide was put in liquid nitrogen and the coverslip was flicked off with a razor blade. The slide was then placed in PBS and stored at 4 °C until staining. For the RNA FISH experiment, all solutions described above additionally contained RNasin Ribonuclease inhibitor (Promega N2511) at 1:1,000 dilution.

## Staining of polytene chromosomes
The slides were incubated in a coplin jar containing PBS and 0.4% Triton X-100 for 30 min at room temperature on an orbital shaker set at 220 r.p.m. The slides were rinsed 2× with PBS and 0.1% Tween. The slides were then incubated on the orbital shaker with blocking buffer (PBS, 0.1% Tween, 0.2% BSA and 5% horse serum; filtered) for 30–60 min at room temperature. The slides were placed in a wet chamber, and incubation with primary antibody in blocking buffer (0.25 ml solution, slide covered with Parafilm) was conducted overnight at 4 °C. The slides were washed in a coplin jar on the orbital shaker 3× in PBS and 0.2% Tween. Secondary antibodies were incubated for 1–2 h in a wet chamber at room temperature (0.25 ml of solution, slide covered with Parafilm). The slides were washed in a coplin jar on the orbital shaker 2× in PBS and 0.2% Tween followed by a 15 min incubation with PBS, 0.1% Tween and DAPI (1:1,000) in a wet chamber as for the antibodies. The slides were rinsed with PBS and then mounted with Prolong Gold.

## Co-immunostaining with RNA FISH
Polytene squashes were prepared as described above. RNA FISH was performed according to the manufacturer's protocol for IF followed by smFISH, referred to as the sequential protocol. PBS was prepared from a 5× sterile PBS solution with DEPC water and 1 µl RNaseIn per

50 ml of 1× buffer was added. Slides with squashes were briefly rinsed 2× in PBS, 0.1% Tween and RNAseIn for 10 min and 1× with PBS. Primary antibody in PBS incubation was performed 60–72 h at 4 °C in a humidified chamber. Excess antibody was washed out 3× with PBS followed by secondary antibody incubation in PBS for at least 3 h. Unbound secondary was washed out 2× in PBS and the slide was then crosslinked in 4% PFA–PBS for 10 min at room temperature. Excess of fixative was removed using PBS washes and then the smFISH protocol was started using 1× wash buffer A (SMF-WA1-60-BS, LGC Biosearch Technologies) supplemented with 10% formamide. This was followed by hybridization in Stellaris RNA FISH hybridization buffer (SMF-WA1-60-BS, LGC Biosearch Technologies) supplemented with 10% with formamide containing 125 nM probe mix targeting the introns of the X-linked gene *act5c* (*AGAP000651*, sequences in Supplementary Table 1), which was incubated overnight in a humidified chamber at 37 °C. Excess probe was removed by two washes with wash buffer A, 30 min each at 37 °C, followed by a brief wash in wash buffer B (SMF-WB1-20-BS, LGC Biosearch Technologies). Slides were mounted in Vectashield vibrance with DAPI (H-1800, Vector Laboratories) and imaged after 1 h using Visiscope Microscope, ×63 water objective.

### CUT&See

The protocol was based on the spatial CUT&Tag[50] with the following modifications. pA–Tn5 produced by the IMB Protein Production Core Facility was loaded with pre-annealed oligonucleotides Tn5MErev, Tn5ME-A-ATTO488 and Tn5ME-B-ATTO488. Adult male midguts were dissected, fixed with 0.2% PFA in PEM buffer with RNAseIN (1:1,000) at room temperature for 5 min. The fixation step was quenched with 2.5 M glycine (1:20). After quenching, the midguts were washed 2 times with the CUT&Tag wash buffer (20 mM HEPES pH 7.6, 150 mM NaCl, 0.5 mM spermidine and 1× protease inhibitor cocktail) and rinsed briefly with RNAse-free water. The midguts were then incubated for 5 min at room temperature in permeabilization buffer (0.1% NP40 and 0.05% digitonin in wash buffer) and washed once with the NP40–digitonin wash buffer (0.01% NP40 and 0.05% digitonin in wash buffer). Subsequently, the midguts were incubated overnight with the SOA antibody (1:100 dilution) at 4 °C on a Nutator in the antibody buffer (2 mM EDTA and 0.1% BSA in NP40–digitonin wash buffer). The next day, the midguts were rinsed once with NP40–digitonin wash buffer, then incubated on the Nutator for 1 h at room temperature with the secondary antibody (1:100 dilution of F(ab′)2-goat anti-rabbit IgG (H+L) cross-adsorbed secondary antibody, Alexa Fluor-555; 555A21430 ThermoFisher) in the same buffer. This was followed by a rinse with the NP40–digitonin wash buffer. Next, the pA–Tn5 complex pre-loaded with fluorescently labelled oligonucleotides was added into Dig-300 buffer (20 mM HEPES pH 7.6, 300 mM NaCl, 0.5 mM spermidine, 0.05% digitonin and 1× protease inhibitor cocktail) at a final concentration of 31 nM and incubated for 1 h at room temperature on the Nutator. After a 5-min wash with the Dig-300 buffer, the midguts were incubated in tagmentation buffer (10 mM MgCl$_2$ in Dig-300 buffer) for 1 h at 37 °C. The tagmentation step was stopped by adding EDTA to final concentration of 40 mM and incubating for 5 min on the Nutator. The midguts were finally washed with 1× NEBuffer 3.1 and then stained with DAPI.

### Microscopy

Slides were mounted using ProLong Gold Antifade mountant with DAPI (P36935, Thermo Fisher Scientific), unless otherwise stated, and imaged using a fluorescence spinning disc confocal microscope, VisiScope 5 Elements (Visitron Systems), which is based on a Ti-2E (Nikon) stand and equipped with a spinning disc unit (CSU-W1, 50 µm pinhole; Yokogawa). The set-up was controlled using VisiView 5.0 software, and images were acquired with a ×100/1.49 NA oil-immersion objective (CFI Apo SR TIRF ×100, Nikon) or ×60/1.2 NA water-immersion (CFI Plan Apo VC60x WI) and a sCMOS camera (BSI; Photometrics). 3D stacks of images were recorded for each sample. Confocal imaging was performed using a Stellaris 8 Falcon (Leica Microsystems) confocal system equipped with white light laser. Images (1,552 × 1,552 pixel format, 0.93 pixel size) were acquired using a HC PL APO CS2 ×63/1.40 NA oil-immersion lens, and fluorescence was detected using a detector HyD S for DAPI (emission band 427–460 nm), HyD X for Alexa488 (500–545 nm) and HyD R for Alexa555 (560–730 nm). Tissue images were acquired through 87 slices at 200-nm step intervals using a line accumulation of 3 times. 3D view of the *z*-stacks and image processing were obtained using Imaris software (v.9.9.1). The IF stainings were replicated in at least four independent experiments.

### Modelling the evolution of *SOA*

Our results indicated that the *SOA*⁺ allele speeds up male development by about 4 h. To investigate the evolutionary implications of such a progression of development, we used the standard one-locus-two-alleles model of viability selection, with different viabilities in males and females[22]. In this model, the relative viability of the three genotypes *SOA*⁻/*SOA*⁻, *SOA*⁺/*SOA*⁻ and *SOA*⁺/*SOA*⁺ is $1$, $1 + h_m \times s_m$ and $1 + s_m$, respectively, in males and $1$, $1 - h_f \times s_f$ and $1 - s_f$, respectively, in females. Here $s_m$ is the selection differential in favour of the *SOA*⁺ allele in males, whereas $s_f$ is the selection differential against *SOA*⁺ in females. The factors $h_m$ and $h_f$ denote the degree of dominance of the *SOA*⁺ allele. Throughout, we assumed that *SOA*⁺ is dominant in males ($h_m = 1$) and recessive in females ($h_f = 0$) based on the general finding that selectively favoured alleles tend to be dominant in each sex[51]. However, we also considered other dominance values, and they led to the same conclusion (persistence of the *SOA*⁺ allele at considerable frequencies for a wide range of selection coefficients) as long as $h_m > 0$.

Our estimate of $s_m$ was based on the rationale that a shorter developmental time is favourable for survival to adulthood. According to population models specifically tailored to the life cycle of *Anopheles* mosquitoes[52], the daily survival probability of males is 0.9. Speeding up development by 4 h (which equates to one-sixth of a day) therefore corresponds to a survival benefit of $0.9^{5/6}/0.9 = 1.0177$. We therefore assume that the developmental advance of *SOA*⁺-bearing males translates into the selection coefficient $s_m = 0.0177$. As this is a crude estimate, and sometimes different survival probabilities are used[53], we also considered other values of $s_m$, ranging from 0.005 to 0.05. We also considered a spectrum of selection coefficients $s_f$ in females, ranging from 0 to 0.05. In Fig. 5h, $s_f$ was set to zero in generation 5,000, corresponding to the assumption that alternative splicing (removing the negative fitness effects of *SOA*⁺ in females) had evolved by then.

### Evolutionary analyses, sequence analyses, alignments and visualizations

DNA and protein sequences were retrieved from VectorBase. Protein and DNA alignments were created using Clustal Omega. The pairwise percentage similarity of the SOA domains were obtained in Jalview (v.2.11.2.3). Alignments were visualized with ESPript. Lists of 1:1 orthologues were obtained using the Biomart tool from VectorBase. The *SOA* locus, its syntenic regions in other species and the analysis of its paralogue were obtained from VectorBase. The phylogeny and evolutionary distance calculations were performed using MEGA software (v.7.0). Figures were assembled using Adobe Illustrator and Adobe Photoshop (2021 version).

### Bioinformatic and web resources

The following resources were used: cutadapt (https://github.com/marcelm/cutadapt); Bowtie2 (https://github.com/BenLangmead/bowtie2); macs2 (https://github.com/macs3-project/MACS); WiggleTools (https://github.com/Ensembl/WiggleTools); MEME (https://meme-suite.org/meme/); Gviz (https://bioconductor.org/packages/release/bioc/html/Gviz.html); STAR (https://github.com/alexdobin/

STAR); DiffBind (https://bioconductor.org/packages/DiffBind/); deep-Tools2 (https://deeptools.readthedocs.io/en/latest/); IGV (https://software.broadinstitute.org/software/igv/); R (https://www.r-project.org); DESeq2 (http://bioconductor.org/packages/DESeq2/); Vector-Base (https://vectorbase.org/vectorbase/app); Clustal Omega (https://www.ebi.ac.uk/Tools/msa/clustalo/); ESPript (https://espript.ibcp.fr/ESPript/ESPript/); Nuclear Localization Signal prediction (https://nls-mapper.iab.keio.ac.jp/cgi-bin/NLS_Mapper_form.cgi); IUPRED2 (https://iupred2a.elte.hu/); and DNA binding site predictor for Cys2His2 Zinc Finger Proteins (http://zf.princeton.edu/).

### Statistics and reproducibility

All statistics were calculated using R Studio. In the violin plots, the centre line represents the median and the shape of the violin represents the distribution of underlying data. For all violin plots, $P$ values were obtained using two-sided Wilcoxon rank-sum test (Extended Data Figs. 7j, 3d, 2g and 10h,k), with additional Bonferroni correction in Fig. 4d and Extended Data Figs. 7d 9e and 10j,l. In the box plots, the line that divides the box into two parts represents the median, box bottom, and top edges represent interquartile ranges (IQRs; 0.25th to 0.75th quartile (Q1–Q3)), whiskers represent Q1 − 1.5× IQR (bottom), Q3 + 1.5× IQR (top). Bar plots represent the mean with overlaid data points representing replicates. Results were considered significant at FDR below 0.05. NA, not analysed. For all pie charts, the $P$ value was obtained with a one-sided Fisher's exact test for the overrepresentation on the X chromosome. For these, we compared SOA peaks to an equal number of peaks homogeneously distributed on all chromosomal arms (CUT&Tag, Figs. 2c, 3f and 5e) or analysed overrepresentation of X-linked genes in the upregulated and downregulated group in comparison with an equal number of genes homogeneously distributed on all chromosomal arms (RNA-seq, Figs. 3c and 4c). In Extended Data Fig. 8b, overrepresentation of CA-repeat-containing promoters on the X chromosome and autosomes were compared with all X-linked and autosomal genes. For scoring the developmental delay in Fig. 4f and Extended Data Fig. 9l, $P$ values were obtained by a log-rank test for stratified data (Mantel–Haenszel test). In Fig. 5g, Benjamini–Hochberg-corrected $P$ values were obtained with a two-sided $t$-test with pairwise comparisons between the genotypes. Further details are provided in the figure legends. Further data, DiffBind/DESeq2 and statistical test results are provided Supplementary Tables 1–3. The immunostainings were reproduced with similar results as follows: Fig. 1g and Extended Data Fig. 5g experiment (WT males, females) was conducted 7 times, each with tissues dissected from at least $n = 5$ adults of each sex (biological replicates); Extended Data Figs. 5h and 6a experiments (polytene squash, larval tissues) were conducted 3 times, each with at least 2 slides per sex, for which each slide contained tissues dissected from at least $n = 4$ larvae (biological replicates); Extended Data Fig. 5i,j (embryos) was conducted twice, each with at least $n = 30$ embryos (biological replicates); Fig. 2a experiment (SOA IF and co-FISH) was conducted 2 times with 2 slides each; each slide contained tissues dissected from at least $n = 4$ adults (8 biological replicates per experiment); Extended Data Fig. 6b experiment (CUT&See) was conducted once with tissue dissected from $n = 1$ adult (biological replicate); Extended Data Fig. 7h experiment (RNase A) was conducted 2 times, each with tissues dissected from at least $n = 5$ adults (biological replicates); Extended Data Fig. 7i experiment (actinomycin D) was conducted once with tissues dissected from at least $n = 5$ adults (biological replicates); Fig. 4b experiment ($SOA$-$KI$) was conducted 2 times, each with tissues dissected from at least $n = 5$ adults of each genotype (biological replicates); Extended Data Fig. 10b experiment ($SOA$-$R$ IF and co-FISH) was conducted once with 2 slides, each slide contained tissues dissected from at least $n = 4$ larvae (biological replicates); and Fig. 5c and Extended Data Fig. 10c experiment ($SOA$-$R$) was conducted 2 times, each with tissues dissected from at least $n = 5$ adults of each sex (biological replicates).

### Reporting summary

Further information on research design is available in the Nature Portfolio Reporting Summary linked to this article.

### Data availability

No restrictions apply and all data are available in the manuscript or the supplementary materials. RNA-seq, CUT&Tag and ATAC–seq data have been deposited into the Gene Expression Omnibus database (identifiers GSE210624 and GSE210630). MS data have been deposited into ProteomeXchange through the PRIDE database (project identifier PXD042353). DNA and protein sequences, and the Ensembl AgamP4 genome with the Ensembl AgamP4 annotation (release 48) were retrieved from VectorBase (www.vectorbase.org, publicly available). Metazoan upstream sequences for *A. gambiae* (AgamP4.34_2019-03-11) or *A. aegypti* (AaegL3.34_2019-03-11) databases used in FIMO are publicly available as part of the https://meme-suite.org/meme/tools/fimo search tool. RNA-seq data from ref. 4 is publicly available from the Sequence Read Archive under accession number SRP083856.

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

**Acknowledgements** We thank V. Benes and the EMBL Genecore for the sequencing of the embryogenesis RNA-seq experiment; T. Sharpe and T. Mühlethaler at the Biozentrum Basel Biophysics Core for SEC–MALS; J. Cartano for technical support with the MS experiment; F. Kielisch (IMB Bioinformatics Core Facility) for help with statistical testing; A. Raj and M. Dunagin for help with RNA FISH probe design; J. H. G. Fritz García for help with optimization of the baculovirus experiments; G. Magnarini for technical assistance; A. Gautier and N. Schallon for insectary maintenance and blood feeding of mosquito lines; J. Barau for sharing expertise and reagents; staff at the IMB Microscopy and Genomics core facilities for support; staff at the IMB Protein Production Core Facility for the supply of recombinant enzymes; and M. Adang for gifting the Ag55 cell line. A.I.K. was supported by a Boehringer Ingelheim Fonds PhD Fellowship. The work of M.K. and F.J.W. is supported by the European Research Council (ERC Advanced grant no. 789240). M.F.B. received financial support for the work from the intramural High Potentials Grant programme of the University Medical Center Mainz. The

research of C.I.K.V. is funded by the Deutsche Forschungsgemeinschaft (DFG, German Research Foundation)–Individual Project Grant 513744403, Scientific Network Grant 531902894 and GRK GenEvo 407023052 and Forschungsinitiative Rheinland-Pfalz (ReALity). Mosquito breeding and transgenesis were supported by ANR grants ANR-11-EQPX-0022 and ANR-19-CE35-0007 GDaMO and by funding from INSERM, CNRS, the University of Strasbourg, and contrat triennal 'Strasbourg capitale européenne' 2018–2020. The IMB Genomics Core Facility, the Microscopy Core Facility and the use of the NextSeq500 (funded by the Deutsche Forschungsgemeinschaft (DFG, German Research Foundation)–INST 247/870-1 FUGG) and spinning disc confocal system (VisiScope, 5-Elements, funded by the DFG–INST 247/912-1FUGG), and the confocal laser scanning microscope (Leica Stellaris 8 Falcon, funded by the DFG–497669232) are acknowledged.

**Author contributions** C.I.K.V. conceptualized the study with A.I.K., E.M. and M.F.B. E.M. and E.J. performed mosquito rearing, transgenesis and characterized the phenotype of the *SOA-KI* and *SOA-R* mosquitoes. M.F.B. performed IF, RNA FISH and microscopy. M.K. and F.W. performed the computational modelling of the spread of *SOA*. M.M.M. generated baculoviruses, recombinant proteins, conducted antibody purification and FP. F.B. performed MS. F.R. supported data processing and analyses. A.I.K. performed all other experiments and data analyses, including bioinformatics. C.I.K.V. and M.F.B. provided mentoring and guidance. C.I.K.V. and A.I.K coordinated the study, secured funding and wrote the manuscript with input from all authors.

**Competing interests** The authors declare no competing interests.

## Additional information

**Correspondence and requests for materials** should be addressed to Claudia Isabelle Keller Valsecchi.

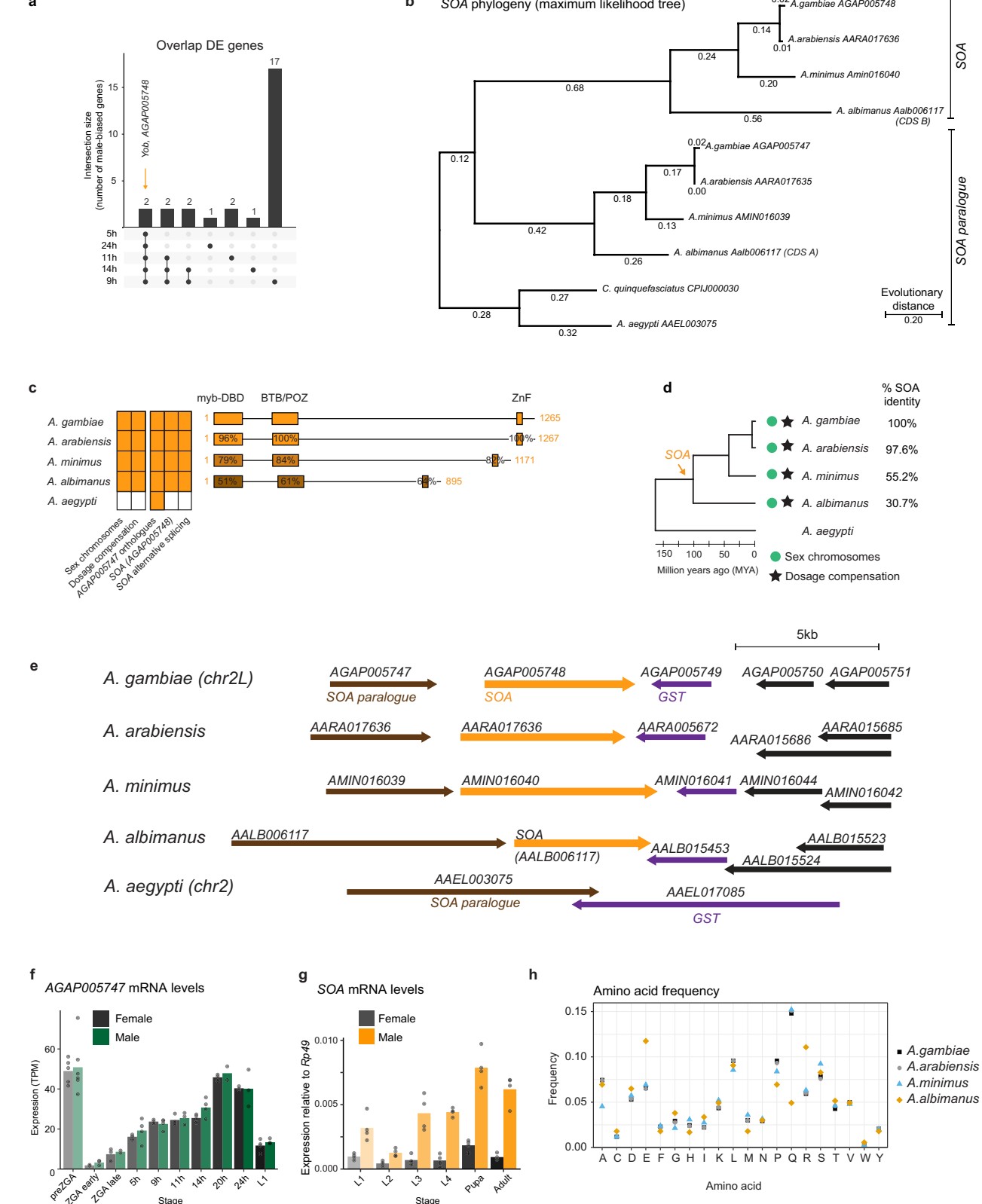

**Extended Data Fig. 1** | See next page for caption.

**Extended Data Fig. 1 | Evolution of *SOA* by a tandem duplication in the *Anopheles* genus.** (**a**) Upset plot showing the overlap between the male-biased differentially expressed (DE) genes obtained at the indicated timepoints in RNA-seq conducted from single male and female *A. gambiae* embryos at various hours (h) after egg laying (also see Fig. 1a). Differentially expressed genes between males and females were obtained with DESeq2. Only two genes were DE at several time-points from early to late embryogenesis, *AGAP005748* (*SOA*) and *AGAP029221* (Yob). (**b**) Maximum-likelihood tree of *SOA* (*AGAP005748*) orthologues and *SOA* paralogues (*AGAP005747*, with respective orthologues). The tree is based on the protein coding DNA sequences of the proteins, aligned with ClustalW in the MEGA 11 software and constructed with the Jones-Taylor-Thornton model. Based on these alignments, a maximum-likelihood tree was generated using default settings. The tree was rooted on the Culicinae outgroup branch. (**c**) Scheme regarding the evolution of *SOA* and its splicing in *Anopheles* genus after its separation from Culicinae. (Left:) Table indicating relevant characteristics for species spanning the *Anopheles* genus and *Aedes aegypti* as an outgroup with no heteromorphic sex chromosomes. (Right:) Schematic illustration of the protein domain architecture of SOA orthologues in the *Anopheles* genus, the conservation level is indicated by percent of identity and shades of respective structured domains. (**d**) Evolutionary tree of 5 representative mosquito species. Length of branches indicates separation of the Anopheline and Culicinae subfamilies based on molecular phylogeny. Additional information on the presence of the *SOA* gene (orange arrow), the presence of sex chromosomes (green dot) and DC (star symbol), as well as the percentage of SOA protein sequence identity (right) is included alongside the tree. (**e**) Synteny of the genomic regions surrounding *SOA* and its orthologues in *Anopheles* and *A. aegypti*. Data was obtained using the synteny tool from VectorBase. All *Anopheles* have both *SOA* and *SOA* paralogues, while *A. aegypti* only contains the paralogue and the *GST* gene in this region. Note that *AALB006117* is mis-annotated as a single gene. However, inspection of the RNA-seq data from[4] clearly reveals two distinct transcription units corresponding to *SOA* and *SOA* paralogue, respectively (also see Extended Data Fig. 4a). (**f**) Barplot showing *AGAP005747* (*SOA* paralogue) RNA levels from RNA-seq in transcript per million (TPM), overlaid data points represent values from biological replicates (single embryos). Raw datapoints and replicate numbers in Supplementary Table 3. No sex bias in expression of *AGAP005747* is observed. (**g**) Bar plot showing *SOA* mRNA levels normalized to *Rp49* in post-embryonic stages measured by RT-qPCR. Height of the bar plot is the mean of *n* = 4 independent experiments as overlaid individual data points. (**h**) Amino acid composition of four SOA orthologues in the *Anopheles* genus. Protein sequences were obtained from VectorBase.

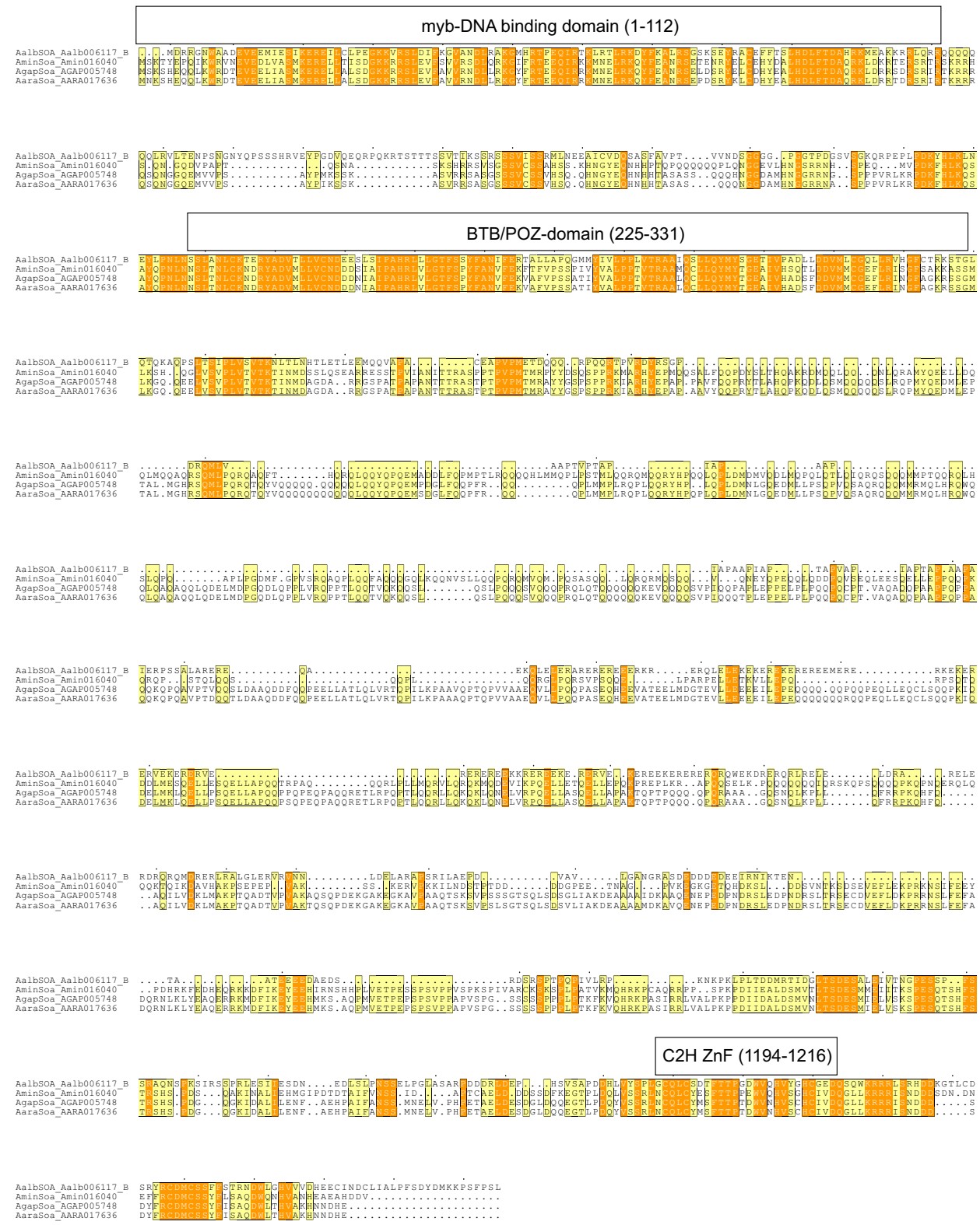

**Extended Data Fig. 2 | Sequence alignment of *SOA* orthologues among four *Anopheles* species.** Alignment of full-length SOA protein sequences in *A. gambiae, A. arabiensis, A. minimus, A. albimanus* with Interpro domain architectures obtained in VectorBase. The alignment was generated in Clustal Omega and visualized with ESPript. Orange shaded residues are conserved in all 4, yellow shaded residues in 3 out of 4 species, respectively.

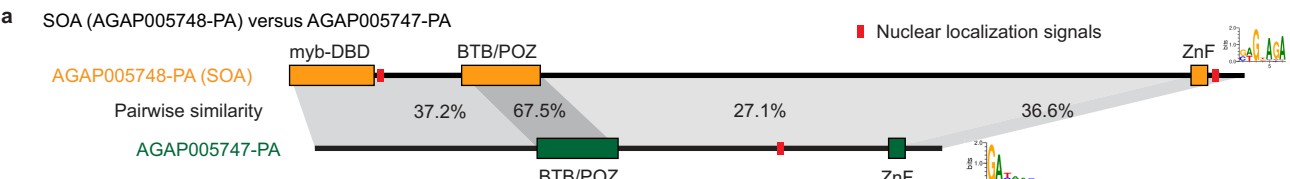

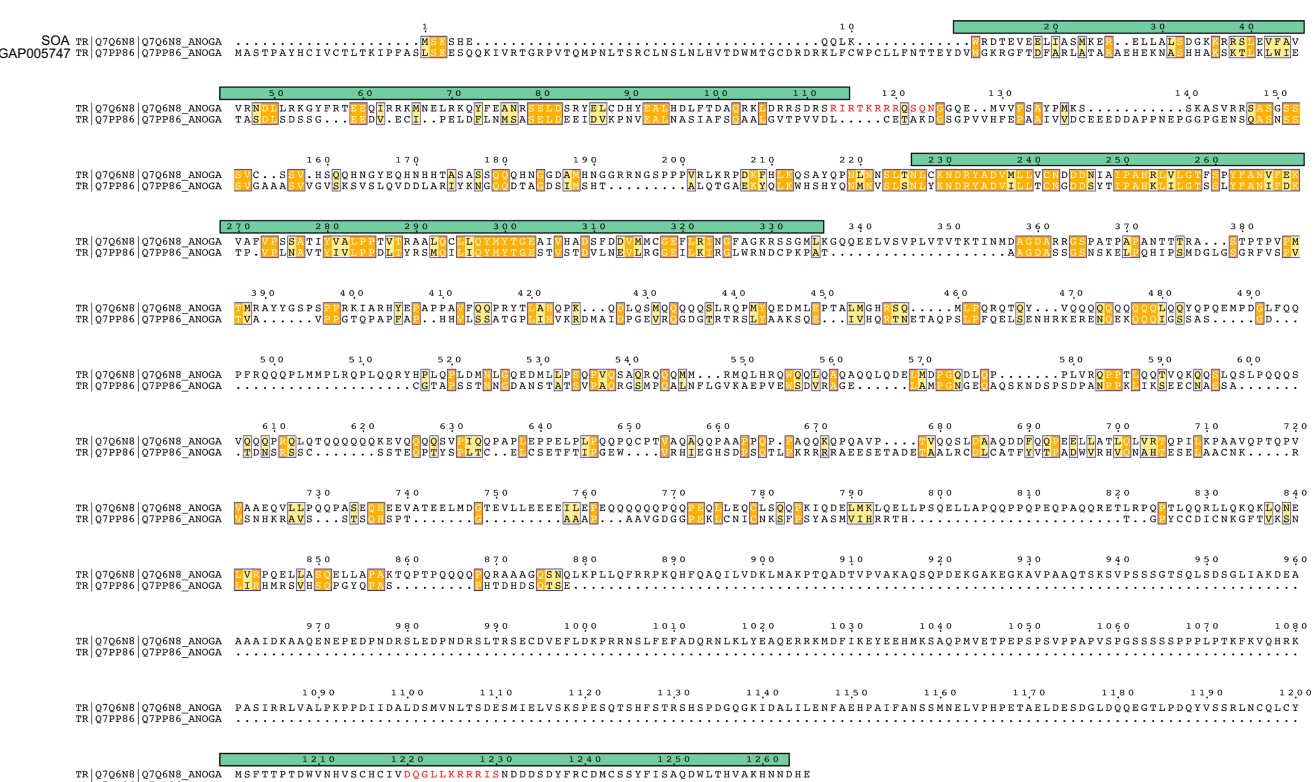

**Extended Data Fig. 3 | Comparison of SOA protein with its paralogue.**
(**a**) Pairwise similarity between SOA (AGAP005748) and AGAP005747 proteins. Protein alignments were generated with Clustal Omega and pairwise similarity obtained in Jalview. Predicted nuclear localization signals are shown in red. The predicted sequence specificity of the C2H2-ZnF domains is shown as a motif logo. (**b**) Sequence alignment of SOA (AGAP005748) with SOA paralogue (AGAP005747) in *A. gambiae*. The alignment was generated in Clustal Omega and visualized with ESPript. Orange shaded residues are conserved, yellow shaded residues are similar, respectively. Green bars indicate the three structured domains of SOA (AGAP005748). Residues in red indicate the nuclear localization signals of SOA.

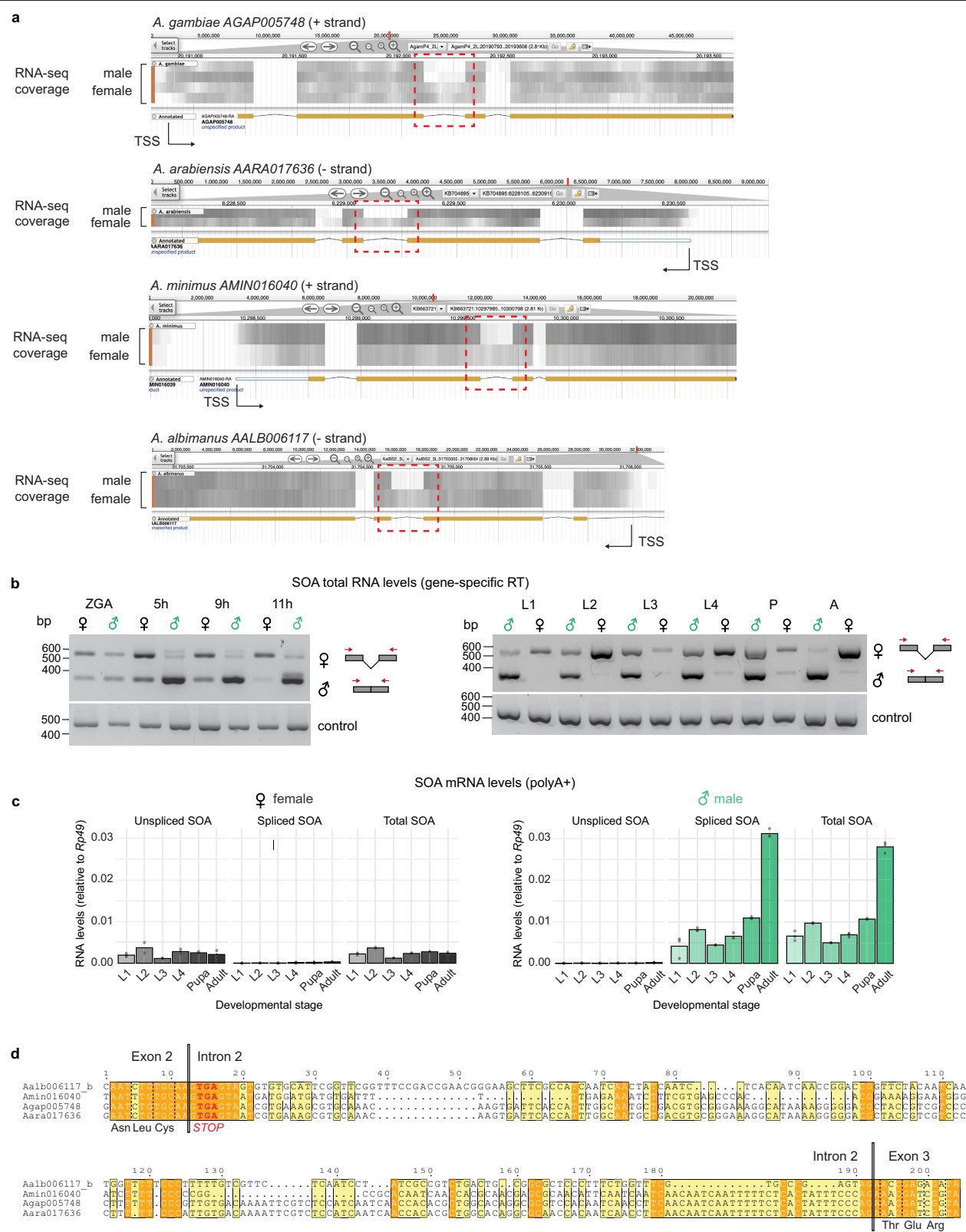

**Extended Data Fig. 4** | See next page for caption.

**Extended Data Fig. 4 | Conservation of SOA alternative splicing in the**
**_Anopheles_ genus.** (**a**) Genome browser snapshots of published RNA-seq data
from adult male and female carcass[4] with RNA-seq coverage represented as
density. The intron 2 is highlighted with a red box, indicating that sex-specific
splicing of intron 2 of SOA orthologues in _Anopheles_ genus is conserved.
In _A. albimanus_, _SOA_ and its paralogue are annotated as one long gene
(_AALB006117_). However, inspection of the RNA-seq data clearly reveals two
distinct transcription units with conserved alternative splicing in _SOA_.
(**b**) Agarose gel showing RT-PCR products of the _SOA_ intron 2 splicing in male
and female (left:) embryos at zygotic genome activation (ZGA), 5h, 9h and 11h of
embryogenesis or (right:) post-embryonic developmental stages: L1-L4 instar
larvae, pupae (P), and adults (A). The reactions were conducted with a one-step
RT-PCR kit, where reverse transcription is primed with the reverse primer in
exon 3. The isoform with retained and excised intron 2 result in long and short
RT-PCR products, respectively. S7 was used as a loading control. The experiment
was conducted twice, results were confirmed with complementary methods:
RNA-seq for embryogenesis and qPCR for post-embryonic stages (Fig. 1d,
Extended Data Fig. 4c, Supplementary Table 1). (**c**) also see (Fig. 1e) qPCR
quantification of polyadenylated SOA mRNA isoform levels in males and females.
The barplot represents the mean levels of spliced, unspliced and total SOA
relative to the _Rp49_ reference gene. Overlaid data points represent the values
of the biologically independent replicates, raw data is provided in Supplementary
Table 1. (**d**) Alignment of pre-mRNAs of _SOA_ (exon2-exon3) in four representative
_Anopheles_ species. Shaded nucleotides are conserved in all 4 species (orange)
or 3 species (yellow), respectively.

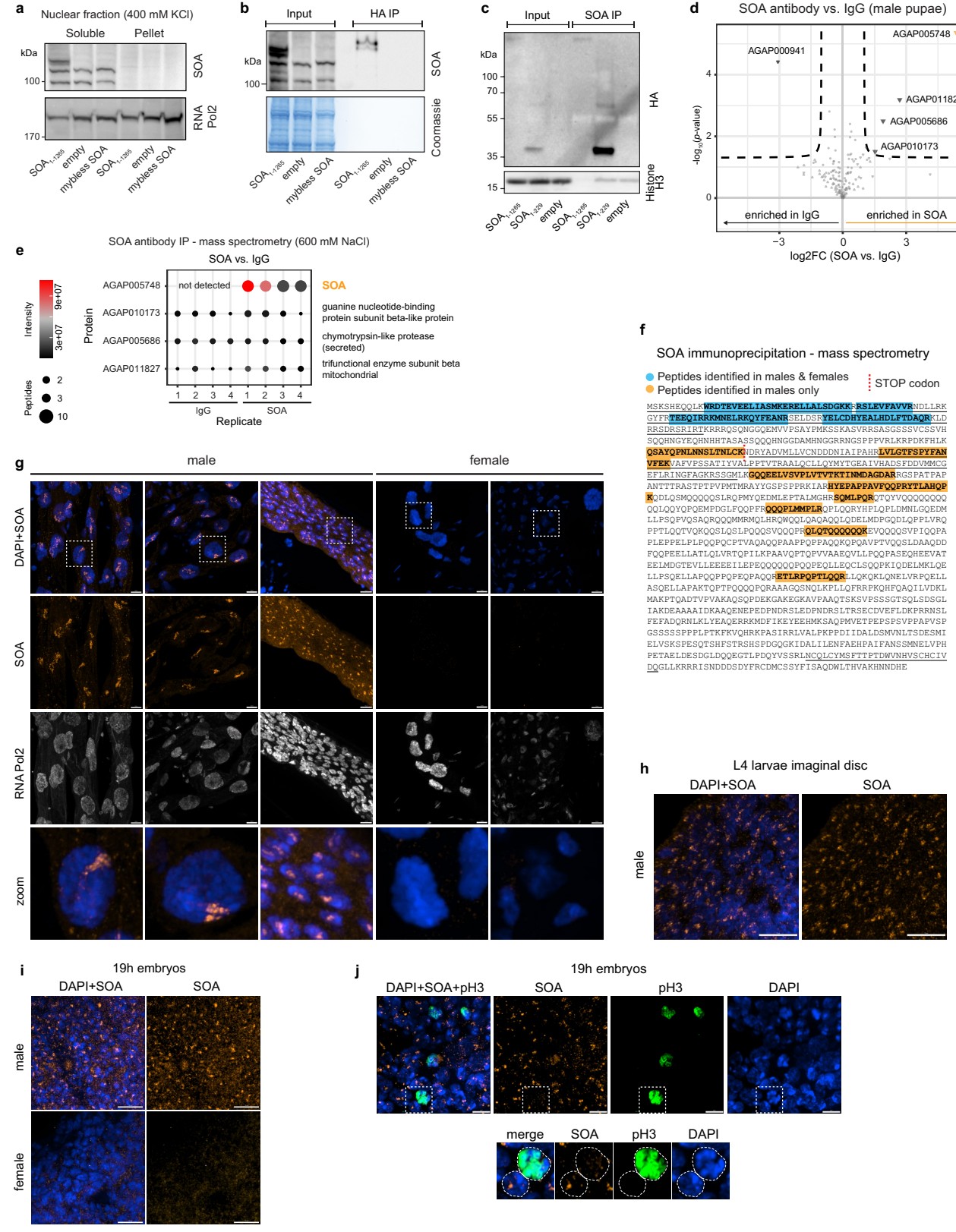

**Extended Data Fig. 5** | See next page for caption.

**Extended Data Fig. 5 | Validation of the SOA antibody and SOA staining in embryonic and larval tissues.** (**a**) Cropped immunoblot of ectopically expressed SOA and two negative controls. Nuclear soluble fraction extracted with 400 mM KCl was isolated from Ag55 cells expressing HA-tagged male SOA$_{1-1265}$, empty vector control, or mybless SOA$_{112-1265}$. Mybless SOA lacks the epitope (amino acids 1-112) used for immunization. RNA Polymerase 2 serves as a loading control. The experiment was repeated twice with similar results. (**b**) Cropped immunoblot of HA antibody immunoprecipitation (IP) with samples prepared as in (**a**). The SOA antibody was used for detecting the proteins immunoprecipitated by HA antibody, Coomassie serves as a loading and negative-IP control. The experiment was performed once. (**c**) Cropped immunoblot of SOA antibody IP with corresponding input samples. Chromatin extracted from Ag55 cells expressing HA-tagged male SOA$_{1-1265}$, female SOA$_{1-229}$ or empty control were used. The HA antibody was used for detecting the proteins immunoprecipitated by SOA antibody, H3 antibody serves as a loading and negative-IP control. The experiment was performed once. (**d**) SOA IP-mass spectrometry experiment represented as a volcano plot, with log2 fold change (log2FC) between SOA and IgG on the $x$-axis and log$10$ ($p$-value) on the $y$-axis. SOA (orange) and the 4 contaminant proteins (black) are highlighted in triangles, the remaining background noise proteins are shown in grey. IP was performed on nuclear extracts from male pupae using the SOA antibody ($n$ = 4 biologically independent experiments) or IgG control ($n$ = 4 biologically independent experiments). Raw data in Supplementary Table 1. (**e**) as in (**d**) Bubble plot representing the results of the SOA antibody IP-mass spectrometry experiment. The 4 significant proteins enriched in SOA versus IgG are shown in the plot. The color of the bubbles represents the measured intensity, and their size the number of unique detected peptides. SOA was the only protein not detected in IgG, while the other 3 were measured in both IPs. (**f**) Mass spectrometry was conducted on immunoprecipitated SOA from nuclear extracts of female and male pupae ($n$ = 5 biologically independent experiments each). The panel shows the amino acid sequence of SOA, the peptides identified in male and female samples (blue shades) or in males only (orange shades). The position of the STOP codon is shown in red, the underlined amino acids correspond to the three structured domains. Raw data in Supplementary Table 1. Note that because SOA proteins were enriched via IP, this experiment cannot directly inform on the relative abundance of SOA protein isoforms in the sexes. Considering the mRNA quantification by qPCR (Extended Data Fig. 4c), SOA$_{1-1265}$ and SOA$_{1-229}$ proteins appear to be mutually exclusive in the two sexes and SOA$_{1-1265}$ in males is at least 3-6 fold more abundant than SOA$_{1-229}$ in females. (**g**) Representative pictures of SOA (orange) and RNA Polymerase 2 (grey) immunostaining conducted on male and female adult mosquito tissues with DAPI in blue. Pictures show Malpighian tubules or gut. The bottom shows a closeup (zoom) of the area highlighted with a white square. The pictures represent a 3D view of a z-stack, scale bar = 10 μm. This panel represents the complete panel of Fig. 1g, where a subset of the very same images (merged DAPI+SOA channels with close up) is presented. (**h**) Representative pictures of SOA immunostaining (orange) with DAPI (blue) conducted on imaginal discs from male L4 larvae. The pictures represent a 3D view of a z-stack. Scale bar = 10 μm. (**i**) Representative pictures of SOA immunostaining (orange) with DAPI (blue) conducted on male and female embryos at 19h after oviposition. The sexes were identified based on their clear differences in SOA staining. The pictures represent a 3D view of a z-stack. Scale bar = 10 μm. (**j**) as in (**i**) Representative pictures of SOA immunostaining (orange) with DAPI (blue) in a male embryo at 19h after oviposition. Mitotic cells were identified by a staining of phosphorylated Histone H3 (pH3, green). The bottom shows a closeup of the area in the white square highlighting two nuclei, where one undergoes mitosis, while the other one is in interphase. The SOA staining can only be detected in the latter. The pictures represent a 3D view of a z-stack. Scale bar = 10 μm.

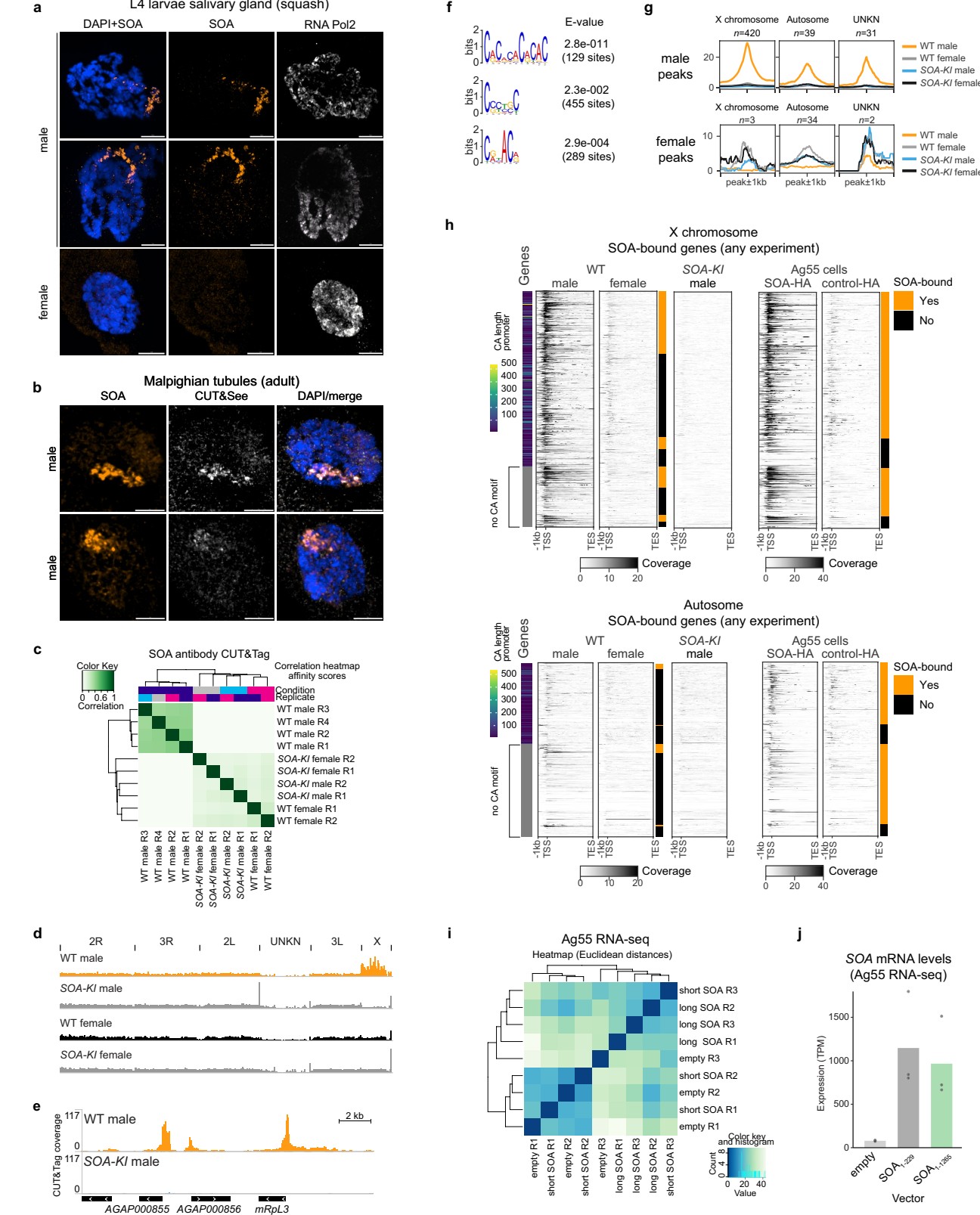

**Extended Data Fig. 6** | See next page for caption.

**Extended Data Fig. 6 | X chromosome binding and regulation by SOA.**
(**a**) Representative pictures of SOA (orange) and RNA Polymerase 2 (grey) immunostaining conducted on polytene squashes of salivary glands dissected from male and female L4 larvae. The pictures represent a 3D view of a z-stack. DAPI in blue, scale bar = 10 μm. (**b**) Pictures of CUT&See: SOA immunostaining (orange) combined with the visualization of SOA-targeted, pA-Tn5-mediated insertion of fluorescently labeled oligonucleotides (grey) conducted on wild-type male adult mosquito tissues. Pictures show Malpighian tubules, DAPI staining in blue. The pictures represent a 3D view of a z-stack. Scale bar = 10 μm. (**c**) Pearson correlation clustering of replicates based on affinity scores after peak calling of the SOA CUT&Tag data from pupae. The experiment was conducted with SOA antibody and IgG in wild-type (WT) male ($n$ = 4 biological replicates) and female ($n$ = 2), as well as homozygous *SOA* knock-in (*SOA-KI*) male ($n$ = 2) and female ($n$ = 2) pupae. The SOA antibody data was filtered using the IgG control and then subjected to clustering. (**d**) as in (**c**) Genome browser snapshot of the SOA CUT&Tag coverage on all chromosomal arms in the WT male and female as well as *SOA-KI* male and female genotypes. Duplicate reads were filtered out, replicates were merged for visualization. The enrichment is lost in the *SOA-KI* loss-of-function mutants. Note that the coverage in *SOA-KI* males is lower on the X due to copy number differences in comparison with XX females and autosomes. (**e**) as in (**c**) Genome browser snapshot of the SOA CUT&Tag coverage on a representative X-linked region in the WT and *SOA-KI* males. Replicates were merged for visualization. (**f**) MEME-ChIP motif analysis was conducted on all significant WT male CUT&Tag peaks. The position-weight matrix image of the three significant motifs ($E$-value ≤ 0.05) with obtained $E$-value from MEME is shown. (**g**) Metaplots showing the mean CUT&Tag enrichment at SOA peaks ±1 kb identified with DiffBind (FDR<0.05) in a comparison of WT males vs. females. (top:) peaks enriched in males (fold>0); (bottom:) peaks enriched in females (fold<0). Each of the colored lines corresponds to a different genotype. The male peaks are specific, as the enrichment is lost in the *SOA-KI* loss-of-function mutant males. The female peaks do not vanish in the mutants and can be considered background. (**h**) Heatmap comparing the SOA CUT&Tag data from pupa with SOA-HA CUT&Tag data from cells. The analysis is focused on genes that have a significant peak called in any of the CUT&Tag experiments (Supplementary Table 2). X chromosomal genes are shown in the top heatmaps, autosomal genes at the bottom. For plotting the enrichments, the transcription start site (TSS) was used as a reference point with 1 kb upstream and gene bodies downstream scaled to 5 kb. To order the genes, they were sorted first according to the presence of CA-motif (Extended Data Fig. 7e) in their promoter as matched by FIMO. Then they were sorted based on their peak status (Yes/No, orange bars) in the Ag55 cells (SOA-HA) experiment and lastly based on peak status (Yes/No, orange bars) in pupae. For the genes that exhibit a CA-motif, a length heatmap indicating the total number of nucleotides that match the motif was created (left of the heatmap). The peak status associated with a gene (orange bar = Yes, black bar = No) was assigned based on the DiffBind (FDR < 0.05) output in a particular experiment. Due to differences in signal-to-noise the scale is different between pupae and Ag55 cells, but maintained in the top and bottom heatmaps, to be able to compare relative binding strengths between X and autosomes. The replicates were merged for visualization. (**i**) Euclidean distance heatmap obtained by DESeq2 representing the similarity of the samples in RNA-seq performed in female Ag55 cells that ectopically express male (long) SOA$_{1-1265}$, female (short) SOA$_{1-229}$ or empty vector control. (**j**) Bar plot showing the mean *SOA* mRNA levels from RNA-seq in transcript per million (TPM) with points showing the values of $n$ = 3 biologically independent replicates.

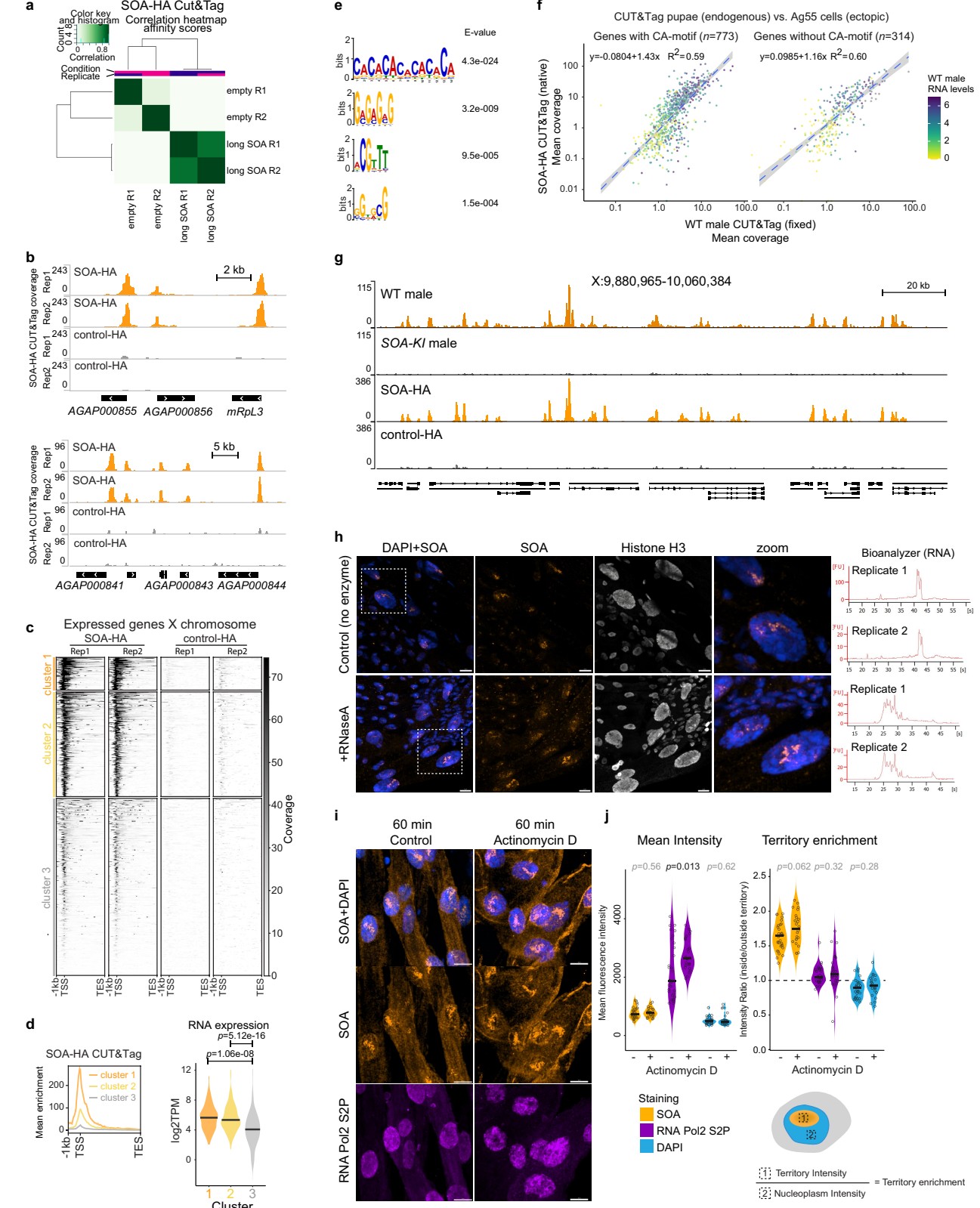

**Extended Data Fig. 7** | See next page for caption.

**Extended Data Fig. 7 | Consequences of SOA expression in female Ag55 cells.** (**a**) Pearson correlation of samples based on affinity scores after peak calling of the SOA CUT&Tag data from Ag55 cells infected with empty vector control or male (long) $SOA_{1-1265}$ baculovirus with the respective replicates. The experiment was conducted with HA antibody and IgG. The HA antibody data was filtered using the IgG control and then subjected to clustering. (**b**) Representative genome browser snapshots of the SOA-HA CUT&Tag coverage at two X-linked regions. (**c**) Heatmap showing the SOA-HA CUT&Tag enrichment with the TSS as reference point, 1 kb upstream and gene bodies scaled to 5 kb at expressed X-linked genes ($\geq$10 average read counts in RNA-seq). 3 random $k$-means clusters were generated revealing three different groups with varying SOA binding strength. (**d**) as in (**c**), the mean enrichment in each of the 3 $k$-means clusters is shown as a metaplot. Replicates were merged for visualization. The bottom panel shows a violin plot with center line representing the median RNA expression in $\log_2$TPM from RNA-seq of Ag55 cells for each of the 3 clusters. The Bonferroni-corrected $p$-values were obtained with a two-sided Wilcoxon rank-sum with pairwise comparisons between the clusters. (**e**) MEME-ChIP motif analysis was conducted on all significant SOA-HA CUT&Tag peaks. The position-weight matrix image of the four significant motifs ($E$-value $\leq$ 0.05) with obtained $E$-value is shown. (**f**) Scatter plot showing the correlation between the mean CUT&Tag coverage in male pupae ($x$-axis) and mean CUT&Tag coverage of SOA-HA in in Ag55 cells. Each dot represents an expressed ($\geq$10 average read counts) X chromosomal gene with the RNA levels $\log_2$(TPM+1) in WT males represented in color. The genes were further split based on the presence of a CA-motif in their promoters as assessed by a match in the FIMO search. The equation and $R^2$ value (coefficient of determination) of the fitted trend line was obtained by linear regression in R. (**g**) Representative genome browser snapshots of the SOA CUT&Tag data from pupae (WT and SOA-KI genotypes) and SOA-HA with empty vector control CUT&Tag data from Ag55 cells. The replicates were merged for visualization. (**h**) (left:) Representative pictures of SOA (orange) and H3 (grey) immunostaining conducted on male adult mosquito tissues after a 10 min treatment with buffer (control, top) or RNAseA (bottom). The pictures show Malpighian tubules. The right panel shows a closeup (zoom) of the area highlighted with a white square. The pictures represent a 3D view of a z-stack, DAPI in blue. Scale bar = 10 $\mu$m. (right:) Agilent bioanalyzer traces of RNA isolated from midguts ($n$ = 2 biological replicates) that were undergoing the same treatment as the immunostaining. The control treatment samples show the characteristic doublet for insect rRNA, while in the RNaseA-treated sample degradation into smaller fragments can be observed. (**i**) Representative pictures of SOA immunostaining conducted on male adult mosquito tissues after treatment for 60 min with Control (top) or Actinomycin D (bottom). The treatment was conducted in L15 tissue culture medium followed by fixation. The pictures show Malpighian tubules with SOA in orange, Ser2-phosphorylated RNA Pol2 (RNA Pol2 S2P) in pink and DAPI in blue. The pictures represent a 3D view of a z-stack that was visualized with Imaris software. Scale bar = 10 $\mu$m. (**j**) Quantification of the staining in (**i**). The violin plots show the mean fluorescence intensity or territory enrichment. The territory was calculated by determining the ratio of the mean intensity in equally sized squares placed inside the territory and outside of the territory as visualized in the illustration on the right. The center line represents the median. SOA is represented in orange, RNA Pol2 S2P in pink and DAPI in blue. $n$ = 30 nuclei were quantified for the control and $n$ = 26 nuclei for the Actinomycin D treatment. $p$-values: two-sided Wilcoxon rank-sum for a comparison between control and Actinomycin D in each staining.

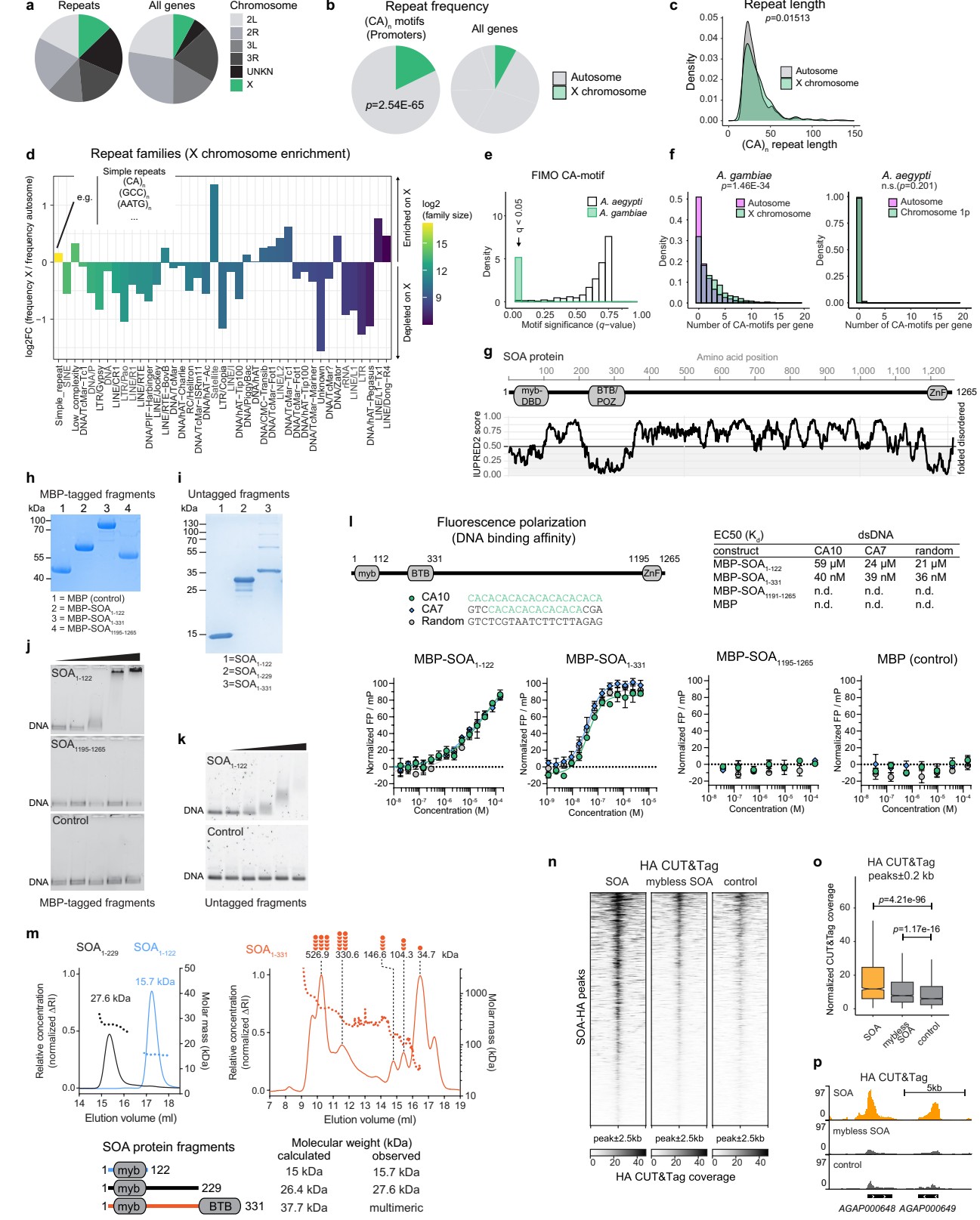

**Extended Data Fig. 8** | See next page for caption.

**Extended Data Fig. 8 | Characterization of SOA X chromosome recruitment mechanism.** (**a**) Pie chart indicating the number of repeats obtained by RepeatMasker on the X chromosome and other chromosomal locations (left) in comparison with the size of the respective regions in the genome (right). (**b**) Pie chart representing the number of (CA)n repeats localized at X-linked versus autosomal promoter region. The coordinates of the $(CA)_n$ simple repeats obtained from RepeatMasker were allocated to different feature classes (i.e. Promoter, intergenic, etc.) using the annotatePeak function of ChIPseeker and the AgamP4.8.gtf annotation. *p*-value: one-sided Fisher's test for overrepresentation of X-linked genes containing $(CA)_n$ compared with the chromosomal localization of all *Anopheles* genes on X and autosomes, respectively. (**c**) as in (**b**) density distribution of the repeat length (difference between start and end coordinates) of the $(CA)_n$ motifs located at promoters on X and autosomes, respectively. *p*-value: two-sided Wilcoxon rank-sum test comparing X and autosomes. (**d**) The fraction of a given repeat class on the X chromosome was compared with the fraction of the same repeat class on autosomes. The log2 ratio fraction X/fraction A was obtained and shown as a barplot for the repeat classes, where the color indicates the family size, i.e. log2 overall number of the given repeat classes. Simple repeats (illustrated below the barplot), low complexity repeats, LINE/RTE−BovB and satellite are the top 4 (by family size) repeat classes enriched on the X. (**e**) Histogram showing the results of a 'Find Individual Motif Occurrences' (FIMO) search[54], in which the promoter regions of *A. aegypti* (control, no sex chromosomes) and *A. gambiae* were scanned for occurrences of the top scoring CA-motif (Extended Data Fig. 6f). The histogram shows the *q*-value of the obtained hits, which indicates the significance of the discovered loci to match the CA-motif used in the search. (**f**) FIMO motif searches as in (**e**). The histogram shows the number of motif hits found per gene promoter. The CA-motif tends to form clusters at X-linked promoters of *A. gambiae*, where often more than one motif per gene is present. Chromosome 1p in *A. aegypti* is homologous to the X of *A. gambiae*[19], but is not a differentiated sex chromosome. *p*-values: one-sided Fisher's exact test for overrepresentation of genes containing a FIMO-match on (left:) the X (*A. gambiae*) or (right:) chromosome 1p (*A. aegypti*). (**g**) Schematic illustration of the predicted domain architecture of SOA obtained on VectorBase from Interpro and intrinsically disordered scores from IUPRED2. (**h**) Coomassie stained gel of purified recombinant MBP and MBP-tagged SOA fragments. The purified fragments were used in the EMSA assay in (**j**) and fluorescence polarization assays (**l**). The SDS-PAGE was performed once to confirm the quality of the purified fragments. (**i**) Coomassie stained gel of purified recombinant N-terminal fragments of SOA without affinity tags. The purified fragments were used in the EMSA assay in (**k**) and Size exclusion multiangle light scattering (SEC-MALS, **m**). The SDS-PAGE was performed once to confirm the quality of the purified fragments. (**j**) EMSA assay of recombinant MBP-tagged myb-DNA binding domain ($SOA_{1-112}$), ZnF domain ($SOA_{1195-1265}$) and negative control protein (MBP). The protein amount in each lane was increased from 0 pmol (probe only) to 125-fold molar excess (12.5 pmol) over the probe (0.1 pmol). The probe was an equimolar mix of 300 bp-long X chromosomal promoter DNA sequences (sequences in Supplementary Table 1). After electrophoresis, the gel was stained with SYBRSafe. The experiment was performed twice with similar results. (**k**) EMSA assay of recombinant SOA myb-DNA binding domain. The protein amount in each lane was increased from 0 pmol (probe only) to 125-fold molar excess (12.5 pmol) over the probe (0.1 pmol). 147 bp 601-DNA sequence (Supplementary Table 1) was stained with SYBRSafe. GST protein was used as a negative control (bottom gel). The experiment was repeated three times with similar results. (**l**) Scheme and results of fluorescence polarization (FP) assay using Cy5-labeled DNA probes containing CA-motifs (CA10 - green circle, CA7 - blue diamond) or a random sequence (grey circle) that were incubated with various concentrations of MBP-$SOA_{1-122}$, MBP-$SOA_{1-331}$, or MBP-$SOA_{1195-1265}$. The mean relative FP values from three independent experiments including error bars indicating the standard deviation are shown over the indicated concentrations. Binding constants ($K_d$ values) were determined by fitting a Michaelis−Menten non-linear regression to the relative FP values. The respective binding constants are given in the table (also see Supplementary Table 1). (**m**) (top:) Normalized differential refractive index (solid lines) and molar mass (dotted lines) from Size exclusion multiangle light scattering (SEC-MALS) for $SOA_{1-122}$, $SOA_{1-229}$ (left panel) and $SOA_{1-331}$ (right panel) with the elution volume by SEC displayed on the *x*-axis. The loading concentrations of the samples were 200 μM for the two short fragments (left) and 11 μM for the longest fragment (red) (bottom:) Schematic Illustration of the 3 purified SOA fragments analyzed by SEC-MALS. The calculated monomeric weight based on the protein sequence, as well as the observed weight-averaged molar mass are indicated. (**n**) Heatmap showing the normalized CUT&Tag coverage on all significant peaks (±2.5 kb) called in the $SOA_{1-1265}$ in comparison with empty vector control expressing Ag55 cells (Fig. 3e). The enrichment at these sites is shown for SOA-HA, SOA-HA lacking myb-domain (mybless) or empty vector control (*n* = 2 biologically independent replicates in all groups, merged for visualization). CUT&Tag was performed with HA antibody. (**o**) Box plot of the mean CUT&Tag enrichment of each peak ± 0.2 kb (*n* = 1787), as in (**n**) calculated with multiBigWigSummary with center line representing the median enrichment, box bottom, and top edges represent interquartile ranges (IQR, 0.25th to 0.75th quartile [Q1-Q3]), whiskers represent Q1 − 1.5*IQR (bottom), Q3 + 1.5*IQR (top). The Bonferroni-corrected *p*-values were obtained with a two-sided Wilcoxon rank-sum with pairwise comparisons between the groups. (**p**) Representative genome browser snapshots of the CUT&Tag data for SOA-HA, SOA-HA lacking myb-domain (mybless) or empty vector control (*n* = 2 biologically independent replicates in all groups, merged for visualization).

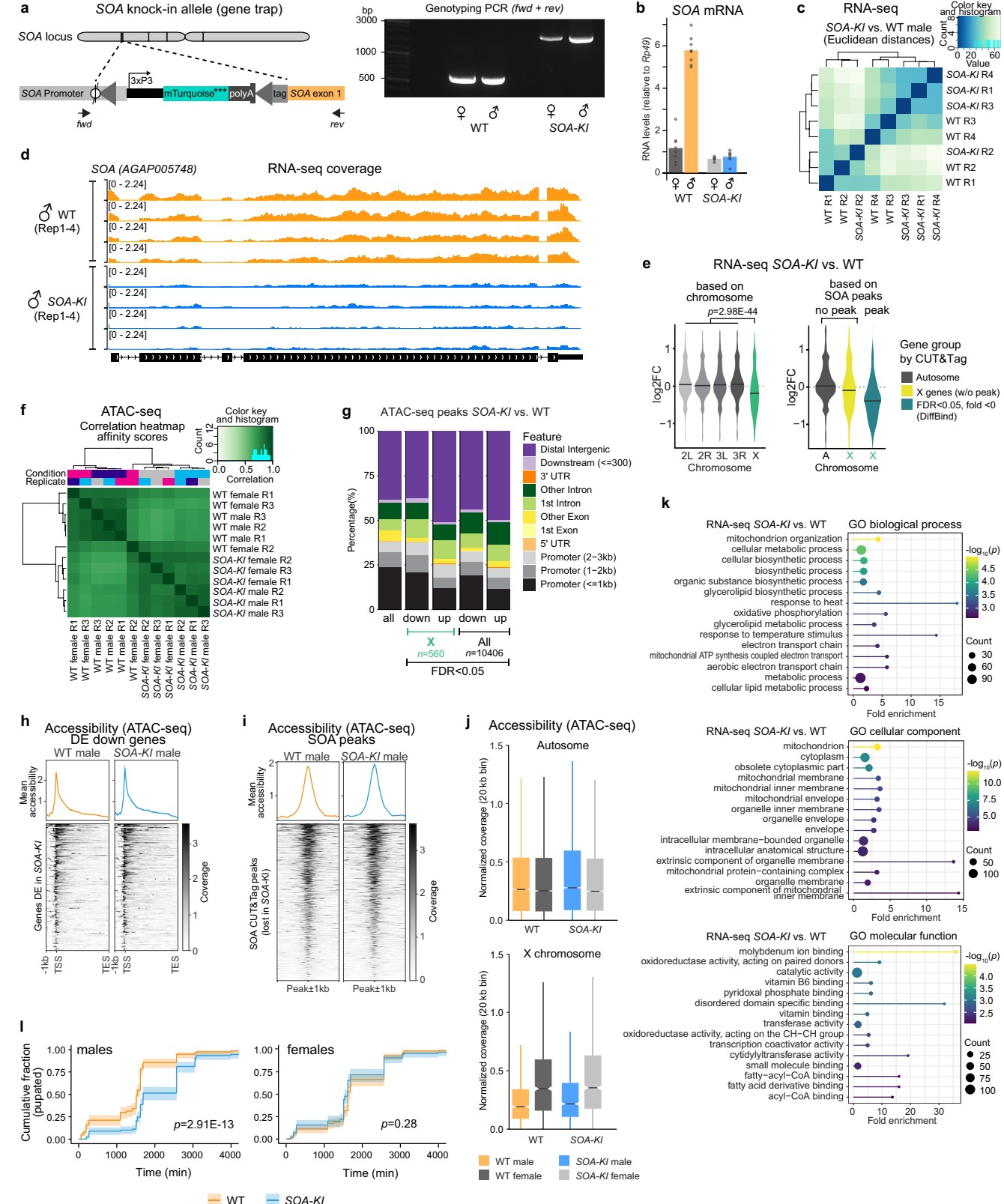

**Extended Data Fig. 9 |** See next page for caption.

**Extended Data Fig. 9 | Characterization of the *SOA-KI* mosquito transgenic line.** (**a**) Schematic illustration of the *SOA* knock-in (*SOA-KI*) allele, in which the first exon of *SOA* and the coding sequence are interrupted by the eye and nervous system-specific 3xP3 promoter, mTurquoise coding sequence, a poly(A) site and a epitope tag. Two inverted loxP sites are illustrated by triangles, which were intended for marker cassette removal. The PhiC31 *att*P landing site is indicated with a circle. The position of the PCR screening primers is shown with arrows. The right panel shows a representative agarose gel of PCR products obtained in WT male, female or *SOA-KI* male, female homozygous transgenic line. (**b**) Bar plot showing *SOA* mRNA levels relative to *Rp49* in WT and homozygous *SOA-KI* pupae measured by RT-qPCR. Height of the bar plot is the mean of $n = 8$ biological replicates with overlaid individual data points. (**c**) Euclidean distance heatmap obtained by DESeq2 representing the similarity of the samples in RNA-seq conducted from WT and homozygous *SOA-KI* male pupae. (**d**) RNA-seq as in (**c**) Representative genome browser snapshot of the *SOA* locus with RNA-seq coverage for each of the $n = 4$ biological replicates. (**e**) RNA-seq as in (**c**) Violin plot with center line representing the median show the DESeq2-obtained log2FC in WT compared to homozygous *SOA-KI* mutant. Each gene with an average read count (baseMean) > 0 was taken into account, irrespective of whether it was scored as differentially expressed or not. (left:) The genes were grouped by chromosomal location. The *p*-value was obtained with a two-sided Wilcoxon rank-sum test comparing X (green) with all autosomes (grey). (right:) The genes were grouped by presence of a peak in CUT&Tag: All autosomal genes (grey), X-linked genes without peaks (yellow) and X-linked genes with a peak (blue) as scored by DiffBind (FDR <0.05, fold<0, *SOA-KI* versus WT) (Supplementary Table 2). Median log2FC values for each group are available in Supplementary Table 3. The Bonferroni-corrected *p*-values were obtained with a two-sided Wilcoxon rank-sum test comparing: autosomal versus X-linked genes without a SOA peak $p = 1.32E\text{-}10$; autosomal versus X-linked genes with a SOA peak $p = 1.02E\text{-}53$; X-linked genes without versus with a SOA peak $p = 1.03E\text{-}15$. (**f**) Heatmap showing the sample relatedness of the ATAC-seq replicates conducted from male WT and homozygous *SOA-KI* pupae based on Pearson correlation coefficient. (**g**) Barplot showing the % of ATAC-seq peaks in each of the genomic locations identified by ChIPseeker. (**h**) Heatmap of ATAC-seq coverage at each genomic region containing a SOA CUT&Tag peak scored as DiffBind in homozygous *SOA-KI* compared to WT males. The center of the peak is used as a reference point. The mean coverage is shown at the top of the heatmap. ATAC-seq replicates were merged for visualization by calculating the mean of normalized bigwigs using WiggleTools. (**i**) The accessibility of each gene with significantly decreased expression in homozygous *SOA-KI* (Fig. 4c) is visualized as a heatmap with the normalized ATAC-coverage using the TSS as reference point, 1kb upstream of the TSS and the scaled gene body (downstream of the TSS). The mean coverage is shown at the top of the heatmap. ATAC-seq replicates were merged for visualization by calculating the mean of normalized bigwigs using WiggleTools. (**j**) Box plot showing the normalized ATAC-seq coverage per 20 kb bin in the indicated chromosomal locations and genotypes. The line that divides the box into two parts represents the median, box bottom, and top edges represent interquartile ranges (IQR, 0.25th to 0.75th quartile [Q1-Q3]), whiskers represent Q1 − 1.5*IQR (bottom), Q3 + 1.5*IQR (top). The experiment was conducted in WT and homozygous *SOA-KI* pupae of both sexes ($n = 4$ biological replicates each). Note that accessibility of autosomes is equal between sexes and genotypes. Due to copy number differences, the expected 2-fold difference between males (XY) and females (XX) is observed on the X chromosome. Since this ratio is not substantially different from 2, we conclude that (regardless of chromosomal location and genotype) accessibility between males and females is highly similar. (**k**) Gene Ontology (Biological Process, top; Cellular Component, middletop; Molecular Function, bottom) analysis of the differentially expressed genes from RNA-seq from WT and homozygous *SOA-KI* male pupae. The lollipop plot shows the fold enrichment of genes in the various classes, with the point size indicative of the gene count and color indicative of the *p*-value. The analyses were conducted with the GO-Term tool on VectorBase. (**l**) 100 neonate larvae of each of the 4 scored genotypes (WT males, WT females, homozygous *SOA-KI* males, homozygous *SOA-KI* females) were seeded in the same culture for development through larval stages. The developmental timing of each of the 4 genotypes was scored by counting the appearance of pupae, which is represented as a cumulative distribution. The $t = 0$ of the x-axis represent the time when the first pupa appeared in the culture. The line represents the average of 4 replicates with shaded 95% confidence interval and *p*-value obtained by a log-rank test for stratified data (Mantel-Haenszel test). A second independent experiment with an additional 4 replicate cultures is presented in Fig. 4f.

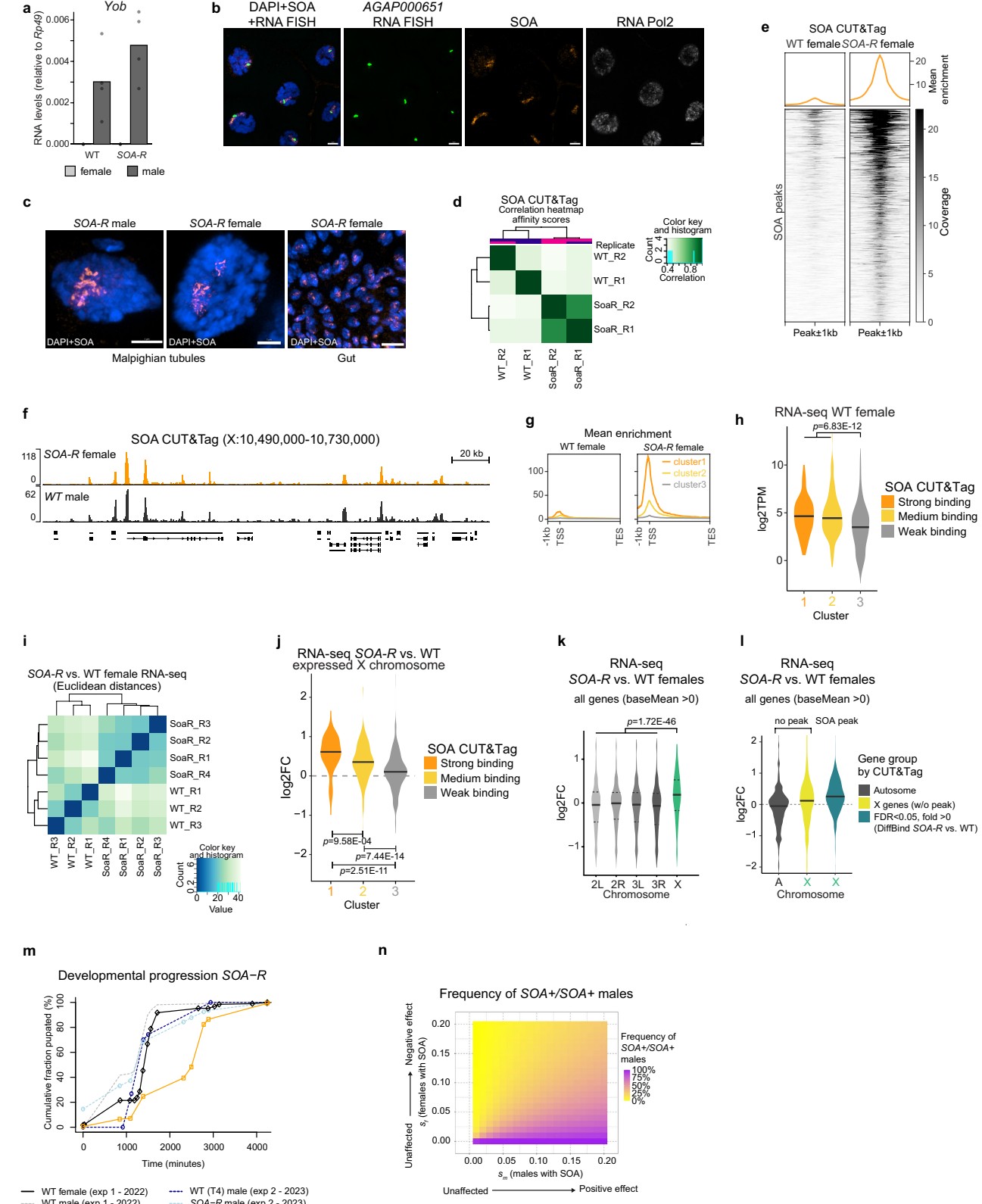

**Extended Data Fig. 10 | See next page for caption.**

**Extended Data Fig. 10 | Characterization of *SOA-R* mosquitoes and computational modelling for the spread of SOA.** (**a**) Bar plot showing *Yob* mRNA levels relative to *Rp49* in WT and homozygous *SOA-R* pupae measured by RT-qPCR. *Yob* mRNA levels confirm the sex of the pupae used in Fig. 5b. The height of the bar plot is the mean of $n = 4$ biological replicates with overlaid individual data points. (**b**) Representative pictures of SOA immunostaining (orange), RNA Polymerase 2 immunostaining (grey) and co-RNA FISH (green) of a X-linked transcription site (*AGAP000651*) on salivary gland nuclei of a homozygous male *SOA-R* L4 larva. The RNA-FISH probes were designed against the introns of the *AGAP000651* gene. DAPI is shown in blue, scale bar = 10 μm. (**c**) Representative pictures of SOA immunostaining conducted on homozygous *SOA-R* male and female adult mosquito tissues. Pictures show nuclei of Malpighian tubules (left, scale bar = 5 μm) or gut (right, scale bar = 10 μm) with SOA in orange and DAPI in blue. The pictures represent a 3D view of a z-stack. Further images in Fig. 5c. (**d**) Pearson correlation clustering of SOA CUT&Tag samples based on affinity scores after peak calling. The experiment was conducted with SOA antibody and IgG in WT and homozygous *SOA-R* female pupae. The SOA antibody data was filtered using the IgG control and then subjected to clustering. (**e**) Heatmap showing the normalized CUT&Tag coverage on all significant peaks (FDR<0.05, fold-change >0) in *SOA-R* in comparison with WT female pupae. The mean enrichment is shown as a metaplot on top ($n = 2$ biological replicates, merged for visualization). (**f**) Genome browser snapshot of the SOA CUT&Tag enrichment obtained in *SOA-R* females in comparison with WT males on a representative region of the X-chromosome. Duplicate reads were filtered out and the replicates were merged for visualization. (**g**) CUT&Tag as in (**e**) Metaplot showing the mean CUT&Tag enrichment on expressed X-linked genes (≥10 average read counts), which were further grouped by unsupervised k-means clustering in 3 groups with strong, medium and weak SOA binding strength. The coverage was calculated using the TSS as a reference point with 1 kb upstream and the gene bodies downstream scaled to 5 kb. The replicates were merged for visualization. (**h**) Violin plot with center line representing the median RNA expression in log2 TPM (transcripts per million) from RNA-seq of WT females for each of the 3 clusters (based on binding in *SOA-R*, see (**g**)). *p*-value: two-sided Wilcoxon rank-sum comparing combined clusters 1 and 2 versus cluster 3. (**i**) Euclidean distance heatmap obtained by DESeq2 representing the similarity of the samples in RNA-seq conducted from WT ($n = 3$ biological replicates) and homozygous *SOA-R* ($n = 4$ biological replicates) female pupae. (**j**) RNA-seq as in (**i**) Violin plots showing the log2FC on expressed X-linked genes (≥10 average read counts), which were further grouped by unsupervised *k*-means clustering in 3 groups with strong, medium and weak SOA binding strength, see (**g**). The center line represents the median log2FC, which equals 0.613, 0.355, and 0.117 (strong, intermediate and weak binding) and corresponds to fold changes of 1.529, 1.279, and 1.084, respectively. (**k**) RNA-seq as in (**j**) but plotting the log2FC for all genes according to the chromosomal location in WT compared to homozygous *SOA-R* pupae as a violin plot. Each gene with an average read count (baseMean) > 0 was taken into account, irrespective of whether it was scored as DE or not. The Bonferroni-corrected *p*-value was obtained with a two-sided Wilcoxon rank-sum test comparing X with all autosomes. The center line represents the median (also see Supplementary Table 3). (**l**) RNA-seq as in (**i**) but plotting the log2FC distribution of autosomal (grey) and X-linked genes. The X-linked genes were split into two groups based on SOA binding in CUT&Tag (Supplementary Table 2). The yellow violin plot shows X chromosomal genes without SOA peaks, the blue violin plot shows peaks that were scored as differentially bound by DiffBind (*SOA-R* versus WT females, FDR<0.05, fold>0). Median log2FC values for each group are available in Supplementary Table 3. The Bonferroni-corrected *p*-values obtained with a two-sided Wilcoxon rank-sum test comparing all groups between each other are: autosomal versus X-linked genes without SOA peak $p = 6.57E\text{-}12$; autosomal versus X-linked genes with SOA peak $p = 1.45E\text{-}44$; X-linked genes without versus with SOA peak $p = 5.48E\text{-}06$. (**m**) A single culture of *SOA-R* (males + females) was conducted in parallel to WT males (T4 strain), cultured separately. For both, the developmental timing of each of the 3 genotypes was scored by counting the appearance of pupae. Pupa appearance is represented as a cumulative distribution with dots representing a given time-point when pupa numbers were scored. The $t = 0$ on the *x*-axis represents the time when the first pupa appeared in the culture. The data represents one experiment. For comparison, the mean WT male and female pupation timings scored in Fig. 4f (exp 1-2022) are plotted in the panel. A separate experiment with additional $n = 3$ independent replicate cultures for *SOA-R* grown together with WT is presented in Fig. 5g. (**n**) Checkerboard plot indicating the relative frequency of *SOA*+/*SOA*+ males (colour-coded) after 10,000 generations of selection depending on the selection coefficients $s_m$ in males (*x*-axis) and $s_f$ in females (*y*-axis). Fitness is normalized to 1 in *SOA*–/*SOA*– males and females. Moreover, we assume that *SOA*+ is dominant over *SOA*– in males and recessive in females. Hence, the fitness of *SOA*+ bearing males is $1 + s_m$, while the fitness of *SOA*+/*SOA*+ females is 1- $s_f$. Even if selection against *SOA*+ in females is stronger than selection in favour of *SOA*+ in males, *SOA*+ is, for most parameter combinations, maintained in the population at considerable frequencies.

# Reporting Summary

## Statistics

For all statistical analyses, confirm that the following items are present in the figure legend, table legend, main text, or Methods section.

| n/a | Confirmed | |
|---|---|---|
| ☐ | ☒ | The exact sample size (*n*) for each experimental group/condition, given as a discrete number and unit of measurement |
| ☐ | ☒ | A statement on whether measurements were taken from distinct samples or whether the same sample was measured repeatedly |
| ☐ | ☒ | The statistical test(s) used AND whether they are one- or two-sided<br>*Only common tests should be described solely by name; describe more complex techniques in the Methods section.* |
| ☒ | ☐ | A description of all covariates tested |
| ☐ | ☒ | A description of any assumptions or corrections, such as tests of normality and adjustment for multiple comparisons |
| ☐ | ☒ | A full description of the statistical parameters including central tendency (e.g. means) or other basic estimates (e.g. regression coefficient) AND variation (e.g. standard deviation) or associated estimates of uncertainty (e.g. confidence intervals) |
| ☐ | ☒ | For null hypothesis testing, the test statistic (e.g. *F*, *t*, *r*) with confidence intervals, effect sizes, degrees of freedom and *P* value noted<br>*Give P values as exact values whenever suitable.* |
| ☒ | ☐ | For Bayesian analysis, information on the choice of priors and Markov chain Monte Carlo settings |
| ☒ | ☐ | For hierarchical and complex designs, identification of the appropriate level for tests and full reporting of outcomes |
| ☐ | ☒ | Estimates of effect sizes (e.g. Cohen's *d*, Pearson's *r*), indicating how they were calculated |

*Our web collection on statistics for biologists contains articles on many of the points above.*

## Software and code

Policy information about availability of computer code

| Data collection | No software was used for data collection. |
|---|---|

| Data analysis | RNA-SEQ: cudadapt (1.18) STAR (v. 2.7.3a) deepTools (v3.1.0). subread (1.6.5) DESeq2 (1.26.0)<br>CUT&Tag: cutadapt (4.0) bowtie2 (2.4.5) macs2 (2.1.2) GreyListChIP (1.22.0) DiffBind (3.0) deepTools (3.5.1) MEME-ChIP (MEME v. 5.4.1)<br>FIMO (Version 5.4.1)<br>ATAC-seq: cutadapt (4.0) bowtie2 (2.4.5) macs2 (2.1.2) DiffBind (3.6.1) deepTools (v3.5.1) WiggleTools (1.2.8) multiBigwigSummary (Galaxy Version 3.5.1.0.0. )<br>Gviz 1.34.1<br>IGV 2.8.9<br>Mass Spectrometry: Andromeda search engine of the MaxQuant software suite v1.6.5.0<br>Microscopy: VisiView 5.0 software, Imaris (v. 9.9.1)<br>Western Blot & Gels: Image Lab Software Version 6.1<br>SEC-MALS: ASTRA 8 software (Wyatt Technology)<br>Fluorescence Polarization: GraphPad Prism 8<br>Sequence analyses, alignments and evolution: Jalview, Version: 2.11.2.3, MEGA software (version 7.0), Clustal Omega on https://www.ebi.ac.uk/Tools/msa/clustalo/ (no version stated) , Prediction of Intrinsically Unstructured Proteins on https://iupred2a.elte.hu/ (IUPred2A version), DNA binding site predictor on http://zf.princeton.edu/ (no version stated), Nuclear Localization Signal prediction on https://nls-mapper.iab.keio.ac.jp/cgi-bin/NLS_Mapper_form.cgi (no version stated, Last Update: 2012/11/7); Alignment visualization with ESPript available online on https://espript.ibcp.fr/ESPript/ESPript/<br>Statistics & Plotting: R StudioVersion 1.4.1717 with R version 4.1.1<br>Figures: Adobe Illustrator & Photoshop 2021 |

For manuscripts utilizing custom algorithms or software that are central to the research but not yet described in published literature, software must be made available to editors and reviewers. We strongly encourage code deposition in a community repository (e.g. GitHub). See the Nature Portfolio guidelines for submitting code & software for further information.

# Data

Policy information about availability of data

All manuscripts must include a data availability statement. This statement should provide the following information, where applicable:
- Accession codes, unique identifiers, or web links for publicly available datasets
- A description of any restrictions on data availability
- For clinical datasets or third party data, please ensure that the statement adheres to our policy

No restrictions apply and all data is available in the manuscript or the supplementary materials. RNA-seq, CUT&Tag and ATAC-seq data have been deposited to GEO (GSE210624, GSE210630). Mass Spectrometry has been deposited to ProteomeXchange via the PRIDE database (project ID PXD042353). DNA and protein sequences are publicly available and were retrieved from VectorBase (www.vectorbase.org). The ensembl AgamP4 genome using the Ensembl AgamP4 annotation (release 48) was also retrieved via vectorbase.org. Metazoan Upstream Sequences for Anopheles gambiae (AgamP4.34_2019-03-11) or Aedes aegypti (AaegL3.34_2019-03-11) databases used in FIMO are publicly available / selectable under https://meme-suite.org/meme/tools/fimo. RNA-seq data from Papa et al. is publicly available in the Sequence Read Archive (SRA; http://www.ncbi.nlm.nih.gov/sra) under accession number SRP083856.

# Human research participants

Policy information about studies involving human research participants and Sex and Gender in Research.

| Reporting on sex and gender | This study does not involve Human Research Participants. |
| Population characteristics | This study does not involve Human Research Participants. |
| Recruitment | This study does not involve Human Research Participants. |
| Ethics oversight | This study does not involve Human Research Participants. |

Note that full information on the approval of the study protocol must also be provided in the manuscript.

# Field-specific reporting

Please select the one below that is the best fit for your research. If you are not sure, read the appropriate sections before making your selection.

☒ Life sciences    ☐ Behavioural & social sciences    ☐ Ecological, evolutionary & environmental sciences

For a reference copy of the document with all sections, see nature.com/documents/nr-reporting-summary-flat.pdf

# Life sciences study design

All studies must disclose on these points even when the disclosure is negative.

| Sample size | Sample size, number of replicates, error bars and statistical tests were chosen based on accepted practices in the field, which are stated in Figure legends and methods. Generally experiments were performed independently and reproduced with at least 3 biological replicates. For |

genome-wide datasets we followed the recommendations and practices of the ENCODE and modENCODE consortia (PMID: 22955991). Examples of field specific studies: PMID 32510132, 28457869, 29562179

| | |
|---|---|
| Data exclusions | No data was excluded. |
| Replication | Each experiment was repeated at least three times with similar results, unless otherwise noted. All attempts of replication were successful. Details are provided in Figure legends. |
| Randomization | Samples were allocated to study groups by genotype. No randomization was applied. |
| Blinding | Investigators were not blinded. Blinding was not relevent as of objective experimental readouts (molecular analysis, visible phenotype) allowing sample allocation. |

# Reporting for specific materials, systems and methods

We require information from authors about some types of materials, experimental systems and methods used in many studies. Here, indicate whether each material, system or method listed is relevant to your study. If you are not sure if a list item applies to your research, read the appropriate section before selecting a response.

## Materials & experimental systems

| n/a | Involved in the study |
|---|---|
| ☐ | ☒ Antibodies |
| ☐ | ☒ Eukaryotic cell lines |
| ☒ | ☐ Palaeontology and archaeology |
| ☐ | ☒ Animals and other organisms |
| ☒ | ☐ Clinical data |
| ☒ | ☐ Dual use research of concern |

## Methods

| n/a | Involved in the study |
|---|---|
| ☒ | ☐ ChIP-seq |
| ☒ | ☐ Flow cytometry |
| ☒ | ☐ MRI-based neuroimaging |

## Antibodies

| | |
|---|---|
| Antibodies used | Anti-SOA Antibody (Custom (Eurogentec), epitope-purified by the IMB PPCF,N.A. (Rabbit 87), #540887-22062021)<br>Anti-HA.11 Antibody (Biolegend, BLD-901502), clone 16B12<br>Anti-Histone H3 Antibody (Cell Signalling, 9715S)<br>Anti-Histone H3 (mAb) Active Motif 39763, clone MABI 0301<br>RNA pol II antibody (mAb) Active Motif 39097, clone 4H8<br>RNA pol II CTD phospho Ser2 antibody (mAb) Active Motif 61984, clone 3E10<br>phospho H3 (S10)Mouse IgG2b, κ Biolegend 650801, clone 11D8<br>IgG control Antibody (Abcam, ab37415)<br>Anti-αMs IgG Antibody (Abcam, ab6709)<br>Anti-Rb IgG Antibody (Sigma-Aldrich, SAB3700377)<br>Anti-Rb IgG coupled to AF555 Antibody (ThermoFisher, A21430)<br>Anti-Mouse IgG (H+L) Antibody (Jackson ImmunoResearch, JIM-715-035-150)<br>Anti-Rabbit IgG (H+L) Antibody (Jackson ImmunoResearch, JIM-711-035-152) |
| Validation | PRIMARY ANTIBODIES:<br><br>SOA antibody was generated in this study. The validations are presented in the paper:<br>In CUT&Tag (Fig. 2 and 4), SOA mutants (SOA-KI) and non-specific isotype antibody (IgG) controls validate the specific enrichments.<br>In Immunofluorescence (Fig. 1 and 4), lack of signal in mutants (SOA-KI) and females validate the specificity.<br>In Western blot (Extended Data Fig. 4e,), the specific immunoprecipitation of SOA, but not control - expressing cells, validate the specificity, which is also confirmed by mass spectrometry (Fig. 1). Detected bands run at the expected molecular weights.<br><br>Anti-HA.11 Antibody (Biolegend, BLD-901502) - this study: validated in the western blot and CUT&Tag by using a negative control condition (empty baculovirus). No western blot signal in the empty control; no CUT&Tag peaks in the empty control. Band with expected molecular weight observed in the HA-SOA1-229 expressing sample.<br>Cited in 424 peer-reviewed article as of 09.12.2022.<br>Monoclonal antibody against the YPYDVPDYA peptide. Search for this peptide in the A.gambiae proteome yielded no identical sequences.<br>Validated by the manufacturer for use in western blot.<br>https://www.biolegend.com/en-gb/products/purified-anti-ha-11-epitope-tag-antibody-11374<br>"Additional tested and reported applications of the 16B12 clone for the relevant formats include: western blot (WB), immunocytochemistry (ICC), immunoprecipitation (IP), and flow cytometry (FC)."<br><br>Anti-Histone H3 Antibody (Cell Signalling, 9715S) - in this study: detected at correct molecular weight (~17 kDa) in the nuclear fraction.<br>https://www.cellsignal.com/products/primary-antibodies/histone-h3-antibody/9715<br>Cited in 957 peer-reviewed article as of 12.09.2022 "For western blots, incubate membrane with diluted primary antibody in 5% w/v nonfat dry milk, 1X TBS, 0.1% Tween® 20 at 4°C with gentle shaking, overnight." |

It has been successfully used in in Drosophila (https://www.ncbi.nlm.nih.gov/pmc/articles/PMC4008575/) and H3 is highly conserved.

Anti-Histone H3 (mAb) Active Motif 39763 - in this study: detected in correct nuclear compartment in IF
RRID:AB_2650522. Clone:MABI 0301
Applications Validated by Active Motif:
ChIP-Seq: 4 µg per ChIP
ChIP: 5 - 10 µg per ChIP
ICC/IF: 1 µg/ml dilution
WB: 0.5 - 2 µg/ml dilution
25 publications using antibody on Active Motif Website since 2001

RNA pol II antibody (mAb) Active Motif 39097 - in this study: detected in correct localization (nucleus/chromatin) in IF and Western blot, correct size in Western blot.
RRID:AB_2732926. 12 citations
Applications Validated by Active Motif:
ChIP: 10 µl per ChIP
ChIP-Seq: 20 µl each
WB: 1:2,000 - 1:5,000 dilution
The following applications have been published using this antibody. Unless noted above, Active Motif may not have validated the antibody for use in these applications:CUT&TagC hIP-Seq, ChIP-qPCR ICC/IF WB
24 publications using antibody on Active Motif Website since 2001

RNA pol II CTD phospho Ser2 antibody (mAb) Active Motif 61984 - in this study: correct pattern in IF
RRID:AB_2687450. 12 citations
RRID:AB_2687450
Clone:3E10
Applications Validated by Active Motif:
WB*: 0.5 - 2 µg/ml
ChIP: 20 µg per ChIP
ChIP-Seq: 20 µg each
IF: 1:500 dilution

phospho H3 (S10)Mouse IgG2b, κ Biolegend 650801 - in this study: correct pattern in IF - specific presence in mitotic cells
RRIDAB_10896911 (BioLegend Cat. No. 650801)
AB_10900065 (BioLegend Cat. No. 650802)
Antigen References
1. Choi HS, et al. 2005. J. Biol. Chem. 280:13545.
2. Goto H, et al. 2002. Genes Cells 7:11.
3. Garcia BA, et al. 2005. Biochemistry 44:13202.
4. Hans F, et al. 2001. Oncogene 20:3021.
Product Citations
Friedman J, et al. 2018. J Immunother Cancer. 6:59. PubMed
Han G, et al. 2018. Nat Protoc. 2.014583333. PubMed
IgG control Antibody (Abcam, ab37415)
https://www.abcam.com/rabbit-igg-monoclonal-epr25a-isotype-control-ab172730.html#lb
Shows only background signal in Cut&Tag (this study)
Validated for a similar application (CUT&RUN): "ChIC/CUT&RUN-seq Use 0.5-2µg for 105 cells."
Cited in 302 peer-reviewed article as of 12.09.2022

SECONDARY ANTIBODIES:

Anti-αMs IgG Antibody (Abcam, ab6709)
https://www.abcam.com/rabbit-mouse-igg-hl-ab6709.html
Cited in 33 peer-reviewed article as of 12.09.2022
Affinity purified: "this product was prepared from monospecific antiserum by immunoaffinity chromatography using Mouse IgG coupled to agarose beads"
Successfully used in peer-reviewed studies for CUT&Tag:
"CUT&Tag was performed with CUT&Tag-IT Assay Kit (53160, ACTIVE MOTIF) in 1.5×106 FaDu cells using anti-SF3B2 (sc-514976, Santa Cruz, 5 µL, 1:20 dilution) and anti-H3 (ab1791, Abcam, 1 µL, 1:100 dilution) antibodies. Rabbit anti-mouse IgG (ab6709, Abcam, 1 µL) was used to enhance the signal. The cells were collected using a cell scraper." (https://cellandbioscience.biomedcentral.com/articles/10.1186/s13578-022-00812-8)
"Secondary antibody (Rabbit Anti-Mouse IgG H&L: abcam, ab6709) was diluted 1:100 in dig wash buffer and cells were incubated at RT for 60 min." (https://www.sciencedirect.com/science/article/pii/S0304383521006170)

Anti-Rb IgG Antibody (Sigma-Aldrich, SAB3700890)
The specificity for rabbit immunoglobulins was validated by the manufacturer
https://www.sigmaaldrich.com/DE/en/product/sigma/sab3700890
"This product was prepared from monospecific antiserum by immunoaffinity chromatography using Rabbit IgG coupled to agarose beads followed by solid phase adsorption(s) to remove any unwanted reactivities. Assay by immunoelectrophoresis resulted in a single precipitin arc against Anti-Guinea Pig Serum, Rabbit IgG and Rabbit Serum. No reaction was observed against Goat, Human and Mouse Serum Proteins."

Anti-Rb IgG coupled to AF555 Antibody (ThermoFisher, A21430)
Cited in 125 peer-reviewed article as of 12.09.2022
The specificity for rabbit immunoglobulins in IF was validated by the manufacturer.

https://www.thermofisher.com/antibody/product/Goat-anti-Rabbit-IgG-H-L-Cross-Adsorbed-Secondary-Antibody-Polyclonal/A-21430
"F(ab')2-Goat anti-Rabbit IgG (H+L) Secondary Antibody, Alexa Fluor 555 was used at concentration of 4µg/mL in phosphate buffered saline containing 0.2 % BSA for 45 minutes at room temperature. [...] No nonspecific staining was observed with the secondary antibody alone (panel f), or with an isotype control (panel e)."

Anti-Mouse IgG (H+L) Antibody (Jackson ImmunoResearch, JIM-715-035-150)
Cited in 534 peer-reviewed article as of 12.09.2022
The specificity for mouse immunoglobulins was validated by the manufacturer
https://www.jacksonimmuno.com/catalog/products/715-035-150
"Based on immunoelectrophoresis and/or ELISA, the antibody reacts with whole molecule mouse IgG. It also reacts with the light chains of other mouse immunoglobulins. No antibody was detected against non-immunoglobulin serum proteins."
"Suggested Working Concentration or Dilution Range: 1:10,000 - 1:200,000 for Western blotting with ECL substrates"

Anti-Rabbit IgG (H+L) Antibody (Jackson ImmunoResearch, JIM-711-035-152)
Cited in 845 peer-reviewed article as of 12.09.2022
The specificity for rabbit immunoglobulins was validated by the manufacturer
https://www.jacksonimmuno.com/catalog/products/711-035-152
"Based on immunoelectrophoresis and/or ELISA, the antibody reacts with whole molecule rabbit IgG. It also reacts with the light chains of other rabbit immunoglobulins. No antibody was detected against non-immunoglobulin serum proteins."
"Suggested Working Concentration or Dilution Range: 1:10,000 - 1:200,000 for Western blotting with ECL substrates"
Successfully used in a closely related dipteran species, Drosophila melanogaster in western blot applications (3 peer-reviewed citations)

# Eukaryotic cell lines

Policy information about cell lines and Sex and Gender in Research

| | |
|---|---|
| Cell line source(s) | AG55 cells were kindly provided by Prof. Mike Adang. |
| Authentication | Cell lines were authenticated by RNA-seq. |
| Mycoplasma contamination | Cells were regularly tested for mycoplasma (MycoAlert PLUS Mycoplasma Detection Kit, Lonza LT07-701). All test were negative. |
| Commonly misidentified lines (See ICLAC register) | No commonly misidentified cell lines were used in this study |

# Animals and other research organisms

Policy information about studies involving animals; ARRIVE guidelines recommended for reporting animal research, and Sex and Gender in Research

| | |
|---|---|
| Laboratory animals | Mosquitos were blood-fed with CD-1 mice of both sexes aged 8-56 weeks. Mice were maintained in social groups of 4-5 individuals with 12h/12h dark/light cycle, 22°C temperature and 50 +/- 10% humidity.<br><br>Anopheles: Pupae used for RNAseq were collected 12h after pupariation. Adult mosquitoes used for immunostainings were 2 - 5 days old. For embryos, different stages (hours after oviposition) were studied, which is specified precisely in the respective Figures and legends. |
| Wild animals | The study did not involve wild animals. |
| Reporting on sex | Not relevant, since laboratory animals were only used for blood-feeding of mosquitos, which is required for female egg laying / husbandry. |
| Field-collected samples | The study did not involve field-collected samples |
| Ethics oversight | We have complied with all relevant ethical regulations regarding the use of animals for this project authorized by the French ministry of higher education, research and innovation under the number APAFIS#20562- 2019050313288887 v3. |

Note that full information on the approval of the study protocol must also be provided in the manuscript.

