## [Peer Review File · Nature]

Manuscript Title: The sex-specific factor SOA controls dosage compensation in *Anopheles* mosquitoes

Reviewer Comments & Author Rebuttals

Reviewer Reports on the Initial Version:

Referees' comments:

Referee #1 (Remarks to the Author):

This is a well written manuscript that reports the discovery and characterization of a novel sex-specific factor SOA that established X-chromosome dosage compensation in *Anopheles gambiae*. I agree with the authors about the novelty and of high impact of the work and I think the evidence presented is in general very strong. Therefore, I recommend acceptance after the following major and minor revisions.

Major

- 1) Although SOA knockout does not result in male lethality, it is not known whether or not ectopic expression of the male-specific SOA isoform could result in female lethality in vivo. If so, their evolutionary model will need to be re-evaluated as the fitness cost would be extremely high in females prior to the evolution of the sex-specific splicing of SOA.
- 2) Fig. 1A, please spell out the number of replicates per time points. Please also spell out the specific time instead of preZGA, earlyZGA and late ZGA.
- 3) Fig 1C. I appreciate the need for brevity. However, it will be useful to see a phylogeny of SOA together with its paralogs in the 4 *Anopheles* and *Aedes aegypti* and *Culex quinquefasciatus* in the supplement. I am intrigued by the low % identity between *gambiae* and *albimanus* SOA. Did the paralogues evolve at a faster or slower rate than SOAs in *Anopheles*? Although not absolutely necessary, it'll be nice if they can show that the *albimanus* SOA can also regulate dosage compensation.
- 4) Figure 1e, explain why higher shift of mobility is observed when molarity increases.
- 5) Figure 1j. It's not clear to me that the authors can claim that signals are in specific compartments based on images presented. Perhaps I missed some supplemental data? I don't think this is an important point as the specificity of SOA is supported by other analyses. Why were Malpighian tubules selected for this analysis?
- 6) Fig. 2i, briefly define/explain FIMO. As far as I remember, *Anopheles* X is homologous to one of the arms of *Aedes* chromosome 1, not the entire chromosome. It may be interesting to analyze the two arms separately.
- 7) Fig 3b, why is there a large cluster of autosomal genes showing down-regulation with SOA1-1265?
- 8) Fig. 3d, it's not clear which bar represents the median in the violin plot. Perhaps use a different color to indicate the median?
- 9) Is there a way to assess the level of overlap or congruence between genes or even peaks showing SOA binding in vivo (figure 2b) and in cells (fig 3f)?
- 10) Fig. 4a, I assume the SOA-KI males are homozygous for the knockout?

Minor:

- 11) Line 15-16 and later, change "n=1 for XY males and n=2 for XX female;" to "1 in males and 2 in females"?
- 12) Line 18, Jiang et al. 2015 (doi:10.1093/gbe/evv115) demonstrated complete dosage compensation in *Anopheles* prior to references 3-5 and should be cited.
- 13) Line 227, approximately how many generations did it take for the SOA-KI allele to disappear?

- 14) Para starting on line 235. Are these homozygous knockouts? Does heterozygous SOA_KI have a phenotype?
- 15) The paragraph starting on Line 306. The ectopic expression of Guy1 in *Anopheles stephensi* up-regulate X chromosome genes and results in female-specific lethality (Criscione et al., 2016, eLife; Qi et al., 2019, eLife). This is relevant and should be discussed.
- 16) Lines 386-388, please explain.

Referee #2 (Remarks to the Author):

This is a very interesting study identifying the SOA gene of *A. gambiae* as a regulator of dosage compensation in male mosquitos. This is a landmark finding that will have broad interest. Prior gene expression studies suggested that at least some compensation occurs in these mosquitoes but did not address mechanism or timing of onset. The present study identified the time of onset and used RNA sequencing to identify SOA by male-preferential expression during this developmental window. SOA arose by tandem duplication and acquisition of an intron. Intriguingly, the authors identified differential splicing of SOA in males and females. Differential splicing is presumed to stabilize transcript accumulation in males, which also produce a full-length protein. Ectopic expression of the female splice form, capable of producing a truncated protein, is without effect in female cells. In contrast, expression of the male splice form in female cells increases expression from the X chromosome. Mutation of SOA is not lethal but results in developmental delay. These are exciting findings, and certainly worthy of publication in a high end journal. What is troubling is the assumption that plausible but not rigorously demonstrated things are true.

Immunofluorescence visualization of SOA reveals a loose domain in the male nucleus, but not the female nucleus. This is assumed (but not demonstrated) to be X chromosome. The authors should not claim that a large blob in the nucleus is the X chromosome without additional support (double labeling using FISH or antibody staining of mitotic or polytene preparations). Concern over this is reduced by the finding that SOA is enriched on the X by genomic methods. But there are abundant examples of immunofluorescence and genomic methods producing different conclusions. Could be a surprise here.

Readers are led to the idea that SOA binds CA repeats to recognize the X chromosome. This is implied in the results but stated bluntly in the discussion, lines 273-4. This may be what is happening, but the DNA binding specificity of SOA has not been determined. The nature of the DNA fragment used for EMS is not revealed. Furthermore, the authors have not demonstrated the necessity of the SOA DNA binding function for dosage compensation! The most important finding of this study is the discovery of a gene that regulates mosquito dosage compensation. Although I do not believe that a full mutational analysis of SOA is warranted for publication of this finding, it is essential that the authors eliminate unproven claims, such as this one.

It is clear that at least some of the shorter form of SOA is made in females, but the abundance of SOA protein isoforms in males and females is not presented. The RT-PCR shown in figure 1 suggests that there is considerable splicing to the male form in females, but this is certainly not a quantitative measure. The authors should include a western blot to support their otherwise outstanding analysis.

Minor (but still important) quibbles

The abstract claims that the authors "reveal the DC pathway". This suggests that the mechanism is uncovered, which is untrue.

Line 23-24 wording suggests that alternative splicing of MSL2 regulates expression. In fact, as stated much later in the discussion, translational repression of MSL2 by Sxl (which is alternatively

spliced) regulates expression.

Line 26 states that H4K16ac results in two-fold upregulation in male flies. This mark is thought to represent only one of the mechanisms that contribute to compensation and does not by itself achieve a two-fold increase in expression.

Speculation on the evolutionary origin and spread of SOA detracts from this manuscript. The authors assume that SOA arose within the context of differentiated sex chromosomes, but, in fact, may have preceded or even enabled sex chromosome differentiation.

Is the SOA promoter itself a binding site? Positive feedback does stabilize other dosage compensation systems and it would be quite satisfying if this were another example.

I stumbled over cryptic legends. differential expression (DE)

Colors in 2J are difficult to distinguish.

Protocol for IP followed by Western blot seems to be copied from a ChIP protocol as it includes a proteinase K step (line 588).

Referee #3 (Remarks to the Author):

SUMMARY

The current manuscript from Kalita et al. presents a first description of the dosage-compensation system in the malaria mosquito *Anopheles gambiae*. In the current study the authors identify the gene and protein SOA (sex chromosome activation) and dissect its role in dosage compensation. The compensation is mediated by the global upregulation of X-linked genes and thus similar to the well-studied dosage-compensation system mediated by the male-specific lethal complex in *Drosophila*. Although, the mosquito and fly systems share common characteristics such as regulation via alternative splicing, chromosome-specific targeting and chromosome-specific transcriptional upregulation, the mosquito system is based on the novel protein SOA. While the manuscript does not provide mechanistic insights into i) what causes the increased transcription output, ii) other protein or ncRNA factors, iii) or details on the targeting, I find this study of very high interest as it opens a new page in dosage-compensation research both considering evolution of these regulatory systems and the molecular mechanisms involved. This is therefore an important contribution, and it will be of great interest to follow coming advances in unraveling this dosage-compensation system, such as, other complex components, targeting mechanism, mechanism to increase transcript output, histone modifications etc. The manuscript is well-written, easy to follow, and the presented results are of very high general importance, relevance, and interest. I find the paper highly intriguing with exciting and convincing data (with some caveats, questions and comments listed below).

CRITIQUE

1. MINOR/OPTIONAL: It is stated in the introduction, third paragraph, that "where the deposition of Histone H4 Lysine 16 acetylation (H4K16ac) results in an approximately 2-fold increase in gene expression.". Although often phrased, this statement to my knowledge still lacks experimental support. The male X-chromosome is upregulated two-fold and that has been shown, but the MSL-complex (and thus H4K16ac) mediates only part of this, as far as been experimentally shown, see e.g. Hamada et al. 2005, Straub et al. 2008, Zhang et al. 2010, Deng et al. 2009, Vaquerizas et al. 2013, Armstrong et al. 2018, Kim et al. 2018.

2. MINOR: Figure 1a.

a. Please add error bars, e.g., 95% confidence interval.

b. It is stated in the figure legend "All genes were taken into account, irrespective of whether scored as differentially expressed (DE) or not". Please clarify if "all genes" or "all expressed genes" are considered and if the latter indicate how expressed were defined. I suggest adding "n=" for the chromosome arms.

c. Panel 1e. Clarify/motivate the DNA probe used. The figure legend states "147 bp 601-DNA sequence" but lacks further description, sequence, and reference.

d. Panel 1j. Is SOA detected on metaphase X? It is implied that SOA localization to a "distinct subnuclear territory" is the X-chromosome. Although, this is likely to be the case, can this be tested/confirmed?

3. MAJOR: Figure 1g. The "male-specific transcript" is present also in females. This needs to be clarified and more clearly quantified, i.e., the presence of the "male-specific transcript" in females and whether the SOA protein is found also in females. Figure 1i and 1j and the associated table from the mass-spec experiment suggest that this is not the case, but the sensitivity of these experiments are difficult to judge.

4. MAJOR: On the same topic, line 103-104 states "We also generated an antibody against SOA, which we validated to be specific in immunoprecipitation and western blot". I don't agree with this statement, the antibody is not validated by western blot, and this should be included. If possible, add western blot analysis on protein extracts from males compared to females (and include SOA-KI). This is needed both to judge if SOA is present in females and the specificity of the generated SOA antibody. The SOA staining in females Fig 1j and the CUT&Tag on SOA-KI males Fig 4a, suggests that the antibody is specific but the validation of the antibody can be improved.

5. Figure 2a: From the CUT&Tag experiment, 490 peaks were identified in males and 39 in females. Please clarify if these 39 is "background noise" or based on traces of SOA protein in females. If the latter, these peaks are likely to be "high-affinity sites". Is the CA-repeat motif found also in these female peaks and are these peaks preferentially on the X-chromosome?

6. Figure 2: General question and linked to the identification of peaks in females. Are the autosomal peaks also enriched in the CA-repeat motif, i.e., are the peaks on autosomes reproducible SOA binding sites or "background noise"?

7. Figure 2: I suggest including a figure like 3f (the CUT&Tag profile on chromosomes) also for the CUT&Tag using SOA antibody on males and females. This could be included in the main figure or as supplementary. Again, it is important to clarify the specificity of the antibody and the nature of the "female peaks".

8. Figure 3. It is stated that "expression of male SOA is sufficient to induce dosage compensation". This is based on a global increased transcript output from the X-chromosome in female cells upon overexpression of SOA.

a. In panel 3b I get the impression that it is less than 50 "green-dots" i.e., X-chromosomal genes with increased expression. Please comment on this.

b. I also suggest adding these datasets to the supplementary Excel file (Supplementary Data 2) to make it easily accessible to interested readers.

c. In panel 3b there are two differentially expressed genes in SOA[1-229]; SOA and one more (a green dot and a black triangle). Please clarify in the text or correct in the figure. Indicate SOA also in the panel Ag55 with SOA[1-1265].

d. What is the magnitude of the increased expression in panel 3d (relative to autosomes), i.e., the log₂FC median value? Please also clarify whether this is calculated on "all genes" or "all expressed genes". I recommend including these values (the median values) as it is very hard to read out from the figure and these values can be compared to what we know in *Drosophila*. It seems like a modest increase (<10%), but this is hard to see.

9. (MINOR). Lines 180-181 states "SOA-HA binding strength correlated with expression level of the

respective genes (Extended Data Fig. 7d-e)". This is a correlation based on three points and differs from the in vivo result in Fig 2g which shows a difference for cluster 3 but no obvious correlation.

10. Figure 4d: Include values (the median values) on the decrease and also include the comparison X to autosome for all expressed genes.

11. In the Excel File "Supplementary Data 2" please

a. include more information as e.g., chromosomal location.

b. I would also recommend adding the data from the Ag55 female cells with SOA overexpression (Figure 2b) as well as the SOA peak values for all genes.

c. Check and ensure full information in the headings of each column in the excel file. We can probably guess what e.g., "baseMean, log2FoldChange, lfcSE, stat, pvalue, padj" means but to include which log2FoldChange is listed etc. improves the accessibility.

12. Discussion, line 320-322. It is stated "it is likely that other proteins and possibly lncRNAs interact with SOA to achieve DC. These co-factors are unlikely to be sex-specific, as expressing SOA in female cells induces upregulation of X-linked genes in the absence of Yob". I am not sure what is meant by this statement? If we consider what happens upon expression of MSL2 in *Drosophila* females. Still, we argue that the other msls and roX RNAs are indeed sex specific.

13. Discussion, paragraph on the role of SOA in dosage compensation and physiological consequences of its loss. In comparing to the *Drosophila* system, note that some male roX1 roX2 mutant escape lethality, so loss of roX is not completely male-lethal, unlike loss of mle, msl1, msl2, msl3 or mof. Still the average reduction of X-chromosome expression in the roX1 roX2 mutant is as high or even higher than different msl mutants (PMID: 30532158). It has been hypothesized that that tolerance of mis-expression is a common outcome in the evolution of sex-chromosomes and that high tolerance of mis-expression in the absence of functional dosage compensation may be selected for during evolution of sex-chromosomes (PMID: 30532158). This may be considered also in the case of SOA and the X-chromosome in mosquito.

Author Rebuttals to Initial Comments:

Point-by-point response to reviewers Kalita et al. Manuscript 2022-09-14788

We thank all three reviewers for their thoughtful and constructive feedback, which helped to strengthen our manuscript. In addition to the point-by-point response to the comments (see below), we provide a short overview of the new experiments we have added upon revision.

1. Transgenic mosquitos expressing male SOA isoform

We created a new *SOA-R* (for “rescue”) transgenic mosquito line where unspliceable, full length *SOA* cDNA is re-integrated into the *SOA* locus (starting from the loss-of-function *SOA-KI* line). Hence, we “repair” the mutant line, while *SOA* is expressed from endogenous promoter in both males and females. We observe a rescue of the male loss-of-function phenotype and corresponding reappearance of the *SOA* territory by immunofluorescence. Likewise, female *SOA-R* mosquitos show *SOA* localized to an X chromosome territory by immunofluorescence and CUT&Tag and a concomitant global upregulation of the X-linked genes. Lastly, the *SOA-R* females, like the male *SOA-KI* loss-of-function mutant, show a developmental delay.

2. Characterization of the DNA-binding properties of SOA

We have performed a quantitative biophysical characterization of recombinant *SOA* fragment binding to various nucleic acid sequences. These fluorescence polarization assays reveal that *SOA* requires both the DNA binding myb- and the BTB-/POZ domains for high affinity binding to DNA. We complemented these *in vitro* experiments with new CUT&Tag data showing that the myb-domain is necessary for high affinity binding to the X. Lastly, RNaseA treatment and transcription inhibition by actinomycin D showed that X chromosome association is independent from the presence of RNA or active transcription.

3. SOA isoform expression in male and female mosquitos along development

Our revised manuscript provides qPCR quantifications of *SOA* male and female mRNA isoforms along development. This showed that *wild-type* females produce typically around 100-fold less spliced *SOA* compared to males. We have also optimized our immunostaining conditions and now convincingly show that the male-specific X chromosome territory association of *SOA* is found across developmental stages and tissues. Lastly, ectopic DC is induced in females expressing unspliceable *SOA* cDNA from its endogenous locus (*SOA-R* line), strongly supporting that intron retention prevents full-length *SOA* expression in females.

4. Further validation of the SOA antibody

We now performed a mass spectrometry experiment where we found that *SOA* is the only protein bound by the *SOA* antibody, but not the IgG control. We also performed IP and western blots using ectopically expressed *SOA* protein in Ag55 cells. Lastly, *SOA* staining re-appears in the new *SOA-R* transgenic line by immunofluorescence.

5. Co-visualization of SOA territory and the X chromosome by microscopy

We have performed polytene stainings, staining on mitotic chromosomes, co-visualized the X chromosome with *SOA* by CUT&See and also performed co-RNA-FISH with *SOA*-immunofluorescence. The results corroborate that immunofluorescence and genomics produce the same result, namely that *SOA* binds to the male X chromosome.

Referee #1

This is a well written manuscript that reports the discovery and characterization of a novel sex-specific factor SOA that established X-chromosome dosage compensation in *Anopheles gambiae*. I agree with the authors about the novelty and of high impact of the work and I think the evidence presented is in general very strong. Therefore, I recommend acceptance after the following major and minor revisions.

We thank the reviewer for their supportive comments and highlighting the high impact of our work.

Major

1) Although SOA knockout does not result in male lethality, it is not known whether or not ectopic expression of the male-specific SOA isoform could result in female lethality in vivo. If so, their evolutionary model will need to be re-evaluated as the fitness cost would be extremely high in females prior to the evolution of the sex-specific splicing of SOA.

We have generated the requested transgenic mosquitos, where the male SOA cDNA (without introns) has been integrated into the SOA locus. Therefore, both males and females express full-length SOA from the endogenous promoter. Regarding the specific question about the fitness cost: There is no lethality in females upon SOA (male isoform) expression, but there is a developmental delay (and hence, fitness effects of either having SOA in females or not having it in males are comparable). We are excited that this is in line with our evolutionary model presented earlier. The phenotype, RNA-seq, immunofluorescence and CUT&Tag results of this new transgenic line are presented in Figure 5.

2) Fig. 1A, please spell out the number of replicates per time points. Please also spell out the specific time instead of preZGA, earlyZGA and late ZGA.

The number of replicates have been spelled out in the legend. The specific times have been added to the figure. Values underlying the Fig.1a are provided in Supplementary Data 3

3) Fig 1C. I appreciate the need for brevity. However, it will be useful to see a phylogeny of SOA together with its paralogs in the 4 *Anopheles* and *Aedes aegypti* and *Culex quinquefasciatus* in the supplement. I am intrigued by the low % identity between *gambiae* and *albimanus* SOA. Did the paralogues evolve at a faster or slower rate than SOAs in *Anopheles*? Although not absolutely necessary, it'll be nice if they can show that the *albimanus* SOA can also regulate dosage compensation.

As suggested, the full phylogeny has been added (Extended Data Fig. 1c). The relatively low % identity is intriguing and we first checked the evolutionary distances of SOA paralogues in Culicinae to the 4 SOA orthologues or 4 SOA paralogues in *Anopheles* (Pairwise Patristic Distance analysis, results in Supplementary Data 1). In comparison with the 4 paralogues (AGAP005747), the 4 SOA orthologues are more diverged from the Culicinae versions AAEL003075 and CPIJ000030. We also observe a greater evolutionary distance between 4 SOA orthologues (average distance = 0.6040) compared to the 4 paralogues in *Anopheles* (average distance = 0.4028), see Supplementary Data 1. This would indicate to us that SOA evolves faster than its paralogue.

We also performed a dN/dS analysis. Here, there was no clear sign of differences, for example when we performed a codon-by-codon maximum likelihood analysis (Reviewer 1 - Figure 1). Here, none of the codon-positions analyzed reached $p < 0.05$ for positive selection.

Reviewer 1 - Figure 1: For each codon, estimates of the numbers of inferred synonymous (s) and nonsynonymous (n) substitutions for SOA (panel a) or SOA paralogue (panel b) are presented along with the numbers of sites that are estimated to be synonymous (S) and nonsynonymous (N). These estimates are produced using the joint Maximum Likelihood (ML) reconstructions of ancestral states under a Muse-Gaut model¹ of codon substitution and Tamura-Nei model² of nucleotide substitution. For estimating ML values, a tree topology was automatically computed. The test statistic $dN - dS$ is used for detecting codons that have undergone positive selection, where dS is the number of synonymous substitutions per site and dN is the number of nonsynonymous substitutions per site. A positive value for the test statistic indicates an overabundance of nonsynonymous substitutions. In this case, the probability of rejecting the null hypothesis of neutral evolution (P -value) is calculated³. Values of P less than 0.05 are considered significant. For both SOA and SOA paralogue, none of the positions reached significance. Normalized $dN - dS$ for the test statistic is obtained using the total number of substitutions in the tree (measured in expected substitutions per site) and has been plotted in the figure for each site. Maximum Likelihood computations of dN and dS were conducted using HyPhy software package⁴. Codon positions included were 1st+2nd+3rd+Noncoding. All positions containing gaps and missing data were eliminated. There were a total of 2466 positions in the final dataset. Evolutionary analyses were conducted in MEGA7⁵.

In addition to the analyses in *Anopheles*, we also performed comparisons including the SOA paralogues present in *Culicinae*. In none of the analyses, we obtained a $p < 0.05$ indicative of positive selection. Instead, significance was scored for neutral evolution or purifying selection.

In our view, these analyses are a bit off from the major storyline of the paper (also see a comment from Reviewer#2) and we therefore decided to not include the dN/dS data in the manuscript (of course we can add it, if the reviewer would recommend).

We also thank the reviewer for suggesting to investigate whether SOA in *Anopheles albimanus* can regulate dosage compensation. Our experiments showed that our antibody unfortunately does not recognize *A. albimanus* SOA (data not shown, but can be added to the manuscript, if requested). Therefore, we could not add immunofluorescence and/or CUT&Tag data in *A. albimanus* to the manuscript. Further experiments are needed to assess how the different SOA versions achieve DC in their respective genome. We hope that the reviewer agrees that those are beyond the scope of our manuscript but are rather exciting avenues for follow-up studies.

4) Figure 1e, explain why higher shift of mobility is observed when molarity increases.

This experiment has been performed with a 147 bp long dsDNA sequence, which - simply by its length - will contain multiple sites for SOA to bind DNA. At the protein concentrations used, there is no formation of a defined complex once molarity increases, and hence, multiple SOA molecules are bound to a given labeled DNA molecule. This explanation was also added to the Figure legend. We would like to draw the reviewer's attention to the new EMSAs (Extended Data Fig. 3f) and biophysical experiments (Extended Data Fig. 8g) added upon revision, where we have quantified SOA's ability to bind DNA in further detail.

5) Figure 1j. It's not clear to me that the authors can claim that signals are in specific compartments based on images presented. Perhaps I missed some supplemental data? I don't think this is an important point as the specificity of SOA is supported by other analyses. Why were Malpighian tubules selected for this analysis?

The images are 3D views of confocal microscopy (z-stacks). Based on the overlap of the signal with the DAPI staining, we can conclude that the SOA signal is nuclear. In these stainings provided earlier we had also obtained some cytoplasmic staining, but since this signal still remains in the *SOA-KI* mutant (Fig. 4b), we conclude that it is background.

Regarding the subcompartments: the SOA staining does not homogeneously stain the entire nuclear area (like e.g. DAPI, or RNA Pol2, now added to the manuscript in Fig. 1j) but can be observed in one defined area. This is a hallmark of the compensated X. We had initially focused on Malpighian tubules because their nuclei are relatively large and subcompartments can be easily visualized.

For the revised manuscript we have greatly optimized our IF protocol (Fig. 1j) and provided more stainings of other tissues and developmental stages (Extended Data Fig. 5f-h). The results corroborate that SOA localizes to a subnuclear territory corresponding to the X chromosome.

6) Fig. 2i, briefly define/explain FIMO. As far as I remember, Anopheles X is homologous to one of the arms of Aedes chromosome 1, not the entire chromosome. It may be interesting to analyze the two arms separately.

FIMO stands for 'Find Individual Motif Occurrences' and is a tool that is part of the well known MEME motif analysis software suite (<https://meme-suite.org/meme/index.html>). FIMO can search a set of sequences (in our case, the *A. gambiae* and *A. aegypti* genomes) for occurrences of known motifs (in our case, the (CA)₇ motif that was obtained by MEME). We have now added further details in the corresponding legend and rephrased the text to include a bit more detail.

We thank the reviewer for the interesting comment regarding the chromosomal arms. Yes, the X of *A. gambiae* is homologous to only the chr1p arm of *A. aegypti*⁶⁻⁸. We have repeated our analysis, but the result was unchanged - there is no significant enrichment of CA-repeats, if the arms (chr1p) are considered separately (Fig. 3j).

7) Fig 3b, why is there a large cluster of autosomal genes showing down-regulation with SOA1-1265?

Because SOA₁₋₁₂₆₅ is transiently expressed for 48 hours a combination of primary and secondary effects on gene expression are induced. An obvious cause (but it may not be the only one) are transcription regulators encoded on the X chromosome, which regulate target genes across the entire genome. An example is the X-linked *AGAP000189* (transcription initiation factor TFIID subunit 3). It is significantly upregulated in Ag55 cells expressing SOA and also the new *SOA-R* transgenic females, in which we - similar to the cells - also observe downregulated autosomal genes. We have explained this in the text accordingly.

8) Fig. 3d, it's not clear which bar represents the median in the violin plot. Perhaps use a different color to indicate the median?

Thanks for the comment. To clarify this, we have now thickened the median line.

9) Is there a way to assess the level of overlap or congruence between genes or even peaks showing SOA binding in vivo (figure 2b) and in cells (fig 3f)?

Thanks for this question. We have added such analyses to the revised paper (Extended Data Fig. 6g, 7f-g). In brief, we observe a good correlation between the datasets. We focused on the overlaps between genes, since peak calling and evaluation of their significance by DiffBind depends on parameters such as number of replicates and background. Here, there are technical differences between our experiments: Ag55 library (Fig. 3, two replicates for each SOA and control) was processed from freshly growing cells and with an ectopically expressed HA-tagged SOA construct. It contains better signal-to-noise compared to the frozen pupa samples (Fig. 2, Fig. 4, Fig. 5), where we had four replicates for WT males, but two replicates for the other genotypes. Please also note the different antibodies (HA-tag antibody for Ag55, SOA antibody for the pupa). Considering this, the similarities of the profiles are in our view really striking.

10) Fig. 4a, I assume the SOA-KI males are homozygous for the knockout?

Yes, these are homozygous mutants (see the genotyping gel in Extended Data Fig. 9a). We have clarified all legends and the text accordingly.

Minor:

11) Line 15-16 and later, change "n=1 for XY males and n=2 for XX female;" to "1 in males and 2 in females"?

This has been implemented.

12) Line 18, Jiang et al. 2015 (doi:10.1093/gbe/evv115) demonstrated complete dosage compensation in *Anopheles* prior to references 3-5 and should be cited.

This reference has been added.

13) Line 227, approximately how many generations did it take for the SOA-KI allele to disappear?

We have added a plot (Fig. 4e) and Table (Supplementary Data 1) with these observations of the allele frequencies in a mixed (*WT/SOA-KI*) laboratory population over the course of (now) more than a year (17 generations). Instead of the expected 50%, the allele was present at 15% in our last measurement in April 2023.

14) Para starting on line 235. Are these homozygous knockouts? Does heterozygous *SOA_KI* have a phenotype?

Yes, we have analyzed homozygous *SOA-KI* mutants throughout. This has been clarified in the text and legends. So far, we did not find a phenotype in heterozygous *SOA-KI/WT* mutants. In contrast to the homozygous, the heterozygous *SOA-KI* adult males emerge normally. We have also once performed the co-culture experiment with *WT* and heterozygous *SOA-KI* neonate larvae and then precisely monitored the appearance of pupae (Reviewer 1 - Figure 2). Compared to our earlier experiment (April 2022) with homozygous mutants (blue and black lines), there was no difference between *WT/WT* and *SOA-KI/WT* (new experiment, May 2023, orange and grey lines). All pupae had formed within around 24 hours (~1400 minutes). In contrast, in *SOA-KI/SOA-KI* the peak starts later, with the latest pupae emerging after 60 hours. To keep the manuscript concise, we have simply mentioned the absence of a phenotype in heterozygous mutants in the text.

Reviewer 1 - Figure 2: 161 neonate larvae of wild-type males (grey line), and 161 neonate larvae of heterozygous *SOA-KI* males (orange line) were seeded in a single culture for development through the four larval stages (L1-L4). The developmental timing of each of the 2 genotypes was scored by counting the appearance of pupae, which is represented as a cumulative distribution in the figure. The $t=0$ of the x-axis represents the time when the first pupa appeared in the culture. The black and blue lines are from the experiment shown in Fig. 4g and are plotted for comparison. Note that the two experiments, as indicated in the legend, were performed at different times.

15) The paragraph starting on Line 306. The ectopic expression of *Guy1* in *Anopheles stephensi* up-regulate X chromosome genes and results in female-specific lethality (Criscione et al., 2016, eLife; Qi et al., 2019, eLife). This is relevant and should be discussed.

This is an interesting point. Based on synteny and RNA isoform expression (alternative splicing in males vs. females), we have identified one part of *ASTE110389* as the *SOA* orthologue. In the latest assembly, this gene is correctly split into two (*ASTE120_044673*,

ASTE120_034627). The latter corresponds to SOA and displays 46.4% identity to the SOA protein of *A. gambiae*.

Because complete female lethality is observed upon expression of *Guy1*⁹, but not *Yob*¹⁰, it is (besides technical differences such as promoter choice) possible that *A. stephensi* SOA is *per se* a more potent “toxin”. Furthermore, factors and processes regulated by *Yob/Guy1* other than X chromosome DC could be responsible for inducing lethality. Of note, the *Guy1*-expressing females show lethality with 100% penetrance in thousands of analyzed individuals. However, in the RNA-seq, 1 out of 4 replicates showed only very mild X chromosome upregulation. This indicates that the connection between phenotype and X upregulation is not as absolute as one would expect from the strong penetrance. Perhaps there is also a combined effect, where X misregulation and the other processes downstream of *Yob/Guy1* together cause lethality, where interfering with only one of them induces milder phenotypes. An interesting future avenue for researchers working with *A. stephensi* will be to inactivate ASTE120_034627 in *Guy1*-expressing females and assess lethality. The prediction from our work would be that lethality remains - but there could be surprises.

The discussion has been amended.

16) Lines 386-388, please explain.

It has been proposed that the ectopic activation of the X chromosome underlies the female embryo-killing property of *Guy1/Yob*-transgenic strains and that this “... *offers an excellent tool to create transgenic An. gambiae sexing strains, which would allow efficient mass production of male-only generations for field releases in various genetic control programmes.*” (PMID: 30583747,¹⁰) and that “*components of dosage compensation may be explored to develop novel strategies to control mosquito-borne diseases*” (PMID: 30888319,⁹). In light of our results, we do not think that X chromosome misregulation *per se* is lethal and the direct cause for lethality induced by misregulation of *Yob* could be another one. In our view, it will be a fruitful research avenue to identify 1) what is the molecular function of *Yob/Guy1* proteins and 2) why its misregulation induces killing, because those factors would be promising targets for vector control (please also see the previous reply regarding *Guy1*).

We have clarified the discussion paragraph accordingly

Referee #2

This is a very interesting study identifying the SOA gene of *A. gambiae* as a regulator of dosage compensation in male mosquitos. This is a landmark finding that will have broad interest. Prior gene expression studies suggested that at least some compensation occurs in these mosquitoes but did not address mechanism or timing of onset. The present study identified the time of onset and used RNA sequencing to identify SOA by male-preferential expression during this developmental window. SOA arose by tandem duplication and acquisition of an intron. Intriguingly, the authors identified differential splicing of SOA in males and females. Differential splicing is presumed to stabilize transcript accumulation in males, which also produce a full-length protein. Ectopic expression of the female splice form, capable of producing a truncated protein, is without effect in female cells. In contrast, expression of the male splice form in female cells increases expression from the X chromosome. Mutation of SOA is not lethal but results in developmental delay. These are exciting findings, and certainly worthy of publication in a high end journal.

We thank the reviewer for their positive comments. We were pleased to read that our study was considered a “landmark finding” that is “very interesting” and “exciting”.

What is troubling is the assumption that plausible but not rigorously demonstrated things are true. Immunofluorescence visualization of SOA reveals a loose domain in the male nucleus, but not the female nucleus. This is assumed (but not demonstrated) to be X chromosome. The authors should not claim that a large blob in the nucleus is the X chromosome without additional support (double labeling using FISH or antibody staining of mitotic or polytene preparations). Concern over this is reduced by the finding that SOA is enriched on the X by genomic methods. But there are abundant examples of immunofluorescence and genomic methods producing different conclusions. Could be a surprise here.

We have now added several experiments to support that immunofluorescence and genomic methods produce the same conclusion, namely that SOA binds to the X chromosome.

- **CUT&See:** This assay is based on the spatial CUT&Tag protocol ¹¹, in which the oligos used for tagmentation by pA-Tn5 are fluorophore-labelled. With CUT&See, we could visualize the tagmented DNA sequences together with SOA antibody staining *in situ* and found that the two signals overlap (Extended Data Fig. 6b).
- **Polytene squashes** (Extended Data Fig. 6a) showing localization of SOA to one specific chromosome in males but not females (please note that unlike *Drosophila*, polytene preparations in male *Anopheles* are not a routine technique, and the prepared squashes are smaller/less endo-replicated).
- We have also visualized SOA on **mitotic cells** of mosquito embryos (Extended Data Fig. 5h). Here, we observe that interphase cells contain a male-specific SOA territory, which is lost during mitosis. We would like to note though that fixation is known to produce artifacts (release of DNA binding proteins) specifically in mitotic cells ¹². We have therefore refrained from strong speculations about mitotic inheritance of SOA-mediated dosage compensation in our text.
- **Co-visualization** of a X-linked transcription site by **RNA-FISH** with SOA antibody immunostaining (Fig. 2a). Also here the signals overlap.

Besides that, we have also optimized our immunofluorescence staining procedure and expanded our analyses (Fig. 1j, Extended Data Fig. 5f-g). The male-specific territory can be observed in multiple developmental stages and tissues. Of note, the SOA territory also appears in the new SOA-R transgenic females (Fig. 5c) concomitant with ectopic SOA association with the X chromosome by genomic methods (Fig. 5d and following).

Taken together, we hope that the reviewer agrees with us that this new data is in full support of the conclusion that the subnuclear territory of SOA in immunofluorescence corresponds to the X chromosome.

Readers are led to the idea that SOA binds CA repeats to recognize the X chromosome. This is implied in the results but stated bluntly in the discussion, lines 273-4. This may be what is happening, but the DNA binding specificity of SOA has not been determined. The nature of the DNA fragment used for EMS is not revealed. Furthermore, the authors have not demonstrated the necessity of the SOA DNA binding function for dosage compensation!

We have now added the DNA sequence used in EMSA to the methods (Supplementary Data 1). We have also performed a complementary EMSA experiment, where we used X-linked gene promoter sequences as a probe and included also other SOA fragments (Extended Data Fig. 3d-f).

To quantify the DNA binding specificity of SOA, we used fluorescence polarization to assess their ability to bind DNA (Extended Data Fig. 8g). This new data shows that the isolated SOA myb-DNA binding domain (1-112) exhibits weak affinity (μM range) towards DNA sequences. In contrast, the longer 1-331 fragment (which includes the BTB/POZ domain) bound with nM affinity to DNA. This is in agreement with cooperativity provided by the BTB - a domain that is not present in the short female isoform - being necessary for high affinity binding to DNA.

To complement these *in vitro* experiments, we assessed the X chromosome association of SOA by immunofluorescence before and after treatment with 1) RNaseA and 2) actinomycin D, which inhibits transcription (Extended Data Figs. 7h and 8a). In both cases, the X chromosome association of SOA remained intact, which is in support of a DNA-guided mechanism for X chromosome recognition.

Lastly, we also investigated the necessity of the myb-DNA binding domain for X chromosome binding. To this end, we performed CUT&Tag in Ag55 cells expressing full-length SOA in comparison with a mutant lacking the myb DNA binding domain ("myb-less SOA"). This new experiment revealed that the myb-domain is necessary for X chromosome binding of SOA (Extended Data Fig. 8h-j).

Altogether, these data are consistent with a model, where X chromosomal DNA binding of SOA provided by the myb-domain is necessary for dosage compensation. However, other co-factors (based on the RNaseA experiment we favor the hypothesis that these are other proteins) are important for SOA to recognize the CA-target sequences on the X. We have now presented the data in our text more carefully incorporating these new findings.

The most important finding of this study is the discovery of a gene that regulates mosquito dosage compensation. Although I do not believe that a full mutational analysis of SOA is

warranted for publication of this finding, it is essential that the authors eliminate unproven claims, such as this one.

We thank the reviewer for their statement that a full mutational analysis is not required. Besides the new experiments added above, we have followed the recommendation and went through our manuscript to eliminate unproven claims.

It is clear that at least some of the shorter form of SOA is made in females, but the abundance of SOA protein isoforms in males and females is not presented. The RT-PCR shown in figure 1 suggests that there is considerable splicing to the male form in females, but this is certainly not a quantitative measure. The authors should include a western blot to support their otherwise outstanding analysis.

Based on the comment of the reviewer, we have now clarified the quantity of the SOA isoforms in the sexes.

Regarding the RT-PCR and the relative **RNA levels** of the spliced and unspliced isoform. We had used a one-step RT-PCR kit in these experiments, where the reverse transcription is primed with a gene-specific primer. Hence, all transcripts can be detected (pre-mRNA unspliced/spliced, with/without polyA tail). Note that in the RT-PCR of the post-embryonic stages the splicing patterns are very distinct (this data was in supplement). We apologize that we were not entirely clear in the description of that experiment, but we have now clarified the text, legends and figure and moved the two panels together (Extended Data Fig. 4b).

Complementary to this, we would like to show to the reviewer the result of a gene-specific RT-PCR experiment with a different primer strategy (Reviewer 2 - Figure 1). Here, the RT and reverse primer binds in exon 4, instead of exon 3 and hence, we can distinguish pre-mRNA (both intron 2 and 3 unspliced) from processed transcripts (intron 2 retained, intron 3 spliced). We conclude that the unspliced form in males corresponds exclusively to the pre-mRNA, while in females the unspliced form is also present as a mature transcript, where intron 3 (but not 2) is removed.

Reviewer 2 - Figure 1: Agarose gel showing RT-PCR products of the SOA intron 2 and 3 splicing in male and female embryos at the indicated stages of embryogenesis. The reactions were conducted with a one-step RT-PCR kit, where reverse transcription is primed with the reverse primer in exon 4 (see scheme on the right). The isoform with retained and excised introns 2 and 3 result in different sized products as indicated.

We thank the reviewer for pointing out that RT-PCR is not quantitative. We have therefore added qPCR quantifications of the polyadenylated mRNA SOA isoforms throughout development. These results were crystal clear: **Males produce about 100-fold more spliced**

SOA mRNA isoform than females, which we consider a very drastic difference (Fig. 1g, Extended Data Fig. 4c, raw values in Supplementary Data 1).

We also extracted the reads spanning splice junctions from our embryogenesis poly(A) mRNA-seq. In the manuscript, this data is provided in Supplementary Data 1 and is here plotted for the reviewer (Reviewer 2 - Figure 2). The sex-specificity of intron 2 splicing in the polyadenylated mRNA population is in our view very clear.

Reviewer 2 - Figure 2: RNA-seq reads spanning exon 2 - 3 junction (intron 2 spliced, green) and exon 3 - 4 junction (intron 3 spliced, grey) are shown as a barplot in females (left) and males (right). The barplot shows the mean read count with overlaid data points reflecting the individual replicates. Intron 2 shows spliced reads predominantly in males, while the following intron 3 is found spliced in both males and females. Note the overall differences in SOA mRNA levels between the sexes.

The fact that sex-specific intron retention prevents SOA expression in females is also corroborated with the new *SOA-R* transgene, where SOA cDNA (introns removed) is expressed from the endogenous locus. In *SOA-R* females, we detect ectopic X chromosome binding in microscopy and by genomic methods (Fig. 5). Of note, *SOA-R* show equal SOA mRNA levels between males and females (Fig. 5b), which provides experimental support that intron 2 promotes overall destabilization of the SOA transcript (likely via the non-sense mediated RNA decay (NMD) pathway).

Taken together, it appears that several co/post-transcriptional mechanisms (splicing, RNA decay) play together to provide robust SOA repression (~100-fold) in females. In other words, we do not have evidence for substantial mRNA amounts enabling full-length SOA production in females.

Regarding the question of the **protein** production in the sexes: The RNA-seq (Fig. 1a) and RT-PCR (now in Extended Data Fig. 4b) was performed on single embryos. Unfortunately, there are no fluorescent markers or visual phenotypes that would allow to separate male and female mosquito embryos *a priori* and therefore, it is technically not possible to analyze the production of the male and female SOA protein in the embryos by bulk methods such as western blot. Instead, we have now performed **immunostainings** on embryos (and later stages), which corroborate the results of the qPCR: **The SOA protein can be only detected**

in males, but not females (Fig. 1j, Extended Data Fig. 5f-g; please note that our antibody was raised against the N-terminus and can *per se* recognize a female SOA protein).

We have made numerous attempts to produce the western blots of males and females at post-embryonic stages, but unfortunately found that the large/Q-rich SOA protein transfers very inefficiently. On top of that, the endogenous SOA protein expression is really low: We were unable to detect it in a bottom-up proteomics approach from nuclear extract without enrichment by IP, which captured around 1300 proteins. If mass spectrometry cannot identify SOA protein in bulk extracts, the sensitivity of western blot is unlikely to be high enough to detect the endogenous SOA protein (see also our reply to Reviewer#3 regarding the sensitivity of mass spectrometry).

Nonetheless, when we immunoprecipitate SOA followed by mass spectrometry, we find that only truncated (1-229) protein can be detected in females (Fig. 1i), which is in full agreement with the RNA isoform quantification by qPCR (Fig. 1g).

Please see our explanation in the text *“Considering the mRNA quantification by qPCR (Extended Data Fig. 4c), our data however suggests that SOA₁₋₁₂₆₅ and SOA₁₋₂₂₉ proteins are mutually exclusive in the two sexes and that SOA₁₋₁₂₆₅ in males is at least 3-6 fold more abundant than SOA₁₋₂₂₉ in females.”*

We hope that these new experiments and explanations added to the manuscript clarify the question about isoforms and SOA expression in the sexes.

Minor (but still important) quibbles

The abstract claims that the authors “reveal the DC pathway”. This suggests that the mechanism is uncovered, which is untrue.

We have rewritten the abstract and discussion paragraphs accordingly.

Line 23-24 wording suggests that alternative splicing of MSL2 regulates expression. In fact, as stated much later in the discussion, translational repression of MSL2 by Sxl (which is alternatively spliced) regulates expression.

Thanks for the comment, the phrasing was indeed confusing. To keep the manuscript concise, we have removed this entirely from the introduction and kept it only in the discussion paragraph.

Line 26 states that H4K16ac results in two-fold upregulation in male flies. This mark is thought to represent only one of the mechanisms that contribute to compensation and does not by itself achieve a two-fold increase in expression.

This has been rephrased to *“... deposition of Histone H4 Lysine 16 acetylation (H4K16ac) contributes to an approximately 2-fold increase in gene expression”*

Speculation on the evolutionary origin and spread of SOA detracts from this manuscript. The authors assume that SOA arose within the context of differentiated sex chromosomes, but, in fact, may have preceded or even enabled sex chromosome differentiation.

Thank you for this valuable comment. Because we have added new data from the SOA-R transgenic line, which experimentally strengthens this aspect of our paper, we have decided to keep this part.

Is the SOA promoter itself a binding site? Positive feedback does stabilize other dosage compensation systems and it would be quite satisfying if this were another example.

Thanks for the interesting question, but there seems no specific CUT&Tag enrichment at the SOA gene itself. If the reviewer requests it, we can add the genome browser screenshot to the paper.

I stumbled over cryptic legends. differential expression (DE)

We apologize for the abbreviations and unclear phrasing. We have now carefully gone through our legends and hope they are better understandable.

Colors in 2J are difficult to distinguish.

This has been changed.

Protocol for IP followed by Western blot seems to be copied from a ChIP protocol as it includes a proteinase K step (line 588).

This mistake has been corrected.

Referee #3

SUMMARY

The current manuscript from Kalita et al. presents a first description of the dosage-compensation system in the malaria mosquito *Anopheles gambiae*. In the current study the authors identify the gene and protein SOA (sex chromosome activation) and dissect its role in dosage compensation. The compensation is mediated by the global upregulation of X-linked genes and thus similar to the well-studied dosage-compensation system mediated by the male-specific lethal complex in *Drosophila*. Although, the mosquito and fly systems share common characteristics such as regulation via alternative splicing, chromosome-specific targeting and chromosome-specific transcriptional upregulation, the mosquito system is based on the novel protein SOA. While the manuscript does not provide mechanistic insights into i) what causes the increased transcription output, ii) other protein or ncRNA factors, iii) or details on the targeting, I find this study of very high interest as it opens a new page in dosage-compensation research both considering evolution of these regulatory systems and the molecular mechanisms involved. This is therefore an important contribution, and it will be of great interest to follow coming advances in unraveling this dosage-compensation system, such as, other complex components, targeting mechanism, mechanism to increase transcript output, histone modifications etc.

The manuscript is well-written, easy to follow, and the presented results are of very high general importance, relevance, and interest. I find the paper highly intriguing with exciting and convincing data (with some caveats, questions and comments listed below).

We thank the reviewer for their supporting comments and considering our findings “highly intriguing with exciting and convincing data”.

CRITIQUE

1. MINOR/OPTIONAL: It is stated in the introduction, third paragraph, that “where the deposition of Histone H4 Lysine 16 acetylation (H4K16ac) results in an approximately 2-fold increase in gene expression.”. Although often phrased, this statement to my knowledge still lacks experimental support. The male X-chromosome is upregulated two-fold and that has been shown, but the MSL-complex (and thus H4K16ac) mediates only part of this, as far as been experimentally shown, see e.g. Hamada et al. 2005, Straub et al. 2008, Zhang et al. 2010, Deng et al. 2009, Vaquerizas et al. 2013, Armstrong et al. 2018, Kim et al. 2018.

This has been rephrased to “*where the deposition of Histone H4 Lysine 16 acetylation (H4K16ac) contributes to an approximately 2-fold increase in gene expression.*”

2. MINOR: Figure 1a.

a. Please add error bars, e.g., 95% confidence interval.

The 95% confidence intervals have now been implemented in Fig. 1a as suggested. Because the figure appears now more busy, we have also added all values and statistical parameters concerning this Figure in Supplementary Data 3.

b. It is stated in the figure legend “All genes were taken into account, irrespective of whether scored as differentially expressed (DE) or not”. Please clarify if “all genes” or “all expressed

genes" are considered and if the latter indicate how expressed were defined. I suggest adding "n=" for the chromosome arms.

The legend has been clarified that our analysis considers all genes with a baseMean expression >0 (as calculated by DESeq2) and hence, genes with a mean count > 0 among replicates are included. Please see Supplementary Data 3 for all statistical parameters including "n=" for the chromosome arms as requested.

c. Panel 1e. Clarify/motivate the DNA probe used. The figure legend states "147 bp 601-DNA sequence" but lacks further description, sequence, and reference.

Thanks for the valuable comment. This experiment was performed very early in our project, before we identified the actual binding elements of SOA by CUT&Tag. Our intention was to simply provide a proof-of-concept experiment for SOA's predicted (= automated domain annotations in vectorbase) DNA binding ability. We have now added the description and sequence to the methods and Supplementary Data 1. The revised manuscript also contains a new EMSA using X-linked promoter sequences as a probe and this recapitulated the earlier findings. To complement the EMSA experiments, we have now also added quantitative biophysical experiments. In those, we determined the binding constant of SOA DNA binding domain to minimal/smaller DNA sequences that contain the motifs identified in the CUT&Tag data.

d. Panel 1j. Is SOA detected on metaphase X? It is implied that SOA localization to a "distinct subnuclear territory" is the X-chromosome. Although, this is likely to be the case, can this be tested/confirmed?

Thanks for this comment. We have now performed CUT&See (Extended Data Fig. 6b), Polytene squashes (Extended Data Fig. 6a), Co-visualization of a X-linked transcription site by RNA-FISH with SOA antibody immunostaining (Fig. 2a), and as specifically asked by the reviewer, immunostainings of SOA on mitotic cells (Extended Data Fig. 5h). Here, we observe that interphase cells contain a male-specific SOA territory, which is lost during mitosis. We would like to note though that fixation is known to produce artifacts (release of DNA binding proteins) specifically in mitotic cells¹². We have therefore refrained from strong speculations about mitotic inheritance of SOA-mediated dosage compensation in our text.

Besides that, we have also optimized our immunofluorescence staining procedure and expanded our analyses (Fig. 1j, Extended Data Fig. 5f-g). The male-specific territory can be observed in multiple developmental stages and tissues. Of note, the SOA territory also appears in the new SOA-R transgenic females (Fig. 5c) concomitant with ectopic SOA association with the X chromosome by genomic methods (Fig. 5d and following).

Taken together, the new data is in full support of the conclusion that the subnuclear territory of SOA in immunofluorescence corresponds to the X chromosome.

3. MAJOR: Figure 1g. The "male-specific transcript" is present also in females. This needs to be clarified and more clearly quantified, i.e., the presence of the "male-specific transcript" in females and whether the SOA protein is found also in females. Figure 1i and 1j and the

associated table from the mass-spec experiment suggest that this is not the case, but the sensitivity of these experiments are difficult to judge.

There was a similar question asked by reviewer#2 (see above) and we have now added several new experiments addressing these points.

The semi-quantitative RT-PCR assays are now presented in a single figure panel allowing the readers to evaluate the splicing pattern in embryonic and post-embryonic stages side-by-side. We have clarified that this is a gene-specific RT-PCR, and hence, all RNA forms (spliced/unspliced/poly(A)/non-poly(A)) are detected. Our revised manuscript now also contains qPCR analyses of the male and female polyadenylated mRNA transcripts (Fig. 1g, Extended Data Fig. 4c) We complemented the qPCR with quantifications of the reads spanning the SOA splice junctions from our RNA-seq dataset along embryogenesis (Supplementary Data 1). From all this data, we conclude that a distinct SOA splicing pattern is established early in embryogenesis and that males produce around 100 times more spliced SOA mRNA compared to females.

Regarding the question of the protein production in the sexes: The RNA-seq (Fig. 1a) and RT-PCR (Extended Data Fig. 4b) was performed on single embryos. Unfortunately, there are no fluorescent markers or visual phenotypes that would allow to separate male and female mosquito embryos *a priori* and therefore, it is technically not possible to quantify the production of the male and female SOA protein in the embryos by bulk methods such as Western blot. Instead, we have now performed immunostainings on embryos (and later stages, Extended Data Fig. 5f-g, Fig. 1j), which corroborate the results of the poly(A) RT-qPCR: The SOA protein can be only detected in males, but not females (our antibody was raised against the N-terminus and can *per se* recognize a female SOA protein).

When we enrich the SOA protein by IP and analyze it by mass spectrometry only truncated (1-229) protein can be detected in females, which is in line with the poly(A) isoform quantification by qPCR. Mass spectrometry is a very sensitive method: Current instruments allow the detection of analytes at concentrations in the femto to attomolar range (10^{-15} - 10^{-18}), which is thereby superior to Western blot where with a good antibody a low picogram range (10^{-12} , for a 130 kDa protein this corresponds to few μ L at 10^{-12} molar concentration) is possible. Because the proteins were IPed prior to Mass Spec, we cannot directly infer the relative abundance of the SOA₁₋₂₂₉ protein in females in comparison with SOA₁₋₁₂₆₅ in males. However, considering the overall SOA mRNA levels (males ~ 3-6 fold higher compared to females) and the immunofluorescence result, we conclude that there must be drastic difference between the sexes.

We have made numerous attempts to produce the western blots of males and females at post-embryonic stages, but unfortunately found that the large/Q-rich SOA protein transfers very inefficiently (if the reviewer requests the negative results of the Western blots, we can provide them). We would also like to mention that the endogenous SOA protein expression is really low: We were unable to detect it in a bottom-up proteomics approach from nuclear extract without IP, which captured around 1300 proteins. If Mass Spec cannot identify SOA protein in bulk extracts, the sensitivity of Western blot is unlikely to be high enough to detect the endogenous SOA protein.

Considering the RNA isoform quantification, immunofluorescence stainings and IP-mass spectrometry data and the data from the *SOA-R* transgene, we conclude that there is no full-length SOA protein (1-1265) present in females. Rather the short (1-229, female) and long (1-1265, male) SOA proteins appear mutually exclusive in the two sexes and this property is controlled by sex-specific intron retention.

We hope that these new experiments and explanations clarify the question of the reviewer regarding isoforms and SOA expression.

4. MAJOR: On the same topic, line 103-104 states "We also generated an antibody against SOA, which we validated to be specific in immunoprecipitation and western blot". I don't agree with this statement, the antibody is not validated by western blot, and this should be included. If possible, add western blot analysis on protein extracts from males compared to females (and include SOA-KI). This is needed both to judge if SOA is present in females and the specificity of the generated SOA antibody. The SOA staining in females Fig 1j and the CUT&Tag on SOA-KI males Fig 4a, suggests that the antibody is specific but the validation of the antibody can be improved.

For the question regarding the presence of SOA protein in females, please refer to the answer of the previous question.

We apologize for the unclear phrasing on (previous) L103-104, but what we meant is immunoprecipitation (IP) followed by western blot: in that experiment we performed IP of ectopically expressed SOA-HA (Ag55 cells) with SOA antibody and detected it by western blot with HA antibody. This showed the ability of our SOA antibody to IP the target protein.

We now added more experiments for antibody validation and hope that this convinces the reviewer that our antibody is specific:

Validation 1) The CUT&Tag experiment in Ag55 cells (Figure 3) was performed with HA-epitope antibody. However, at the same time, we had also created profiles with SOA antibody, which we simply did not add to the manuscript so as not to inflate it too much. A representative screenshot of this data is shown in the following Reviewer 3 - Figure 1, where Ag55 cells (HA vs. SOA antibody) as well as the pupa wild-type and *SOA-KI* (both with SOA antibody) CUT&Tag datasets are compared. The HA and SOA antibody datasets look highly similar and a comparison between Ag55 and pupa is now contained in the manuscript (as requested by Reviewer 1). We do not think that adding yet another dataset (SOA antibody for Ag55 experiment) adds much to the story, except for antibody validation, and we therefore have mentioned this as „data not shown“. However if the reviewer recommends, we can of course upload the data on GEO.

Reviewer 3 - Figure 1: Representative genome browser snapshot of CUT&Tag data obtained with HA-tag and SOA antibody. In the first and second row, Ag55 cells express full length SOA (1-1265) for 48 hours. CUT&Tag was then performed from the same sample with HA or SOA antibodies, respectively. The third and fourth track show SOA antibody tracks in male *wild-type* and *SOA-KI* mutant pupae. A representative X chromosomal region was plotted. All profiles look highly similar, but no binding is obtained in the *SOA* mutant corroborating the specificity of the antibody.

Validation 2) Our revised manuscript now contains Western blots and IP experiments of the ectopically expressed SOA protein in Ag55 cells (Extended Data Fig. 5a-b). Of note, the SOA band disappears when we express a mutant lacking the DNA binding myb-domain, which is the epitope against which the antibody was raised. We have also made numerous attempts to produce the western blots of endogenous SOA (males vs. females, *wild-type* versus *SOA-KI*) suggested by the reviewer. In our optimizations, we found that the large/Q-rich SOA protein transfers very inefficiently and SOA is very low abundant (kindly see the previous point 3 by the reviewer). We conclude, the endogenous SOA protein expression is below the detection limit for our antibody and the Western blot technique.

Validation 3) To identify in an unbiased fashion the protein(s) that our antibody binds to, we performed immunoprecipitation experiments with SOA antibody from male pupa extracts followed by mass spectrometry. For this, we used moderately stringent conditions (600 mM NaCl) to get rid of putative SOA interaction partners. This revealed that SOA was clearly the most highly enriched and significant protein and the only one not present in the IgG control, and hence not a contaminant (Extended Data Fig. 5d-e, raw data in Supplementary Data 1).

Validation 4) In the immunofluorescence stainings and CUT&Tag experiments in the new transgenic *SOA-R* females, the SOA X chromosome association can now be detected (while being absent in wild-type females), providing yet another evidence for the SOA antibody specificity (Fig. 5c).

In summary, we think that the new experiments - in addition to having all antibody experiments validated in *SOA-KI* mutant - provide ample support for the specificity of our antibody against SOA.

5. Figure 2a: From the CUT&Tag experiment, 490 peaks were identified in males and 39 in females. Please clarify if these 39 is "background noise" or based on traces of SOA protein in

females. If the latter, these peaks are likely to be "high-affinity sites". Is the CA-repeat motif found also in these female peaks and are these peaks preferentially on the X-chromosome?

Thanks for this question. The 39 peaks mentioned by the reviewer were the ones with higher signal in females compared to males. 34 of those are located on autosomes. These female-specific peaks are likely background, as they display low enrichment *per se*, but also remain enriched in *SOA-KI* males and *SOA-KI* females (Extended Data Fig. 6f). This is different from the autosomal peaks enriched in males, which seem to be specific (see next point).

6. Figure 2: General question and linked to the identification of peaks in females. Are the autosomal peaks also enriched in the CA-repeat motif, i.e., are the peaks on autosomes reproducible SOA binding sites or "background noise"?

The 39 male-specific peaks on autosomes are reproducible, because they vanish in the *SOA-KI* mutant (Extended Data Fig. 6f). Some, but not all have a CA motif (Extended Data Fig. 6g). They display slightly lower enrichment levels compared to the X and interestingly, seem to be often located towards the chromosome ends (Reviewer 3 - Figure 2). Perhaps this is a result of the special chromosomal organization in *Anopheles*, where the telomeres (and chromocenters) of different chromosomes cluster together¹³.

AgamP4_2L	Anopheles gambiae PEST	2L	13	49364325	AgamP4_3L	Anopheles gambiae PEST	3L	11	41963435	AgamP4_2R	Anopheles gambiae PEST	2R	9	61545105	AgamP4_3R	Anopheles gambiae PEST	3R	2	53200684	
Reviewer 3 - Figure 2: Screenshot created in vectorbase showing the chromosomal location of the genes to which the autosomal SOA peaks were annotated to. Blue lines show peaks associated with genes on the + strand, red lines with the - strand. Some genes have more than one peak assigned to them, hence the difference between the number of peaks listed and number of genes shown in the plot.

7. Figure 2: I suggest including a figure like 3f (the CUT&Tag profile on chromosomes) also for the CUT&Tag using SOA antibody on males and females. This could be included in the main figure or as supplementary. Again, it is important to clarify the specificity of the antibody and the nature of the "female peaks".

This was a nice suggestion and it has been added to the manuscript as requested (Extended Data Fig. 6d).

8. Figure 3. It is stated that "expression of male SOA is sufficient to induce dosage compensation". This is based on a global increased transcript output from the X-chromosome in female cells upon overexpression of SOA.

a. In panel 3b I get the impression that it is less than 50 "green-dots" i.e., X-chromosomal genes with increased expression. Please comment on this.

Thank you for the comment, it is indeed 50 dots. The full DESeq2 results tables are now provided with the manuscript (Supplementary Data 2), enabling readers to easily access the data underlying our plots.

Regarding the absolute number of upregulated X-linked genes: It is important to note that due to the transient approach in the cell culture experiment (Ag55), not all cells in the population will be infected and hence, express male SOA. This does not affect the CUT&Tag performed with HA-tag (cells without SOA simply will not have any antibody binding), but in the RNA-seq this results in a “mixed” transcriptome dataset of cells with and without SOA. We have now clarified this point in the legend. Lastly, we would like to draw the reviewers’ attention to the new experiments in *SOA-R* transgenic mosquitos (Fig. 5), where all female cells express SOA, and therefore, we recover many more X-linked genes with increased expression.

b. I also suggest adding these datasets to the supplementary Excel file (Supplementary Data 2) to make it easily accessible to interested readers.

This was a nice suggestion that we have implemented.

c. In panel 3b there are two differentially expressed genes in SOA[1-229]; SOA and one more (a green dot and a black triangle). Please clarify in the text or correct in the figure. Indicate SOA also in the panel Ag55 with SOA[1-1265].

The SOA constructs contain a C-terminal 2xHA tag followed by a T2A cleavage site and *EGFP* allowing us to easily assess the fraction of cells expressing a given construct in an experiment. We had added *EGFP* to our *Anopheles* genome in the mapping of the RNA-seq. In the MA-Plot one dot corresponds to SOA, the other one to *EGFP*. This has now been clarified in the legend. We did not mark the dot for SOA[1-1265] because it is slightly lower expressed compared to SOA[1-229] and would appear as downregulated in the MA-Plot, which likely causes confusion. As a clarification, we now provide the SOA RNA levels of the two experiments in Extended Data Fig. 6i.

d. What is the magnitude of the increased expression in panel 3d (relative to autosomes), i.e., the log₂FC median value? Please also clarify whether this is calculated on “all genes” or “all expressed genes”. I recommend including these values (the median values) as it is very hard to read out from the figure and these values can be compared to what we know in *Drosophila*. It seems like a modest increase (<10%), but this is hard to see.

The legend has been clarified that our analysis considers all genes with a baseMean expression >0 (as calculated by DESeq2, see above). The new additional Supplemental Tables (Supplementary Data 2 and 3) lists all statistical parameters including “n= “ for the chromosome arms in all experiments and plots as requested. Please note again our previous point that not all cells are expressing SOA in Ag55 cells. Of note, in the new *SOA-R* transgene RNA-seq the median log₂ fold change is much more pronounced.

9. (MINOR). Lines 180-181 states “SOA-HA binding strength correlated with expression level of the respective genes (Extended Data Fig. 7d-e)”. This is a correlation based on three points and differs from the in vivo result in Fig 2g which shows a difference for cluster 3 but no obvious correlation.

This has been rephrased to: “As in the *in vivo* context, SOA-HA associated with gene promoters of active X-linked genes, showed more pronounced enrichment at highly expressed genes, and motif analysis revealed binding to CA-repeats.”

10. Figure 4d: Include values (the median values) on the decrease and also include the comparison X to autosome for all expressed genes.

We have now added a comprehensive table with all the relevant values and statistical comparisons (Supplementary Data 3). Please note that also in *Drosophila*, the ectopic expression of *msl-2* in females causes a rather modest increase in X-linked gene expression (see for example PMID: 30194291¹⁴, ~1.5 fold for *Klp3a*, ~1.2 fold to no detectable increase for *Ucp4a* and *CG5254*, Fig. 4g and S3f of that paper).

11. In the Excel File "Supplementary Data 2" please

a. include more information as e.g., chromosomal location.

This has been implemented.

b. I would also recommend adding the data from the Ag55 female cells with SOA overexpression (Figure 2b) as well as the SOA peak values for all genes.

This has been implemented. Because sometimes a given gene can have two peaks, we have provided the DESeq2 (RNA-seq) and DiffBind (CUT&Tag) results separately, yet for both with Gene-IDs and hence overlaps can be easily created.

c. Check and ensure full information in the headings of each column in the excel file. We can probably guess what e.g., "baseMean, log2FoldChange, lfcSE, stat, pvalue, padj" means but to include which log2FoldChange is listed etc. improves the accessibility.

This has been clarified.

12. Discussion, line 320-322. It is stated "it is likely that other proteins and possibly lncRNAs interact with SOA to achieve DC. These co-factors are unlikely to be sex-specific, as expressing SOA in female cells induces upregulation of X-linked genes in the absence of Yob". I am not sure what is meant by this statement? If we consider what happens upon expression of MSL2 in *Drosophila* females. Still, we argue that the other *msl*s and *roX* RNAs are indeed sex specific.

The reviewer is correct, the *roX* RNAs in *Drosophila* are indeed sex-specific. However, at least MOF, MLE and MSL1 are expressed in both sexes and perform important functions in female flies (e.g. PMID: 22039099¹⁵, PMID: 27183194¹⁶, PMID: 10707979¹⁷). In other words, a pre-existing regulatory complex is brought to the X by association with the sex-specific MSL2 and *roX* RNAs. We have now clarified this passage in the discussion using MOF as (unarguably) clear example:

“Akin to Drosophila MOF, which is expressed in both sexes, these SOA co-factors do not necessarily need to be sex-specific, as SOA expression in females triggers upregulation of X-linked genes without Yob.”

13. Discussion, paragraph on the role of SOA in dosage compensation and physiological consequences of its loss. In comparing to the *Drosophila* system, note that some male *roX1* *roX2* mutant escape lethality, so loss of *roX* is not completely male-lethal, unlike loss of *mle*, *msl1*, *msl2*, *msl3* or *mof*. Still the average reduction of X-chromosome expression in the *roX1*

roX2 mutant is as high or even higher than different msl mutants (PMID: 30532158). It has been hypothesized that that tolerance of mis-expression is a common outcome in the evolution of sex-chromosomes and that high tolerance of mis-expression in the absence of functional dosage compensation may be selected for during evolution of sex-chromosomes (PMID: 30532158). This may be considered also in the case of SOA and the X-chromosome in mosquito.

This is an interesting point. The paper from the Larsson group discusses this issue very nicely, in particular that it remains unclear how very few *roX1 roX2* double mutants can survive, despite other *msl* proteins being lethal. It would be interesting to e.g. perform RNA-seq or H4K16ac ChIP-seq in the few escapers and then identify markers that would allow to distinguish escapers from lethal mutants for analysis at earlier stages. From the mentioned paper it is also clear that *roX1* mutants show expression imbalance, but are viable; but on the other hand the fold changes for *roX1* are really mild compared to what we observe in *SOA-KI*. We have now cited this paper and mentioned the escaper males in the discussion paragraph as follows:

“It is also noteworthy that a lack of DC in Drosophila is not fully incompatible with development: expression imbalance in msl-mutants manifests as early as at few hours of embryogenesis, but lethality only occurs at the larval/early pupal stage and hence around 6 days later, while rare escapers reaching the adult stage can be observed in roX1/2 mutants.”

References

1. Muse, S. V. & Gaut, B. S. A likelihood approach for comparing synonymous and nonsynonymous nucleotide substitution rates, with application to the chloroplast genome. *Mol. Biol. Evol.* **11**, 715–724 (1994).
2. Tamura, K. & Nei, M. Estimation of the number of nucleotide substitutions in the control region of mitochondrial DNA in humans and chimpanzees. *Mol. Biol. Evol.* **10**, 512–526 (1993).
3. Suzuki, Y. & Gojobori, T. A method for detecting positive selection at single amino acid sites. *Mol. Biol. Evol.* **16**, 1315–1328 (1999).
4. Pond, S. L. K., Frost, S. D. W. & Muse, S. V. HyPhy: hypothesis testing using phylogenies. *Bioinformatics* **21**, 676–679 (2005).
5. Kumar, S., Stecher, G. & Tamura, K. MEGA7: Molecular Evolutionary Genetics Analysis Version 7.0 for Bigger Datasets. *Mol. Biol. Evol.* **33**, 1870–1874 (2016).
6. Matthews, B. J. *et al.* Improved reference genome of *Aedes aegypti* informs arbovirus vector control. *Nature* **563**, 501–507 (2018).
7. Timoshevskiy, V. A. *et al.* Genomic composition and evolution of *Aedes aegypti* chromosomes revealed by the analysis of physically mapped supercontigs. *BMC Biol.* **12**, 27 (2014).
8. Dudchenko, O. *et al.* De novo assembly of the *Aedes aegypti* genome using Hi-C yields chromosome-length scaffolds. *Science* **356**, 92–95 (2017).
9. Qi, Y. *et al.* Guy1, a Y-linked embryonic signal, regulates dosage compensation in *Anopheles stephensi* by increasing X gene expression. *Elife* **8**, (2019).
10. Krzywinska, E. & Krzywinski, J. Effects of stable ectopic expression of the primary sex determination gene Yob in the mosquito *Anopheles gambiae*. *Parasit. Vectors* **11**, 648 (2018).
11. Deng, Y. *et al.* Spatial-CUT&Tag: Spatially resolved chromatin modification profiling at the cellular level. *Science* **375**, 681–686 (2022).

12. Teves, S. S. *et al.* A dynamic mode of mitotic bookmarking by transcription factors. *Elife* **5**, (2016).
13. Lezcano, Ó. M., Sánchez-Polo, M., Ruiz, J. L. & Gómez-Díaz, E. Chromatin Structure and Function in Mosquitoes. *Front. Genet.* **11**, 602949 (2020).
14. Valsecchi, C. I. K. *et al.* Facultative dosage compensation of developmental genes on autosomes in *Drosophila* and mouse embryonic stem cells. *Nat. Commun.* **9**, 3626 (2018).
15. Feller, C. *et al.* The MOF-containing NSL complex associates globally with housekeeping genes, but activates only a defined subset. *Nucleic Acids Res.* **40**, 1509–1522 (2012).
16. Chlamydas, S. *et al.* Functional interplay between MSL1 and CDK7 controls RNA polymerase II Ser5 phosphorylation. *Nat. Struct. Mol. Biol.* **23**, 580–589 (2016).
17. Reenan, R. A., Hanrahan, C. J. & Ganetzky, B. The *mle(napts)* RNA helicase mutation in *Drosophila* results in a splicing catastrophe of the para Na⁺ channel transcript in a region of RNA editing. *Neuron* **25**, 139–149 (2000).

Reviewer Reports on the First Revision:

Referees' comments:

Referee #1 (Remarks to the Author):

I appreciate the significant addition and careful revision. I have no further comments and recommend acceptance.

Referee #2 (Remarks to the Author):

This nicely done revision that fills most of the holes in the original work. Validation of the antibody is adequate and additional work supporting the idea that the male X chromosome is bound by SOA are well done. Descriptions and figure legends occasionally leave questions about exactly what was done, or is being compared. Examples are noted below.

In figure 1e note that a smear is produced because the probe "contains multiple binding sites for SOA". This probe is identified as the 147 bp fragment used by Luger for assembling nucleosomes and is presumably not even of mosquito origin. As the authors have not yet demonstrated sequence-specific binding by SOA, perhaps best to walk back this statement.

Lack of detail, often in regard to controls, in figure legends. Note in figure legend if the control is wild type animals, unmodified cells or empty vector.

The legend to figure 1g states "spliced relative to average spliced". Presumably this is a typo?

The legend to figure 3e refers to "control expression Ag55 cells" but the text suggests that empty vector is the control. Clarify in figure legend.

Referee #3 (Remarks to the Author):

With this revised version from Kalita et al. and their description of the dosage-compensation system in the malaria mosquito *Anopheles gambiae*, an excellent paper has been further improved. My initial concerns have all been thoroughly addressed, new data has been added and I acknowledge and highly value the careful description and discussion of the results. I just have some optional/minor comments as listed below.

1. In figure panel 5b, I suggest adding a Y-axis and removing the grey lines.
2. Check the font size and font colors in the figures (e.g. panels 1d). Some fonts might become very small in printed version.
3. I recommend adding a y-axis to the browser figure in panel 3f.
4. Please check and revise the reference list according to journal format including required information (e.g. refs 10, 12, 14, 25, 28 etc), remove capital letters when present on all words in the title (e.g. refs 4,6,24...).
5. In Material and Methods there are some inconsistencies in writing of concentrations in having (or not) a space between the number and the unit and how e.g., the addition of Tween to a buffer is denoted.

6. Line 897: "FP value" is stated without explaining "FP" (Fluorescence Polarization).
7. Line 1065: Spelling of "fluorescently".
8. Maybe not required but I think "WT" should be spelled out first time used.

Author Rebuttals to First Revision:

Point-by-point response to reviewers Kalita et al. 2022-09-14788A

We thank all three reviewers for their constructive feedback along the revision process and their last suggestions to improve the final version of our paper.

Referee #1

I appreciate the significant addition and careful revision. I have no further comments and recommend acceptance.

We thank the reviewer for their positive evaluation of our revision and recommending acceptance.

Referee #2

This nicely done revision that fills most of the holes in the original work. Validation of the antibody is adequate and additional work supporting the idea that the male X chromosome is bound by SOA are well done.

We thank the reviewer for complimenting our revision.

Descriptions and figure legends occasionally leave questions about exactly what was done, or is being compared. Examples are noted below.

In figure 1e note that a smear is produced because the probe "contains multiple binding sites for SOA". This probe is identified as the 147 bp fragment used by Luger for assembling nucleosomes and is presumably not even of mosquito origin. As the authors have not yet demonstrated sequence-specific binding by SOA, perhaps best to walk back this statement.

The reviewer is correct that our understanding of the sequence specificity of SOA-DNA binding is still incomplete. The EMSA data has been put to the Extended Data Figures, while emphasis on this aspect in the text has been decreased.

Lack of detail, often in regard to controls, in figure legends. Note in figure legend if the control is wild type animals, unmodified cells or empty vector.

The legend to figure 1g states "spliced relative to average spliced". Presumably this is a typo?

The legend to figure 3e refers to "control expression Ag55 cells" but the text suggests that empty vector is the control. Clarify in figure legend.

We thank the reviewer for pointing out the specific issues. We have rewritten all Main Figure Legends and amended the Extended Data Legends as necessary.

Referee #3

With this revised version from Kalita et al. and their description of the dosage-compensation system in the malaria mosquito *Anopheles gambiae*, an excellent paper has been further improved. My initial concerns have all been thoroughly addressed, new data has been added and I acknowledge and highly value the careful description and discussion of the results.

We thank the reviewer for considering our manuscript excellent.

I just have some optional/minor comments as listed below.

1. In figure panel 5b, I suggest adding a Y-axis and removing the grey lines.

This has been implemented.

2. Check the font size and font colors in the figures (e.g. panels 1d). Some fonts might become very small in printed version.

The font sizes and colors have now been adjusted according to the instructions provided by the editorial office and art editors.

3. I recommend adding a y-axis to the browser figure in panel 3f.

This whole chromosome coverage track has been generated with IGV using the “group autoscale” option for SOA-HA and control. We zoomed out to the maximum to display all chromosomes. Since the program needs to shrink the information for the whole chromosome, the coverage is averaged out for a bigger fragment of the X. This would mean that for example in figure 3f, the maximum value on the y-scale would be ~22. This might suggest to the reader that the coverage for the highest peak is ~22, which is not the case (as can be seen in Extended Data Figure S7b, f). Therefore, IGV does not provide a value. From the user guide: *“At all but the lowest zoom levels, each pixel represents a significant amount of data. IGV divides the data to be displayed into “windows” of equal length each corresponding to a single pixel, summarizes the values across each window, and then displays the summarized values in the track.”*

We would like to draw the attention of the reviewer to the bigwig coverage files that were provided as part of our GEO datasets. Hence, our genomics datasets can be easily explored by interested scientists.

4. Please check and revise the reference list according to journal format including required information (e.g. refs 10, 12, 14, 25, 28 etc), remove capital letters when present on all words in the title (e.g. refs 4,6,24...).

This will be implemented together with the production team.

5. In Material and Methods there are some inconsistencies in writing of concentrations in having (or not) a space between the number and the unit and how e.g., the addition of Tween to a buffer is denoted.

Thanks for pointing this out, we have gone through this again and ensured consistency.

6. Line 897: "FP value" is stated without explaining "FP" (Fluorescence Polarization).

Thanks, this has been implemented.

7. Line 1065: Spelling of "fluorescently".

Thanks, this has been implemented.

8. Maybe not required but I think "WT" should be spelled out first time used.

Thanks, this has been implemented.